# Connecting multiple microenvironment proteomes uncovers the biology in head and neck cancer

Ariane F. Busso-Lopes [1], Leandro X. Neves [1], Guilherme A. Câmara[1], Daniela C. Granato[1], Marco Antônio M. Pretti [2], Henry Heberle [3], Fábio M. S. Patroni [4], Jamile Sá [1], Sami Yokoo [1], César Rivera [1,5,6], Romênia R. Domingues[1], Ana Gabriela C. Normando[1,5], Tatiane De Rossi[1], Barbara P. Mello[7], Nayane A. L. Galdino[8], Bianca A. Pauletti[1], Pammela A. Lacerda[9], André Afonso N. Rodrigues[10], André Luis M. Casarim [10], Reydson A. de Lima-Souza[5], Ingrid I. Damas[11], Fernanda V. Mariano[11], Kenneth J. Gollob[8,12], Tiago S. Medina [8], Nilva K. Cervigne[9], Ana Carolina Prado-Ribeiro[5,13], Thaís Bianca Brandão[13], Luisa L. Villa[7,13], Miyuki Uno[13], Mariana Boroni[2], Luiz Paulo Kowalski [14,15], Wilfredo Alejandro González-Arriagada [16,17] & Adriana F. Paes Leme [1] ✉

The poor prognosis of head and neck cancer (HNC) is associated with metastasis within the lymph nodes (LNs). Herein, the proteome of 140 multi-site samples from a 59-HNC patient cohort, including primary and matched LN-negative or -positive tissues, saliva, and blood cells, reveals insights into the biology and potential metastasis biomarkers that may assist in clinical decision-making. Protein profiles are strictly associated with immune modulation across datasets, and this provides the basis for investigating immune markers associated with metastasis. The proteome of LN metastatic cells recapitulates the proteome of the primary tumor sites. Conversely, the LN microenvironment proteome highlights the candidate prognostic markers. By integrating prioritized peptide, protein, and transcript levels with machine learning models, we identify nodal metastasis signatures in blood and saliva. We present a proteomic characterization wiring multiple sites in HNC, thus providing a promising basis for understanding tumoral biology and identifying metastasis-associated signatures.

Head and neck cancer is the eighth leading cause of cancer worldwide, and 90% of these tumors are mucosal head and neck squamous cell carcinomas (HNSCC)[1]. HNSCC can severely impact the quality of life of patients due to treatment sequelae and high rates of locoregional recurrences. Unfavorable outcomes are largely related to the presence of lymph node metastasis that reduces survival by approximately 50%, and this is the primary argument supporting the use of elective neck treatment[2]. The detection of lymph node alterations can be challenging, and the discovery of molecular markers that can allow for an accurate identification of locoregional spread would enable clinicians to avoid unnecessary extensive operations and could reduce postoperative morbidity[3].

While cancer research has previously focused on characterizing malignant cells in primary tumor tissues, the investigation of additional environments implicated in HNSCC regulation may lead to an improved understanding of the mechanisms underlying

carcinogenesis[4,5]. In addition to cancer cells, the tumor microenvironment (TME) comprises distinct cell subsets, as the immune portion and cancer-associated fibroblasts (CAFs), and it is of special interest once the intense crosstalk among these heterogeneous populations reprogram key processes responsible for tumor growth and invasion[6]. Additionally, neoplastic cells can enter lymphatic vessels and migrate to lymph nodes where they interact with the host immune environment and establish metastasis[7]. Thus, a better understanding of the molecular signals within the TME and in the metastatic microenvironment may provide insights into tumor biology, thus helping to guide clinical investigations.

Body fluids are able to wire the diverse microenvironments and their composition can also be affected by cancer. Tumor-specific T cells, circulating tumor cells (CTCs), macrophage-like cells, tumor endothelial cells, cancer-associated fibroblasts (CAFs), free molecules, and exosomes have all been identified in the peripheral blood of cancer patients and are valuable in clinical decision-making[8–11]. Saliva has also been proven to be a promising source of biomarkers in HNSCC due to its proximity to tumor lesions[12–14]. Thus, the analysis of fluids or liquid biopsies raises the possibility of probing the molecular profile of tumors in a non-invasive manner and may provide a valuable means of tracking biomarkers in HNSCC.

In this scenario, clinical proteomics has emerged as a promising approach for the identification and quantification of potential markers, leveraging the development of new tools that can be used in clinical practice. Technological advances in proteomics, particularly in the mass spectrometry (MS) field[15], have the power to provide a deeper understanding of the molecular mechanisms and guide the discovery of biomarkers[16–18].

In this work, we use a multisite mass spectrometry-based discovery approach in a 59-patient cohort, and this is followed by a deep biological characterization of the proteomes and application of a multiparametric machine learning model to prioritize targeted molecules. Taken together, this study presents the basis for understanding the response of multiple microenvironments to lymph node metastasis and indicates prognostic signatures in HNSCC.

## Results

### Global proteomes are collectively implicated in immune response

To obtain a comprehensive view of the proteome composition in HNSCC, we selected 27 primary tumors, 27 metastasis-negative or -positive lymph nodes, 24 buffy coats, and 24 saliva samples from a 59-patient cohort to evaluate the protein content using label-free quantitative MS in the discovery phase (Fig. 1a; Supplementary Data 1-1 and 1-2). A histology-guided approach was employed to harvest malignant and non-malignant enriched cell populations in both primary tumor and lymph node tissues. While malignant cells refer to the tumoral or metastatic cells themselves, the non-malignant portion includes microenvironment cells that surround and support the malignant populations[19]. The populations that were evaluated included (i) malignant cells from primary tumors, (ii) non-malignant cells located adjacent to primary tumors (mucosal margins), (iii) malignant cells from metastatic lymph nodes, (iv) non-malignant cells located adjacent to sites of metastasis from lymph nodes, (v) buffy coat samples, and (vi) saliva cell samples. MS quality control measures were utilized for all the experiments (Supplementary Fig. 1). Two primary tumor samples (malignant cells) were excluded due to inconsistent detection of control peptide precursor ions (patients 2875 and 4417), and this resulted in 25 remaining malignant samples from primary tumors that were used for analysis. In total, 140 samples from the multiple sites monitored by DDA had an appropriate quality and were kept in further analysis. We identified an average of 2035 protein groups that exhibited MS signals covering close to five orders of magnitude (Fig. 1b). Malignant cells from primary sites yielded the highest number of

identified proteins ($n = 2444$ proteins), and this was followed by malignant cells from lymph nodes ($n = 2308$ proteins). The ranking according to MS signal revealed a buffy coat proteome with a wider dynamic range among all of the sites evaluated ($n = 2188$ proteins), although different LC gradients were used. A total of 313 proteins were shared across multiple sites (Supplementary Fig. 2a).

We then investigated the HNSCC proteome to retrieve insights from the biology of multiple sites. Proteins from tissues and fluids were separated into clusters (PC: protein cluster) based on the hierarchical relationship among the label-free quantitation (LFQ) intensities using a single clustering parameter for all the six datasets (Ward's method based on Bray-Curtis distance) (Fig. 1c; Supplementary Fig. 2b), and the PCs were associated with Gene Ontology (GO) biological processes (adjusted $p \leq 0.05$; two-sided Fisher's exact test followed by Benjamini-Hochberg correction) (Fig. 1d). It is noteworthy that at least one PC from each region was enriched for immune-related processes that included mainly the neutrophil mediated immunity (GO:0002283, GO:0002446, GO:0043312; -Log$_{10}$ [adjusted $p$] = 10.14 to 66.68). Additionally, several other immune processes were overrepresented for the PCs and involved antigen processing and presentation of peptides via MHC class I or II specifically in tissues (GO:0019886, GO:0002495, GO:0002478, GO:0042590, GO:0002479; -Log$_{10}$ [adjusted $p$] = 5.10 to 29.54), regulation of inflammatory response particularly in fluids (GO:0002673; -Log$_{10}$ [adjusted $p$] = 11.69 to 14.19), regulation of humoral response in buffy coats (GO:0030449, GO:0002920; -Log$_{10}$ [adjusted $p$] = 12.71 to 13.19), and phagocytosis dependent or independent of the Fc-gamma receptor signaling for saliva and lymph nodes non-malignant samples (GO:0038096, GO:0006909, GO:0006911; -Log$_{10}$ [adjusted $p$] = 11.61 to 17.96).

Taken together, these findings revealed specific protein composition for malignant and non-malignant cells derived from primary tumors and lymph nodes, buffy coat, and saliva samples. We also identified subsets of proteins across distinct cell populations that exhibit similar abundance profiles and may modulate common biological functions in HNSCC, particularly immune-related processes.

### Protein profiles indicate specific immune phenotypes across datasets

Based on the observation that proteomes from multiple sites were enriched for immune processes (Fig. 1d), we inferred the immune composition associated with our bulk proteomes using signatures from publicly available single-cell RNA sequencing data in CIBERSORTx version 1.0[20]. Non-immune subpopulations were also inferred from the proteomic data.

Because CIBERSORTx version 1.0 relied on transcriptome matrices to estimate cell types, we first performed a series of analysis to verify if predicting immune composition for proteomes could add any bias to our results. The global proteome levels herein generated for primary tumors (malignant cells) were compared to the transcriptomic levels using data from 500 HNSCC tumor samples retrieved from The Cancer Genome Atlas (TCGA). We observed a moderate degree of correlation between proteome and transcriptome levels ($\rho = 0.53$; $p \leq 2.2E\text{-}16$; two-sided Spearman correlation) (Supplementary Fig. 3a). Subsequently, public scRNASeq datasets comprised of 18 HNSCC tissue samples[5] and a human peripheral blood mononuclear cell (PBMC) sample from a healthy donor (https://www.10xgenomics.com) were used to generate signatures that were further employed as reference matrices in this study (Supplementary Fig. 3b, c; Methods section 'scRNASeq processing and differential expression'). We then analyzed the correlation between protein abundance and gene expression of genes considered in the signature matrix generated for HNSCC[5]. For this, we retrieved data from an expression atlas of healthy tissues[21]. The majority of cell types had median correlation scores around 0.4 ($p \leq 0.05$; two-sided Spearman correlation), showing that the signature

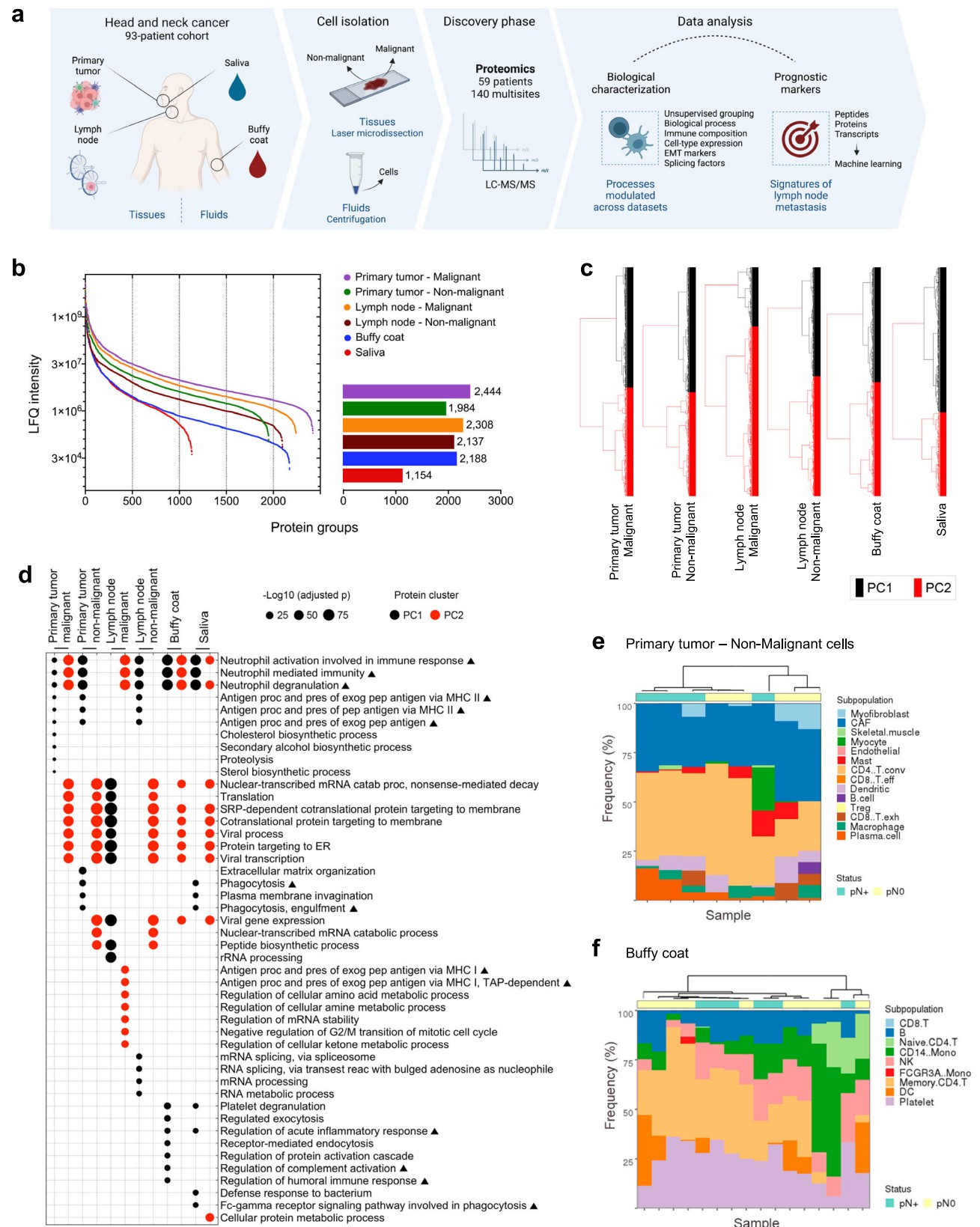

**a** Head and neck cancer 93-patient cohort; Cell isolation; Discovery phase; Data analysis

**b** Protein groups: Primary tumor - Malignant 2,444; Primary tumor - Non-malignant 1,984; Lymph node - Malignant 2,308; Lymph node - Non-malignant 2,137; Buffy coat 2,188; Saliva 1,154

**c** PC1 PC2

**d** Biological processes across cell types

**e** Primary tumor – Non-Malignant cells

**f** Buffy coat

is overall similar across cell types and should not disfavor the estimation of a given subpopulation (Supplementary Fig. 3d). Additionally, to assess the validity of our approach, we deconvoluted proteomic data from Rieckmann et al.[22] using the PBMC reference matrix (https://www.10xgenomics.com) and observed consistent results for each cell type (Supplementary Fig. 3e). These verifications indicated that RNA-based cell signatures could be used to estimate cell types from bulk proteomes in an unbiased manner.

Next, the HNSCC tissue[5] and PBMC (https://www.10xgenomics.com) matrices were applied as references to deconvolute the whole proteome information generated in this study for non-malignant cells isolated from tumors (n = 25 samples) and lymph nodes

**Fig. 1 | Proteomic profile of tissues and fluids in a 59-HNSCC patient cohort.**
**a** Experimental design to uncover the biological aspects and prognostic markers in the proteomes from multiple HNSCC sites. Created with BioRender.com.
**b** Dynamic range of proteomics quantitative data for primary tumor malignant (*n* = 25 samples) and non-malignant (*n* = 27 samples) cells, lymph node malignant (*n* = 13 samples) and non-malignant (*n* = 27 samples) cells, buffy coat (*n* = 24 samples) and saliva cells (*n* = 24 samples). The bar sizes and respective numbers in the right-sided graph indicate the total number of proteins identified per site. **c** Groups identified by clustering of the protein datasets for the multisites using the Ward's method based on Bray-Curtis distance (primary tumor – malignant: 2444 proteins; primary tumor – non-malignant: 1984 proteins; lymph node –

malignant: 2308 proteins; lymph node – non-malignant: 2137 proteins; buffy coat: 2188 proteins; saliva: 1154 proteins). **d** Top-10 significant GO biological processes enriched for the PC groups of the global proteomes (adjusted *p* ≤ 0.05; two-sided Fisher's exact test followed by Benjamini-Hochberg correction). Immune-related processes are labeled with a triangle. Predicted composition of immune populations based on proteomic data of non-malignant cells from primary tumors (**e**) and buffy coat (**f**) samples using CIBERSORTx version 1.0. Samples were clustered using the Euclidean method and Ward.D distance, and the pN status was annotated. EMT Epithelial-mesenchymal transition, HNSCC head and neck squamous cell carcinoma, LC-MS/MS liquid Chromatography with tandem mass spectrometry. Source data are provided as a Source Data file.

(*n* = 27 samples), which used HNSCC tissue matrix as background, and buffy coat (*n* = 24 samples), which considered the PBMC matrix as reference, in CIBERSORTx version 1.0. The subpopulations could not be evaluated for saliva cells due to a lack of a suitable scRNASeq reference. In addition, malignant cells were not included on this inference because they were isolated by laser microdissection and are primarily composed of neoplastic cells. According to the prediction, non-malignant cells from tumors were enriched with high fractions of CD4+ T cells (eight out of eight samples; deconvolution *p* ≤ 0.1) (Fig. 1e). Mast cells, dendritic cells, macrophages, and plasma cells were frequently detected in HNSCC (six to seven out of eight samples; deconvolution *p* ≤ 0.1); however, they were present at lower percentages. It is of interest to highlight the enrichment of the non-immune population of CAFs in all 8 HNSCC tissue samples. This observation indicates that further studies focusing on CAFs may benefit these patients once several targeting strategies have been proposed to block CAF-mediated tumor support[23]. Remarkably, the predominance of T lymphocytes and fibroblasts in HNSCC tumors has been demonstrated previously at the RNA level[5] and agrees with predictions based on the use of tumor proteomes. Buffy coat prediction revealed a high percentage of memory CD4+ T cells, CD14 monocytes, B cells, and NK cells (14 to 15 out of 16 samples; deconvolution *p* ≤ 0.1) (Fig. 1f), thus revealing that distinct immune profiles could be inferred for the tumor microenvironment and blood cells, even though common biological processes were enriched for these proteomic datasets (Fig. 1d). Moreover, the presence (pN+) or absence (pN0) of nodal metastasis could not segregate samples perfectly using hierarchical clustering (Fig. 1e, f). Populations could not be significantly predicted for lymph node non-malignant samples (deconvolution *p* > 0.01). Even though neutrophil-mediated processes have been herein enriched for non-malignant and blood cells in the GO biological processes enrichment (Fig. 1d), we did not explore this population in CIBERSORTx version 1.0 analysis due to the lack of data in the PBMC (https://www.10xgenomics.com) and tissue processed matrix from the literature[5] used as backgrounds.

These analyses represent a detailed immune characterization of the proteome from HNSCC microenvironments and can be used to predict immune populations enriched in multiple sites based on proteomic data.

## HNSCC multisites exhibit immune-associated nodal metastasis markers

We further explored the multiple sites to identify common metastasis-dependent markers (primary tumor – malignant: 11 pN+, 14 pN0 samples; primary tumor and lymph node – non-malignant: 13 pN+, 14 pN0; buffy coat: 11 pN+, 13 pN0; saliva cells: 13 pN+, 11 pN0). Malignant cells from the lymph nodes (pN+) were not included in the analysis, as the patients did not possess the pN0 counterpart to allow for comparisons. A mean of 106 ± 56 differentially abundant proteins was associated with locoregional metastasis across the tissues and fluids (pN+ vs. pN0; *p* ≤ 0.05; two-sided unpaired Student's *t*-test or proteins detected exclusively in one group) (Fig. 2a; Supplementary Data 2-1 to 2-5). The highest number of differentially abundant proteins

was observed in non-malignant cells from lymph nodes (*n* = 201 proteins), and this was followed by malignant cells from the primary tumor (*n* = 110 proteins) and non-malignant cells from the primary tumor samples (*n* = 85 proteins) (Fig. 2a). Additionally, 80 and 54 proteins were associated with nodal status in the buffy coat and saliva samples, respectively (Fig. 2a). Remarkably, malignant cells from primary tumor, buffy coat and saliva had a similar proportion of proteins deregulated between pN+ and pN0 conditions, with a balance between up- and downregulated proteins, whilst non-malignant environments from primary tumors and lymph nodes demonstrated a closer differential profile with an elevated number of proteins highly abundant in the pN+ counterpart (Supplementary Data 2-1 to 2-5). A comparative GO biological process enrichment for the lymph node metastasis proteins highlighted over-represented immune-related terms at multiple sites and were predominantly associated with granulocyte activation (GO:0036230; three datasets; *FDR* = 2.10E-12 to 7.17E-07), leukocytes (GO:0002366, GO:0002443; three datasets; *FDR* = 7.62E-14 to 6.05E-05), myeloid cells (GO:0002275; 3 datasets; *FDR* = 3.78E-14 to 5.14E-07), and neutrophils (GO:0002283, GO:0042119, GO:0002446, GO:0043312; three and four datasets; *FDR* = 7.62E-14 to 6.05E-05) (Fig. 2b). It is becoming increasingly clear that neutrophils possess various functions that dynamically regulate the metastatic cascade, including a role in establishing a premetastatic niche[24] or via neutrophil extracellular traps (NETs) by mediating the trapping of circulating cancer cells[25], or still awakening dormant tumor cells[26]. Our enrichment analysis indicated that neutrophils may be important players in the development of local metastasis in HNSCC.

Considering that an interconnection of multiple environments is necessary to support the tumoral niche[27], we next verified if the metastasis-associated proteins are shared among the five datasets. Twenty-three proteins were significantly associated with lymph node metastasis at two or more sites (Fig. 2c). The 23 proteins are implicated in a series of connected immune-associated GO biological processes (Enrichment *FDR* ≤ 0.05; hypergeometric test followed by FDR correction) (Fig. 2d) that have already been highlighted in the global and metastasis-dependent profiles (Figs. 1d; 2b), and this suggests a role of the immune response in HNSCC multisites. In parallel, some of these proteins are associated with metabolism (Enrichment *FDR* ≤ 0.05; hypergeometric test followed by FDR correction) and reflect the results previously shown in HNSCC metastatic cell-derived extracellular vesicles[28]. Alterations in the metabolic profile have been also demonstrated to be associated with the metastatic potential of cancer cells[29].

Based on the significance of the immune system in lymph node metastasis, we next investigated if the differentially abundant proteins in the proteomes are cluster markers of immune subpopulations. Cluster markers are herein defined as markers that designate populations and were identified across HNSCC tissues[5] and PBMC scRNASeq public data (Supplementary Fig. 3b, c; Methods section 'scRNASeq processing and differential expression'). Then, the cluster markers were compared to the differentially abundant proteins (pN+ vs. pN0) identified from our datasets of non-malignant populations and buffy coat (Supplementary Data 2-2 to 2-4). Transcripts from a subset of

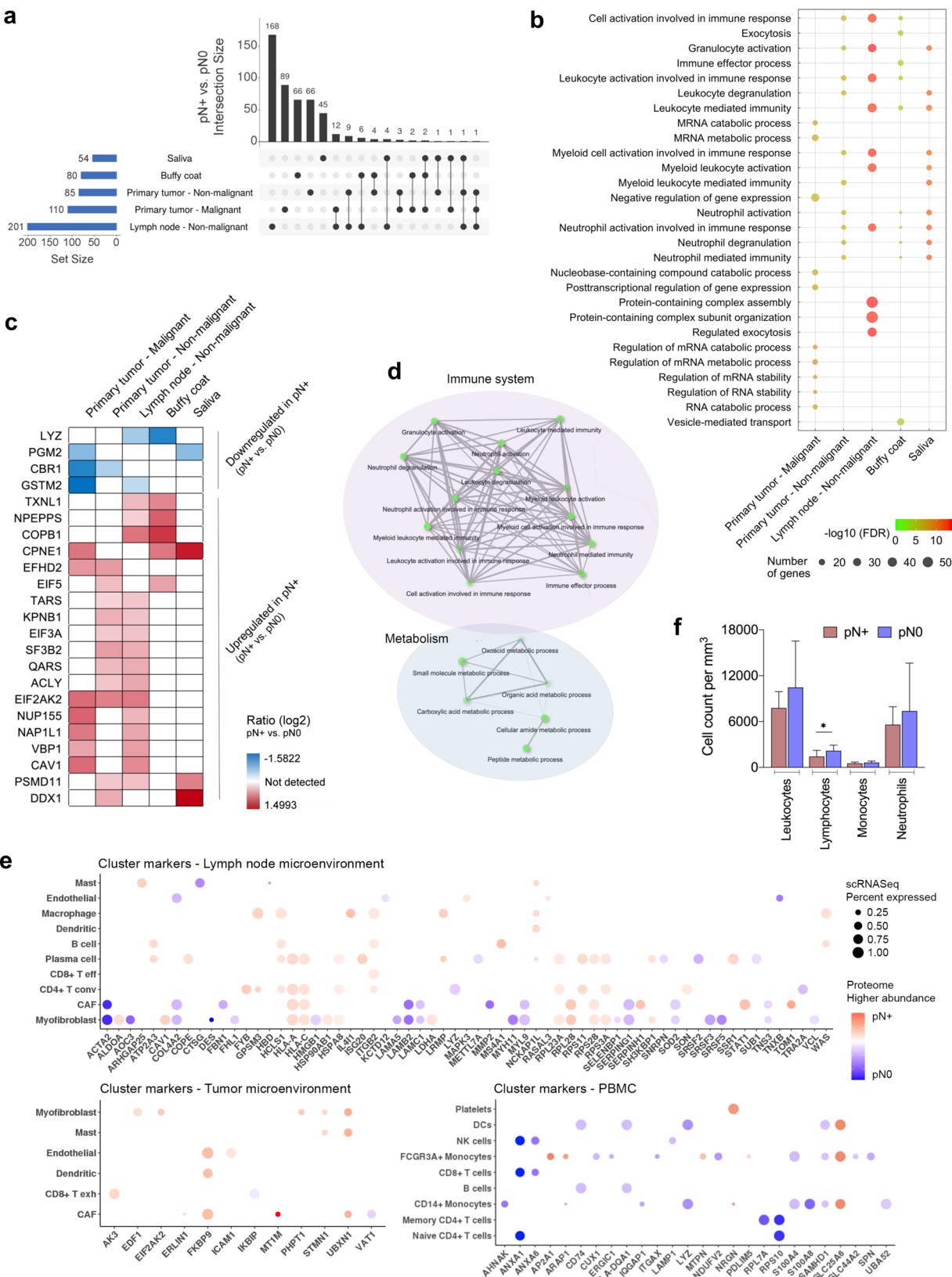

metastasis-related markers were identified as cluster markers of immune subpopulations, and these included 31 genes from a 201-protein signature from lymph node non-malignant cells that were differentially expressed primarily in CD4+ T cells, plasma cells, B cells, and macrophages (Fig. 2e), 5/85 genes from primary tumor non-malignant cells that are cluster markers of dendritic cells, mast cells,

and CD8+ exhausted T cells, and 26/80 molecules from the buffy coat that were primarily identified in CD14+ and FCGR3A+ monocytes (Fig. 2e), thus indicating that immune populations may be modulated in the metastatic phenotype. Remarkably, genes such as *ERLIN1, MT1M, VAT1, FBN1, MMP2, STAT1, TOM1* and *SERPINH1* that were identified as potential metastasis markers in the primary tumor or lymph node

**Fig. 2 | Protein profile and immune characterization of markers associated with lymph node metastasis. a** Upset plot presenting the intersection of differentially abundant proteins associated with lymph node metastasis in the multiple sites (pN+ vs. pN0; $p \leq 0.05$; two-sided unpaired Student's $t$-test or proteins detected exclusively in one group; primary tumor – malignant: $n = 11$ pN+, 14 pN0 patients; primary tumor and lymph node – non-malignant: $n = 13$ pN+, 14 pN0; buffy coat: $n = 11$ pN+, 13 pN0; saliva cells: $n = 13$ pN+, 11 pN0). **b** Combined view of the top-10 GO biological processes overrepresented for tissues and fluids considering differentially abundant proteins between pN+ and pN0 from **a** (Enrichment $FDR \leq 0.01$; hypergeometric test followed by FDR correction). **c** Abundance of 23 proteins significantly associated with lymph node metastasis in multiple sites (pN+ vs. pN0; $p \leq 0.05$; two-sided unpaired Student's $t$-test or proteins detected exclusively in one group). **d** Relationship between GO biological processes significantly enriched for the 23 common proteins listed in **c** (Enrichment $FDR \leq 0.05$; hypergeometric test followed by FDR correction). Darker nodes are more significantly enriched gene sets, bigger nodes represent larger gene sets and thicker edges represent more overlapped genes. **e** Differentially abundant proteins from proteomics data identified as cluster markers of cells from the lymph node microenvironment[5] (upper), the tumor microenvironment[5] (bottom left), and PBMC (bottom right). The size of the circles indicates the percentage of cells expressing the clusters markers in the scRNASeq external dataset[5]. Colors represent the LFQ intensity of the metastasis-associated proteins herein evidenced from the proteomes of lymph nodes or primary tumor non-malignant cells or buffy coat. Higher LFQ levels in pN+ are represented in red and higher LFQ levels in pN0 are in blue. **f** Immune cell count/mm[3] from 9 pN+ and 16 pN0 HNSCC patients. Lymphocyte cell count/mm[3] was significantly associated with lymph node metastasis (pN+ vs. pN0; $p = 0.0261$; two-sided unpaired Student's $t$-test). The results obtained for basophils and eosinophils were not plotted to improve visualization (low cell count), and the values were not statistically different between pN+ and pN0 samples. Data are expressed as mean ± standard deviation. *$p \leq 0.05$. Source data are provided as a Source Data file.

microenvironments are cluster markers that are exclusively expressed by CAF populations, thus indicating that CAFs can provide an excellent source of candidates as metastasis biomarkers for HNSCC (Fig. 2e). We would like to reinforce that detection of clusters markers could not be achieved for saliva cells due to a lack of an appropriate scRNASeq reference.

Next, to strengthen the association between the immune response and lymph node metastasis, we evaluated the clinical data of cell counts obtained from the clinical laboratory in a group of 25 HNSCC patients (9 pN+ and 16 pN0 samples). Although all cell populations were reduced in pN+ when compared to pN0, our results revealed a significant lower cell count/mm[3] of lymphocytes in pN+ ($p = 0.0261$; two-sided unpaired Student's $t$-test) (Fig. 2f). We also correlated the immune subpopulations that were estimated from bulk RNA sequencing (RNASeq) immune data using xCell algorithm[30] with pN outcome. The data was retrieved from TCGA for a 428-HNSCC (180 pN0, 248 pN+) cohort. Interestingly specific immune populations were also associated with the presence of lymph node metastasis with significant correlation coefficients that ranged from −0.23 to 0.12 ($p \leq 0.05$; two-sided Spearman correlation) (Supplementary Fig. 3f).

Therefore, through wiring multiple comparisons among our proteomics data, publicly available datasets and cell counting, we strengthened the relevance of the immune system in HNSCC by identifying a proteome composition that was associated with the metastatic profile and immune populations that potentially express metastasis markers. Based on the relevance of the granulocytes/neutrophils, lymphocytes, dendritic cells, monocytes, and macrophages depicted in the biological process enrichment and public data comparison, we prioritized a list of immune markers based on their identification in the proteomes. From the initially 19 selected targets that represent these specific immune populations, nine potential markers passed the SRM-MS quality control assessment and were verified using targeted approaches and further combined with other targets to investigate the ability to distinguish nodal status using machine learning (ML) models. These immune markers included CD3, CD4, CD8, CD11b, CD14, CD16, CD19, CD45, and CD66b (please see Methods section 'Selection of targets for ML analysis').

## Nodal metastasis cells resemble the molecular signature from tumors

We next sought to explore the protein profile of the malignant and non-malignant populations in the tumor and lymph nodes to elucidate the functional effect of tumor cell spread from the primary site to the lymph nodes. To achieve this, we compared the proteome composition of malignant and non-malignant cells isolated from 11 and 27 matched primary tumors and lymph nodes, respectively (Fig. 3a). Of the 2478 protein groups identified for the malignant portion, 126 proteins were differentially abundant between lymph nodes and primary tumors ($q \leq 0.05$; two-sided unpaired Student's $t$-test followed by

Benjamini-Hochberg correction or proteins detected exclusively in one group) (Supplementary Data 2-6). Considering the non-malignant cells, 2,360 protein groups were identified, and 869 of these were differentially abundant between the two environments ($q \leq 0.05$; two-sided unpaired Student's $t$-test followed by Benjamini-Hochberg correction or proteins detected exclusively in one group) (Supplementary Data 2-7).

Even though the total number of quantified proteins in the malignant and non-malignant portions are similar, the proteins that are differentially abundant between primary tumors and lymph nodes in the malignant microenvironment are reduced when compared to the non-malignant portion (37% vs. 6% of all proteins in the non-malignant and malignant cells, respectively; $p \leq 0.0001$; Chi-square test) (Fig. 3b). Thus, the current proteomics data suggest that less proteomics changes are observed in the primary versus metastatic tumor compartment compared to the non-malignant stromal cells. We also performed a hierarchical clustering analysis to visualize the relationship between tumor and nodal patients' samples in the two populations of malignant and non-malignant cells. The proteomic profile grouped malignant cells from the two regions by patient, while non-malignant cells were clustered according to site (Fig. 3c). This is another evidence showing a more similar proteome pattern between malignant cells from the primary and metastasis sites. Non-malignant cells derived from sample 400, a tonsil tumor, were clearly grouped with non-malignant samples from the lymph nodes (Fig. 3c) and exhibited a similar proteomic pattern that may result from the enhanced lymphoid composition of tonsillar HNSCC. These results may reflect the high heterogeneity and dynamism in the cell constitutions that surround the tumor and metastasis in primary tumor and lymph node sites.

We then determined the GO biological processes overrepresented for the differential proteomes and found distinct processes enriched for malignant and non-malignant portions. Proteins from non-malignant cells upregulated in the lymph nodes were mainly involved in ribosomal and translational processes (Enrichment $FDR = 1.63E-37$ to $7.36E-30$) (Fig. 3d; Supplementary Data 3-1 and 3-2), and this may be related to the heterogeneous cellular composition of tumor and lymph node microenvironments. For malignant cells, the proteome differentially abundant between primary sites and lymph nodes (metastasis) was strongly involved in actin-based cell movement through the deregulation of a group of nine proteins (Enrichment $FDR = 1.22E-04$ to $1.79E-02$) (Fig. 3d; Supplementary Data 3-1 and 3-3). We evaluated the protein or gene levels of these molecules using parallel reaction monitoring-mass spectrometry (PRM-MS) and real-time quantitative reverse transcription PCR (RT-qPCR) in the same set of 11 matched FFPE malignant samples from lymph nodes and primary tumors used in the discovery phase. Two out of the nine proteins were not measured due to the lack of proteotypic peptides, and seven out of the nine genes had no transcript levels detected, resulting in seven

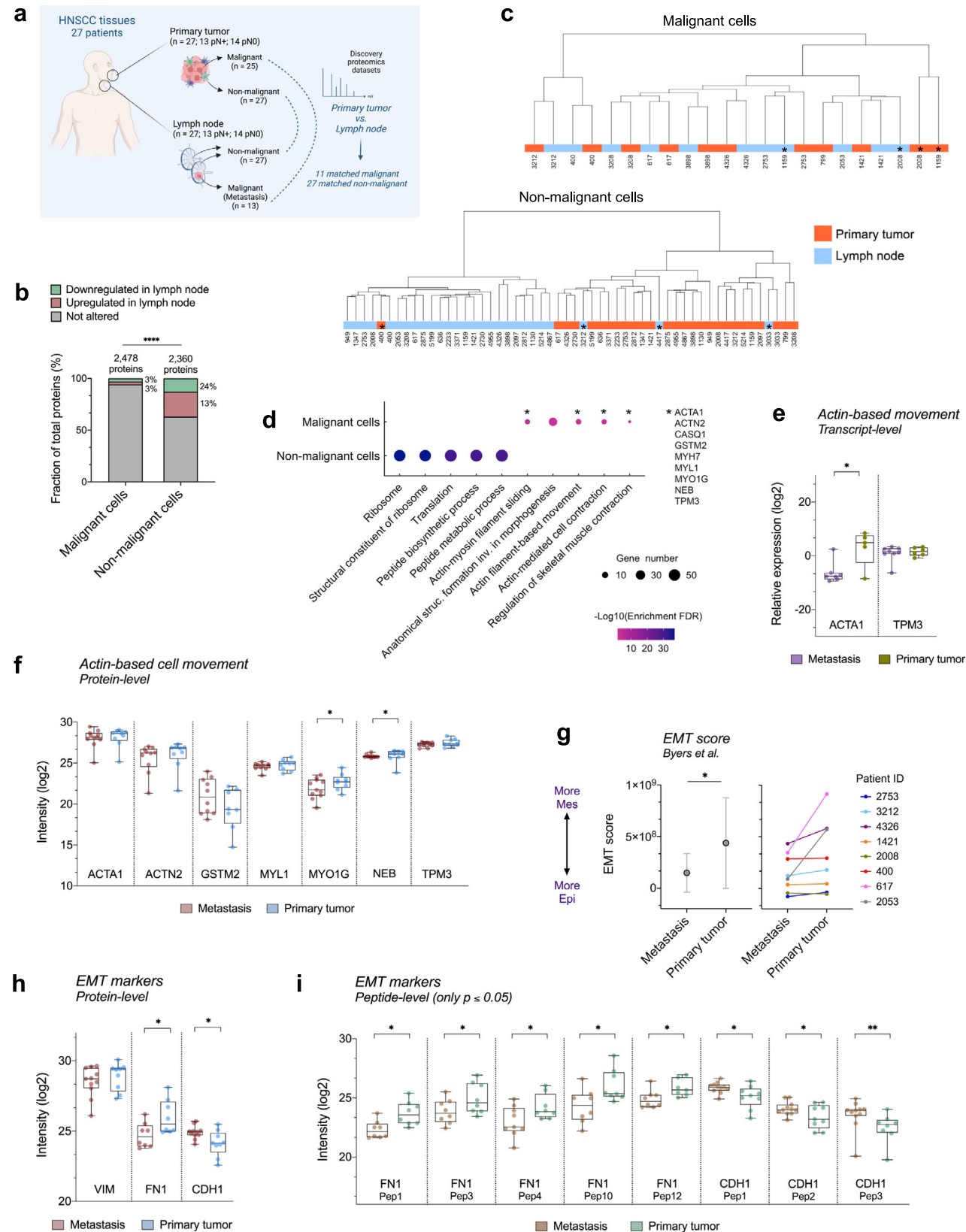

genes/proteins tested: ACTA1, ACTN2, GSTM2, MYL1, MYO1G, NEB, and TPM3. Although the pattern of iRT peptides was consistent across samples in PRM-MS, two primary tumor FFPE tissues with low signal-to-noise ratios (1159 and 2008) were excluded, and the nine remaining matched pairs were considered for subsequent analysis (Supplementary Fig. 4a, b). We confirmed the reduction of *ACTA1*, MYO1G, and NEB

in lymph nodes metastasis at transcript or protein levels (Fig. 3e, f; Supplementary Data 4-1 and 4-4), reinforcing that actin-related proteins are altered during HNSCC spread by independent methodologies, possibly modulating cell motility ($p \leq 0.05$; two-sided Wilcoxon signed-rank test for PRM-MS and two-sided Mann-Whitney test for RT-qPCR).

**Fig. 3 | Comparative proteomics analysis of primary tumor and lymph node sites for malignant and non-malignant cells. a** Experimental design to define the differential protein profile of the malignant and non-malignant populations between the tumors and lymph nodes (matched malignant samples from primary tumor and lymph nodes, $n = 11$ samples per site; matched non-malignant samples from primary tumor and lymph nodes, $n = 27$ samples per site). Created with BioRender.com. **b** Frequency of downregulated or upregulated proteins between tumors and lymph nodes in malignant and non-malignant populations (malignant vs. non-malignant; $p \leq 0.0001$; two-sided Chi-square test). ****$p \leq 0.0001$. **c** Hierarchical clustering when considering the proteomic profile of malignant and non-malignant cells from primary tumor and lymph nodes ($n = 2478$ and $2360$ proteins, respectively). Clustering was performed using the Ward method with Euclidean distance. *Samples that do not follow the main clustering pattern. **d** Top-5 GO biological processes enriched for proteins that were differentially abundant between lymph nodes and primary sites in malignant and non-malignant samples (Enrichment $FDR \leq 0.05$; hypergeometric test followed by FDR correction). *Actin-based cell movement processes modulated in malignant cells through the deregulation of nine proteins. These proteins were selected for verification. Box plots representing the abundance of transcripts and proteins that are involved in actin-

based cell movement using RT-qPCR (**e**) and PRM-MS (**f**), respectively (primary tumor malignant samples, $n = 9$ vs. lymph node matched malignant samples, $n = 9$; two-sided Wilcoxon signed-rank test for PRM-MS and two-sided Mann-Whitney test for RT-qPCR). The boxplots depict the 25–75% interquartile range (IQR) (box limits), with the median shown as a central line; whiskers indicate the minimum and maximum values. *$p \leq 0.05$. **g** EMT scores of HNSCC malignant cells considering the 76-gene signature proposed by Byers et al.[33] in the discovery datasets (primary tumor malignant samples, $n = 11$ vs. lymph node (metastasis) matched malignant samples, $n = 11$; $p = 0.0391$; two-sided Wilcoxon signed-rank test). *$p \leq 0.05$. The left-sided data are mean ± standard deviation and the right image represent individual samples. Box plots representing the abundance of EMT markers at protein- (**h**) and peptide-levels (**i**) in malignant cells using PRM-MS (primary tumor malignant samples, $n = 9$ vs. lymph node matched malignant samples, $n = 9$; two-sided Wilcoxon signed-rank test or two-sided paired Student's $t$-test). Only statistically significant results are shown for peptides. Boxplots depict the 25–75% interquartile range (IQR) (box limits), with the median shown as a central line; whiskers indicate the minimum and maximum values. *$p \leq 0.05$; **$p \leq 0.01$. HNSCC head and neck squamous cell carcinoma, EMT epithelial-mesenchymal transition. Source data are provided as a Source Data file.

Interestingly, the tightly control of motility programmes is critical for establishing the epithelial–mesenchymal transition (EMT) necessary for tumor invasion and metastatic dissemination[31]. During EMT, as previously described in HNSCC at the RNA level[5], tumor cells undergo a loss of intrinsic polarity and lose cell–cell junctions through extensive reorganization of the cytoskeleton and initiation of actin-based cell motility[31]. The alteration of motility proteins may also be related to the acquisition of a mesenchymal-epithelial transition (MET) program in the metastatic site (the reverse transition of EMT) where metastatic cells recapitulate the pathology and became even less dedifferentiated than are their corresponding primary tumors[32]. To gain insights into the EMT and MET programs in HNSCC, we next determined EMT scores in the 11 matched FFPE proteomes evaluated in the discovery phase considering a 76-gene signature computed by Byers et al.[33] (for details, please see Methods section 'Comparison between tumor and metastasis proteomes'). As hypothesized, primary tumor malignant samples had a more mesenchymal ('Mes') phenotype when compared to the lymph node cells, which have a more epithelial ('Epi') profile ($p = 0.0391$; two-sided Wilcoxon signed-rank test) (Fig. 3g). We then verified if the abundances of three canonical EMT markers in the proteomes of the discovery phase were associated with the sites from where the malignant cells were isolated: E-cadherin (CDH1), fibronectin (FN1), and vimentin (VIM) (11 matched FFPE tumor vs. metastasis; $q > 0.05$; two-sided unpaired Student's $t$-test followed by Benjamini-Hochberg correction or proteins detected exclusively in one group; Supplementary Data 2-6). Because the abundance of these proteins between groups was not statistically significant, we evaluated the three proteins in the same cohort using PRM-MS due to its sensitivity and accuracy. Elevated FN1 protein and peptide abundances were observed in malignant populations from primary tumors, indicating the predominance of 'Mes' cells, while CDH1 had higher abundances in lymph nodes and implicated the presence of more 'Epi' phenotypes (nine matched FFPE tumor and metastasis that passed the PRM-MS quality control; $p \leq 0.05$; two-sided paired Student's $t$-test or two-sided Wilcoxon signed-rank test) (Fig. 3h, i; Supplementary Data 4-1 and 4-4). This analysis indicates a role of EMT programs, as well as of the reverse process of MET, in HNSCC spread.

Overall, we clarified the proteome composition that is associated with locoregional metastasis in HNSCC, thus providing a detailed view of potential processes and molecules that are implicated in tumor invasion and spread. Again, the results achieved using proteomics resemble the data obtained using scRNASeq for HNSCC[5], thus indicating that the methods applied in this study effectively identified and quantified protein content patterns that can be used to reveal the heterogeneity of the populations in the nodal and primary sites.

## Microenvironment proteomes group samples according to metastasis and highlight candidate markers of locoregional spread

We further investigated the potential of multisite proteomic profiles in regard to grouping patients according to lymph node metastasis. Hence, we first evaluated the ability of global HNSCC proteomes to separate pN+ and pN0 patients. For that, the whole proteomic contents identified for the multisites (Fig. 1b) were used to calculate the hierarchical clustering in dendrograms, and the samples segregation patterns (patients' clusters C) were associated with clinical and pathological features (2444; 1984; 2137; 2188; and 1154 proteins, respectively, identified across 25 samples of malignant cells from the primary tumor, 27 samples of non-malignant cells adjacent to the primary tumor and lymph node, and 24 buffy coat and saliva samples) (Fig. 4a). A single clustering criterion was considered to group samples from four out of five datasets (Ward Chebyshev), but a distinct method and metric combination was used to segregate the proteomes of malignant cells from primary tumors (Complete Canberra) due to the formation of a highly unbalanced cluster pattern when applying Ward Chebyshev. The cluster behavior provides insights into the analysis of clinical data and, interestingly, C1 and C2 from lymph node microenvironment cells were non-randomly associated with pN status ($p = 0.046$; two-sided Fisher's exact test) (Fig. 4b). Moreover, clusters were significantly associated with smoking habits (primary tumor and lymph nodes: non-malignant, buffy coat; $p = 0.028$, $p = 0.012$, and $p = 0.033$, respectively), pT (primary tumor - non-malignant; $p = 0.021$), desmoplasia status (buffy coat; $p = 0.044$), and overall survival (saliva cells; $p = 0.042$) (two-sided Fisher's exact test or two-sided log-rank tests) (Supplementary Fig. 5a, b). Based on the knowledge that HPV plays a role in the etiology of HNSCC (primarily in oropharynx squamous cell carcinomas)[34], we also evaluated the presence of this virus in the 27 HNSCC primary tumors to determine its association with patient clustering. HPV16 DNA positivity was detected in 29.6% of HNSCC, including five larynx, two oral, and one oropharyngeal tumor (Supplementary Fig. 5c). Within this subgroup, three tumors from the oropharynx ($n = 1$), oral ($n = 1$), and larynx ($n = 1$) sites were positive for $E6/E7$ viral transcripts, thus indicating infection by transcriptionally active HPV (Supplementary Fig. 5c). HPV DNA or RNA positivity was not associated with patient clustering patterns ($p > 0.05$; two-sided Fisher's exact test), thus revealing that other etiological factors were associated with the observed proteomic profiles.

To further explore the proteomic differences between pN+ and pN0, we employed principal component analysis (PCA) using the sets of deregulated proteins from multiple sites (pN+ vs. pN0; $p \leq 0.05$; two-sided unpaired Student's $t$-test or proteins detected exclusively in one group; 201, 110, 85, 80, and 54 proteins from non-malignant cells

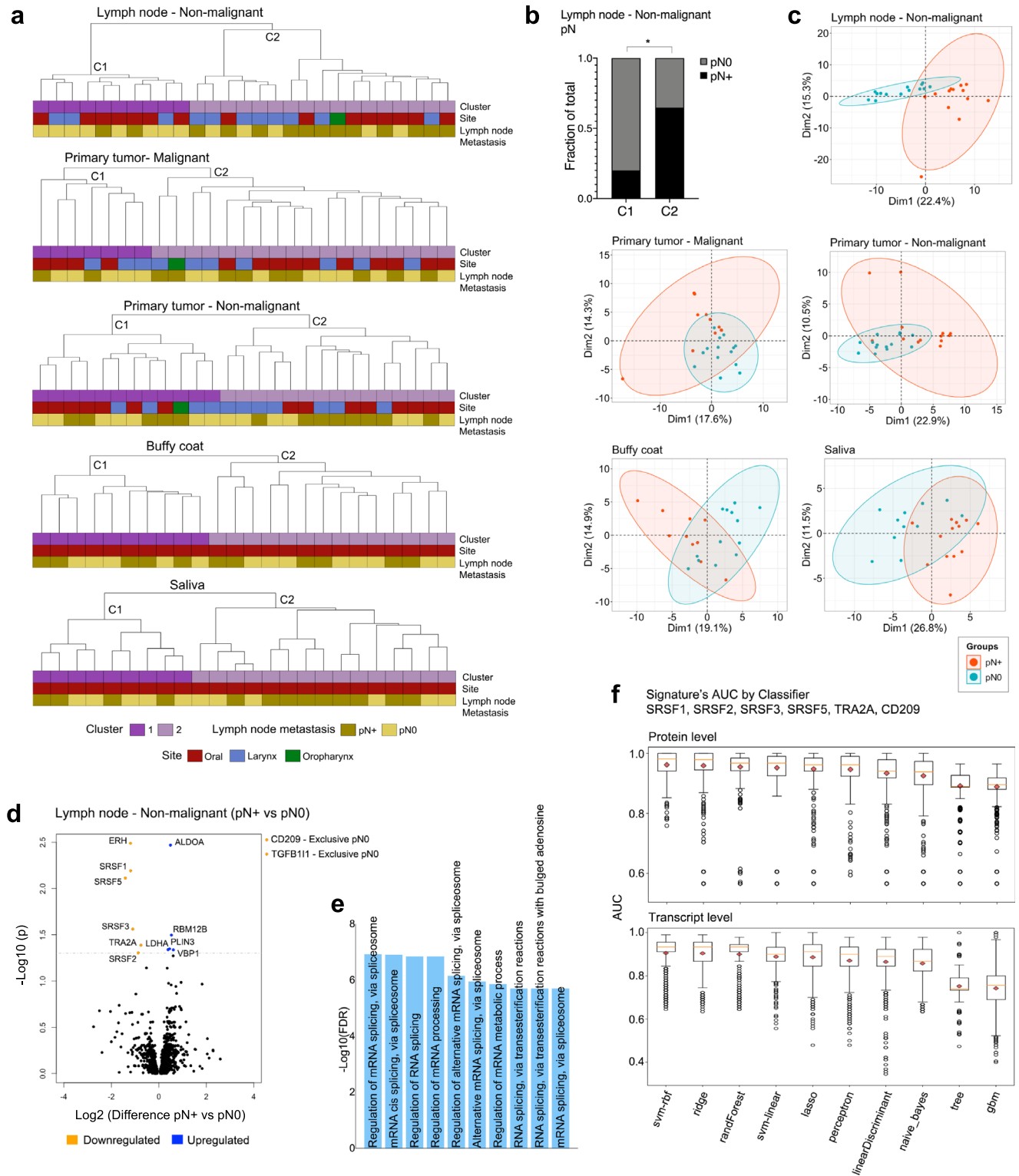

from lymph nodes, malignant cells from primary tumor, non-malignant cells from primary tumor, buffy coat, and saliva, respectively) (Supplementary Data 2-1 to 2-5). The best segregation was observed for the differentially abundant proteins determined for non-malignant cells from lymph nodes that separated the groups according to the first and second components with 22.4% and 15.3% of the total data variation, respectively (Fig. 4c).

Based on the observation that non-malignant cells from lymph nodes exhibited the most promising association with lymph node metastasis status according to hierarchical clustering (global

proteome) and PCA (differential proteome) analysis, we applied *FDR* to control for multiple hypotheses and to filter out proteins that were strongly associated with the metastatic phenotype (pN+ vs. pN0; $q \leq 0.05$; Benjamini-Hochberg test or proteins detected exclusively in one group). Eight downregulated and five upregulated proteins (pN+ vs pN0) had a highly significant differential abundance according to the status of locoregional metastasis ($q = 3.25E\text{-}03$ to $4.91E\text{-}02$ for nine proteins; two proteins were exclusively detected in 50% of pN0 samples: CD209 and TGFB1I1) (Fig. 4d; Supplementary Data 2-8). This set of 13 proteins is primarily involved in mRNA splicing processes

**Fig. 4 | Proteome-grouping pattern associated with nodal metastasis and refinement of targets for ML analysis. a** Clustering of tissues and fluids based on the global proteomic profile (C1 and C2 clusters) (*n* = 59 patients). C1 and C2 groups were generated using Complete Canberra (25 primary tumor – malignant samples: 2444 proteins), and Ward Chebyshev (27 primary tumor – non-malignant: 1984 proteins, 27 lymph node – non-malignant: 2137 proteins, 24 buffy coat: 2188 proteins; 24 saliva samples: 1154 proteins). **b** Association between lymph node metastasis and patient clustering for non-malignant cells from the lymph node (*p* = 0.046; two-sided Fisher's exact test). *$p \leq 0.05$. **c** PCA plots presenting clusters of samples based on the abundance of proteins that were differentially abundant between pN+ and pN0 (*p* ≤ 0.05; two-sided unpaired Student's *t*-test or proteins detected exclusively in one group; 201, 110, 85, 80, 54 proteins from non-malignant cells from lymph nodes, malignant cells from primary tumor, non-malignant cells from primary tumor, buffy coat samples, and saliva samples, respectively).

**d** Volcano plot for the differential protein abundance between pN+ and pN0 non-malignant cells from lymph nodes. Differentially abundant proteins are presented as blue and orange dots (pN+ vs pN0; *q* ≤ 0.05; two-sided unpaired Student's *t*-test followed by Benjamini-Hochberg test or proteins detected exclusively in one group in at least 50% of samples). **e** Top-10 GO biological processes that were significantly enriched for the 13 proteins associated with lymph node metastasis from **d** (Enrichment *FDR* ≤ 0.05; hypergeometric test followed by FDR correction). **f** AUC distribution per classifier using ML analysis of SRSF1, SRSF2, SRSF3, SRSF5, TRA2A, and CD209 proteins (upper panel) and transcripts (lower panel). Details about the AUCs plotted for each classifier are available within Supplementary Data 5-2 and 5-3. Boxplots show the median (central line), the 25–75% interquartile range (IQR) (box limits), and the ±1.5×IQR (whiskers). Source data are provided as a Source Data file.

(enrichment *FDR* = 1.2E-07 to 2.0E-06) (Fig. 4e). *FDR* correction was also applied to define proteins associated with locoregional metastasis in the other four datasets; however, no significant results were observed. We then performed an additional analysis to refine the 13 targets. Downregulation of this set of proteins was associated with other poor prognostic features (*p* ≤ 0.05; two-sided unpaired Student's *t*-test, ANOVA, and two-sided Fisher's exact test) (Supplementary Fig. 6a), and they were demonstrated to be physically or functionally associated according to protein-protein interaction networks (Supplementary Fig. 6b).

Finally, using an ML model, we predicted the power of the five splicing proteins to distinguish pN+ and pN0 patients using data from the discovery phase. We also included the protein CD209 in ML analysis due to the involvement of immune processes in HNSCC that are described at several sites (please see Methods section 'Selection of targets for ML analysis') (Supplementary Data 5-1). Distinct pairs <Signature Si, Classifier Cj> could discriminate pN+ and pN0 HNSCC patients with elevated AUCs (mean AUC = 0.933), thus indicating the high performance of the six proteins in distinguishing patients according to pN (Fig. 4f; Supplementary Data 5-2). Similarly, transcripts of the six genes also exhibited lower expression in pN+ compared to that in pN0 (RT-qPCR; *p* = 1.14E-04 to 2.45E-02; two-sided unpaired Student's *t*-test), and this corroborates the proteomic pattern (Supplementary Fig. 6c, d). Additionally, transcript pairs <Cj, Si> could discriminate pN+ and pN0 HNSCC patients with high AUCs (mean AUC = 0.853) (Fig. 4f; Supplementary Data 5-3). In summary, we demonstrated the importance of the six metastasis markers in HNSCC clinics and then investigated their relationship with the immune compartment.

## Microenvironment and other splicing markers may be expressed by immune populations

Based on our data and the knowledge that the lymph node metastasis proteome was deeply associated with the immune response (Fig. 2b, d–f; Supplementary Fig. 3f), we next investigated if the six selected targets (SRSF1, SRSF2, SRSF3, SRSF5, TRA2A, and CD209) from non-malignant cells from lymph nodes are likely to be expressed by the immune cell types from the lymph node microenvironments. We also evaluated the gene expression in non-immune populations. From the publicly available scRNASeq data analysis, we observed that *SRSF2* and *TRA2A* were cluster markers of immune cells from lymph nodes (Fig. 2e); additionally, we did determine that all the six targets are expressed by the immune milieu from pN+ lymph nodes in an immune cell-type-specific manner (Fig. 5a). Globally, *SRSF1* and *SRSF2* genes were highly expressed in CD8+ T effector cells from lymph node pN+ microenvironments (non-malignant cells) when compared to other cell populations, while the splicing genes *SRSF3, SRSF5,* and *TRA2A* had elevated transcript levels in regulatory T cells (Tregs) (Fig. 5a). *CD209* exhibited the most elevated transcript levels in macrophages (Fig. 5a). We also analyzed the differential expression of the six targets in pN+ *versus* pN0 single cell environments. Once the pN0 single cell

counterpart of lymph node non-malignant cells is not available in the literature, we used primary tumor information (Supplementary Fig. 3g)[5]. *SRSF5* was downregulated in macrophages and Treg cells from pN+ HNSCC in relation to pN0 samples, and this sense of alteration is in accordance with the results obtained in proteomics (Fig. 5b) (adjusted *p* ≤ 0.05; two-sided Wilcoxon test and Benjamini-Hochberg correction). Particularly, *TRA2A* was downregulated in the non-immune population of CAFs (pN+ *vs.* pN0; *p* ≤ 0.05; two-sided Wilcoxon test and Benjamini-Hochberg correction) (Supplementary Fig. 3h).

We then provided a deeper understanding of the modulation of alternative splicing in the immune cells by evaluating the differential expression patterns associated with metastasis in a larger set of splicing genes. The gene expression of a list of 421 gene ontology (GO) splicing factors (https://www.uniprot.org) was verified in immune cells based on scRNASeq data from non-malignant cells of HNSCC primary tumors[5], as performed for the splicing genes *SRSF1*, *SRSF2*, *SRSF3*, *SRSF5*, and *TRA2A*. From the 421, 41 GO splicing genes were significantly down- or upregulated in immune populations from pN+ when compared to pN0 (adjusted *p* ≤ 0.05; two-sided Wilcoxon test and Benjamini-Hochberg correction) (Fig. 5c; Supplementary Data 6-1). As observed for the microenvironment markers, most of the 41 GO splicing genes were downregulated in pN+ cells (28 downregulated and 13 upregulated genes) and this is a significant and prevalent sense of alteration for CD8+ T effector, CD8+ T exhausted, dendritic cells, macrophages, Treg, and plasma cell populations (Fig. 5d) (*p* = 0.021; two-sided Fisher's exact test).

In parallel, we searched the 421 splicing proteins in our bulk proteome datasets from the multisites to have a view of splicing across HNSCC non-malignant cells and fluids. Even though we just considered sites that contain immune cells, we unfortunately could not separate the splicing modulation at single-cell level because these are bulk data. The datasets contained information of mixed subpopulations and included metastasis-associated proteomes of non-malignant cells isolated from primary tumor (*n* = 85 proteins differentially abundant in pN+ vs. pN0) or lymph nodes (*n* = 201 proteins) and fluid cells (*n* = 80 proteins for buffy coat and *n* = 54 proteins for saliva), as presented in Supplementary Data 2. A group of 21 GO splicing proteins was associated with the metastatic phenotype (pN+ vs. pN0) across the multisites, mostly in non-malignant cells from lymph nodes (Fig. 5e; Supplementary Data 6-2). From these, downregulated splicing factors prevailed in the pN+ phenotypes of non-malignant cells from lymph nodes and buffy coat samples, which have the largest proportions of immune cells, demonstrating again that the downregulation of splicing factors in immune cells is associated with the metastatic phenotype in HNSCC (*p* = 0.0008; two-sided Fisher's exact test) (Fig. 5f).

Altogether, we had 61 non-redundant splicing factors differentially expressed between pN+ and pN0 when considering single cell RNA and proteomic data, and these are mainly molecules downregulated in the pN+ phenotype. The 61 splicing factors can target 41 unique genes according to the SpliceAid-F database[35]. Interestingly,

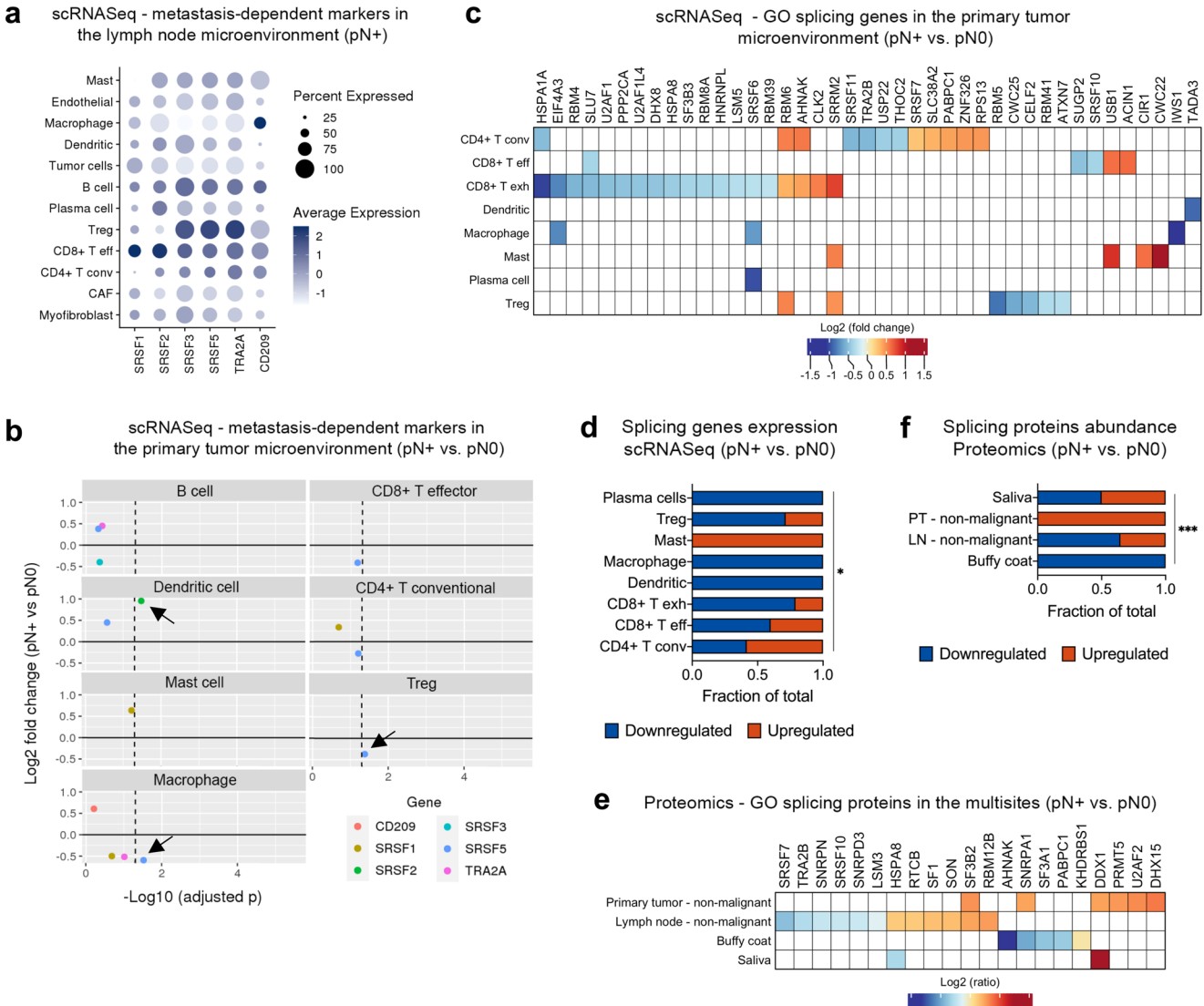

**Fig. 5 | Molecular patterns of splicing factors in immune cell-type-specific gene expression or multisite proteomes and their relationship with metastasis.**
**a** Expression of the six selected targets (SRSF1, SRSF2, SRSF3, SRSF5, TRA2A, and CD209) in immune populations from HNSCC that were identified by scRNASeq public data[5] in lymph node pN+ microenvironment. **b** Average fold change (y axis) and adjusted $p$ (x axis) between pN+ and pN0 tumor microenvironments[5] for the six targets evaluated in individual immune populations (two-sided Wilcoxon test followed by Benjamini-Hochberg correction). The dashed lines represent a $p$ threshold of 0.05 and significant results are indicated by arrows. The differences could not be tested for plasma and CD8+ T exhausted cells because there were not enough cells for comparison. **c** Heat-map representing the average fold change for GO splicing genes differentially expressed between pN+ and pN0 immune populations from tumor microenvironments[5] (pN+ vs. pN0; adjusted $p \leq 0.05$; two-sided Wilcoxon test followed by Benjamini-Hochberg correction). Blank cells indicate

transcripts without difference between pN+ and pN0. **d** Proportion of up- and downregulated genes from (**c**) that are expressed by the immune cells. Gene expression (up- and downregulation in pN+ when compared to pN0) was significantly associated with the immune cell types ($p = 0.0210$; two-sided Fisher's exact test). *$p \leq 0.05$. **e** Heat-map representing the average ratio for GO splicing proteins differentially expressed between pN+ and pN0 multisites (pN+ vs. pN0; $p \leq 0.05$; two-sided unpaired Student's $t$-test). The log2 LFQ ratios are shown for the statistically significant proteins. Blank cells indicate proteins that were not differentially abundant between pN+ and pN0. **f** Proportion of up- and downregulated proteins from (**e**) expressed in the multisites. Protein abundance (up- and downregulation in pN+ when compared to pN0) was significantly associated with the environments ($p = 0.0008$; two-sided Fisher's exact test). LN: lymph node; PT: primary tumor. ***$p \leq 0.001$. Source data are provided as a Source Data file.

two of these targets (HBB and TRA2B) were associated with metastasis in the proteome of non-malignant cells isolated from lymph nodes with downregulation in the pN+ phenotype (Supplementary Data 6-3). All these data reinforced the modulation of splicing in immune cells, mainly through the downregulation of these factors, which may significantly impact metastasis in HNSCC.

**Metastasis-associated tissue targets can be detected in liquid biopsies**
Once the six selected metastasis-dependent markers (SRSF1, SRSF2, SRSF3, SRSF5, TRA2A, and CD209) were all associated with clinics and

presented a value in the immune response and/or composition, they were evaluated collectively with the nine immune markers that were previously prioritized (CD3, CD4, CD8, CD11b, CD14, CD16, CD19, CD45, and CD66b) for their ability to discriminate pN+ and pN0 patients in liquid biopsies using targeted methodologies followed by an ML model (please see Methods section 'Selection of targets for ML analysis'). Liquid biopsies exhibit great potential for cancer management, as they all for the assessment of markers in a minimally invasive manner[36]. Moreover, lymph node cells can broadly circulate throughout the body[37], and nodal markers can be identified in body fluids as a more convenient source of biomarkers. We used selected reaction

monitoring-mass spectrometry (SRM-MS) to measure the relative abundance of selected peptides and proteins in 19 buffy coat (7 pN+, 12 pN0) and 25 saliva samples (16 pN+, 9 pN0) from a 36-HNSCC patient cohort. Quality control revealed an average carryover of iRT and heavy-isotope-labeled peptides of 0.16% (0.1%–0.45%) between SRM-MS assays, and this is similar to the 0.1% reported in the literature[38] (Supplementary Fig. 4c). Quality parameters (Methods section 'Quality control in SRM-MS and PRM-MS') were satisfactory for all evaluated samples (Supplementary Fig. 4d–g). Selected peptides from the 15 immune and microenvironment candidate markers were detected in the buffy coat HNSCC cohort, whereas for saliva samples only the peptides SRSF3_Pep1, SRSF5_Pep1, SRSF5_Pep2, CD45_Pep1, and CD4_Pep1 could not be confidently measured due to low signal-to-noise ratios (Supplementary Data 4-2–4-6; Supplementary Data 7-2 – 7-4). We also evaluated the gene expression of the six micro-environment targets in a 24-patient cohort from buffy coat samples (10 pN+, 14 pN0) and a 22-patient cohort from saliva cells (10 pN+, 12 pN0) using RT-qPCR (Supplementary Data 7-1). All targets were identified in both fluids (Supplementary Data 4-5 and 4-6). Thus, we demonstrated that blood and saliva contain the metastasis markers that were previously detected for lymph nodes, and we next employed a multi-parametric ML approach.

## Machine learning predicts liquid biopsy-metastasis signatures with high performance

Finally, we used ML to evaluate the power of candidate prognostic markers in predicting nodal status in liquid biopsies (Fig. 6a). Twelve individual (peptide, protein, or transcript) or combined (peptides + proteins + transcripts) datasets from the 15 immune and micro-environment markers (CD3, CD8, CD4, CD11b, CD14, CD16, CD19, CD45, CD66b, SRSF1, SRSF2, SRSF3, SRSF5, TRA2A, and CD209) assessed in saliva and buffy coat by SRM-MS or RT-qPCR were tested in ML analysis (Supplementary Data 4-2–4-6; Supplementary Data 5-1; Supplementary Fig. 7a).

A strategy was defined to select high-performance signatures that could stratify pN+ and pN0 patients among the pairs signature-classifier <Si, Cj> defined by ML (for details, please see Methods section 'Definition of prognostic signatures using machine learning') (Supplementary Fig. 7b). Due to the sample size limitation for this ML analysis, we did not intend to select one model, but to report a set with many candidates' high-performance signatures that could be used in the decision-making process regarding future studies with larger samples sizes. In summary, we first selected the pair <Si, Cj> with the highest ROC AUC (named top-1) and equivalent pairs that statistically did not discriminate from the best one (two-sided unpaired Student's t-test, $p \geq 0.05$). Then, pairs with permutation $p \leq 0.05$ and AUC $\geq 0.85$ were prioritized for every dataset (Fig. 6b; Supplementary Fig. 7c; Supplementary Data 5-4). An overrepresentation of the best pairs <Si, Cj> was yielded when combining peptides, proteins, and transcripts levels for the 15 targets evaluated in both buffy coat and saliva (Fig. 6b; Supplementary Data 5-4). Thus, subsequent analysis considered the 112 and 1,727 signature pairs <Si, Cj> that were generated by ML for the combined datasets of buffy coat and saliva, respectively, and could stratify pN+ and pN0 patients with elevated AUCs (Fig. 6b; Supplementary Data 5-4). The individual markers SRSF5_Pep1, SRSF3_RNA, and TRA2A_RNA were the most frequent targets present in the buffy coat high-performance pairs <Si, Cj >, identified in around 15% of these signatures, while SRSF1_Pep1, TRA2A_Protein and TRA2A_Pep1 were highly identified among the high-performance signatures from saliva, and occurred in 10% to 15% of the pairs <Si, Cj > (Fig. 6c).

Among the high-performance signatures generated for the combined datasets, the most elevated AUCs (top-1 signature) included a four-target signature for buffy coat (CD11b_Protein, CD11b_Pep3, SRSF3_Pep1, and TRA2A_Pep2; linear discriminant; AUC = 0.953; 95% CI = 0.91 – 1.00; sensitivity = 0.933; specificity = 0.907) and a four-

target signature defined for saliva (CD16_Pep1, SRSF1_Pep1, TRA2A, SRSF1 transcript; RBF SVM; AUC = 0.919; 95% CI = 0.86 – 0.92; sensitivity = 0.834; specificity = 0.936) (Fig. 6d). We then selected the top-1 signature for buffy coat to indicate the target circulating immune populations responsible for this signature (SRSF3 and TRA2A proteins). Flow cytometry was performed in blood samples from an independent 20-HNSCC patient cohort (10 N+; 10 N0) (Supplementary Fig. 7d, e). Both proteins were highly expressed in an elevated proportion of buffy coat's myeloid immune populations (Fig. 6e, f; Supplementary Data 8). N+ samples had a lower proportion of myeloid cells expressing SRSF3 when compared to N0 ($p = 0.0004$; two-sided Mann-Whitney test) (Fig. 6e), as well as downregulation or a reduced proportion of myeloid cells ($p = 0.050$; two-sided unpaired Student's t-test), neutrophils ($p = 0.046$; two-sided unpaired Student's t-test and $p = 0.050$; two-sided unpaired Student's t-test), and lymphocytes ($p = 0.030$; two-sided unpaired Student's t-test) expressing TRA2A (Fig. 6f). Notably, the reduced levels of SRSF3 and TRA2A in the circulating cells of N+ patients reflect the lower abundance of these proteins found in the pN+ microenvironment of lymph nodes (Fig. 4d; Supplementary Fig. 6c, d; Supplementary Data 2-7). Moreover, the relevance of neutrophils detected for TRA2A is in line with the enrichment of neutrophils-dependent biological processes identified for metastasis markers in buffy coat (Fig. 2b), strengthening the significance of this immune population in the metastatic cascade.

Taken together, the results of this study depicted the framework of the wired microenvironments in HNSCC and provide a promising basis for understanding tumor biology, indicating a potential signature of metastasis for this disease.

## Discussion

Cancer is a systemic disease, and the crucial contribution of multiple environments to the regulation of HNSCC implicates a fundamental role of varied populations in supporting a tumoral niche. Hence, further progress in head and neck oncology will require a global understanding of the diverse molecular landscape of the neoplasm. Additionally, molecular profiling of isolated cell populations from tissues reduces the intrinsic heterogeneity caused by the mixing of cell types and is essential for deciphering cancer. Therefore, we performed a comprehensive mass spectrometry-based proteomic analysis of isolated cell populations from tissues and fluids to characterize HNSCC.

Our key findings are in regard to the modulation of the immune system retrieved from the global and metastasis-dependent HNSCC proteomes (Figs. 1d, e, f; 2b, d, e). The tumor immune infiltrating composition has been previously explored in HNSCC[39–41]; however, an advance in our study is the identification of the overrepresentation of immune processes in the multisite samples evaluated. The immune response is coordinated across tissues, and its relationship to cancer must encompass the peripheral immune system in addition to the TME[27]. The association between the immune population and global proteomes identified here is of special interest, as the immune context exerts profound effects on the response to immunotherapies. Resistance to immunotherapy remains a bottleneck in regard to the successful treatment of cancer, and greater than 80% of HNSCC patients with metastatic disease do not respond to PD-1 blockade[42]. Hence, biomarkers for patient stratification or the identification of new immune targets are urgently needed. The multisite proteomic analysis presented in this study elucidated the immune cell composition of HNSCC TME and blood using CIBERSORTx version 1.0 (Fig. 1e, f), and this is of paramount relevance to select targeted therapies against the cell types individually in a clinical setting[43]. Even though the use of an RNA signature to estimate cell types based on protein abundance in CIBERSORTx version 1.0 might not be as accurate as using RNA data, external validations showed that cell type estimation using transcript matrix for proteomics data may not skew the prediction

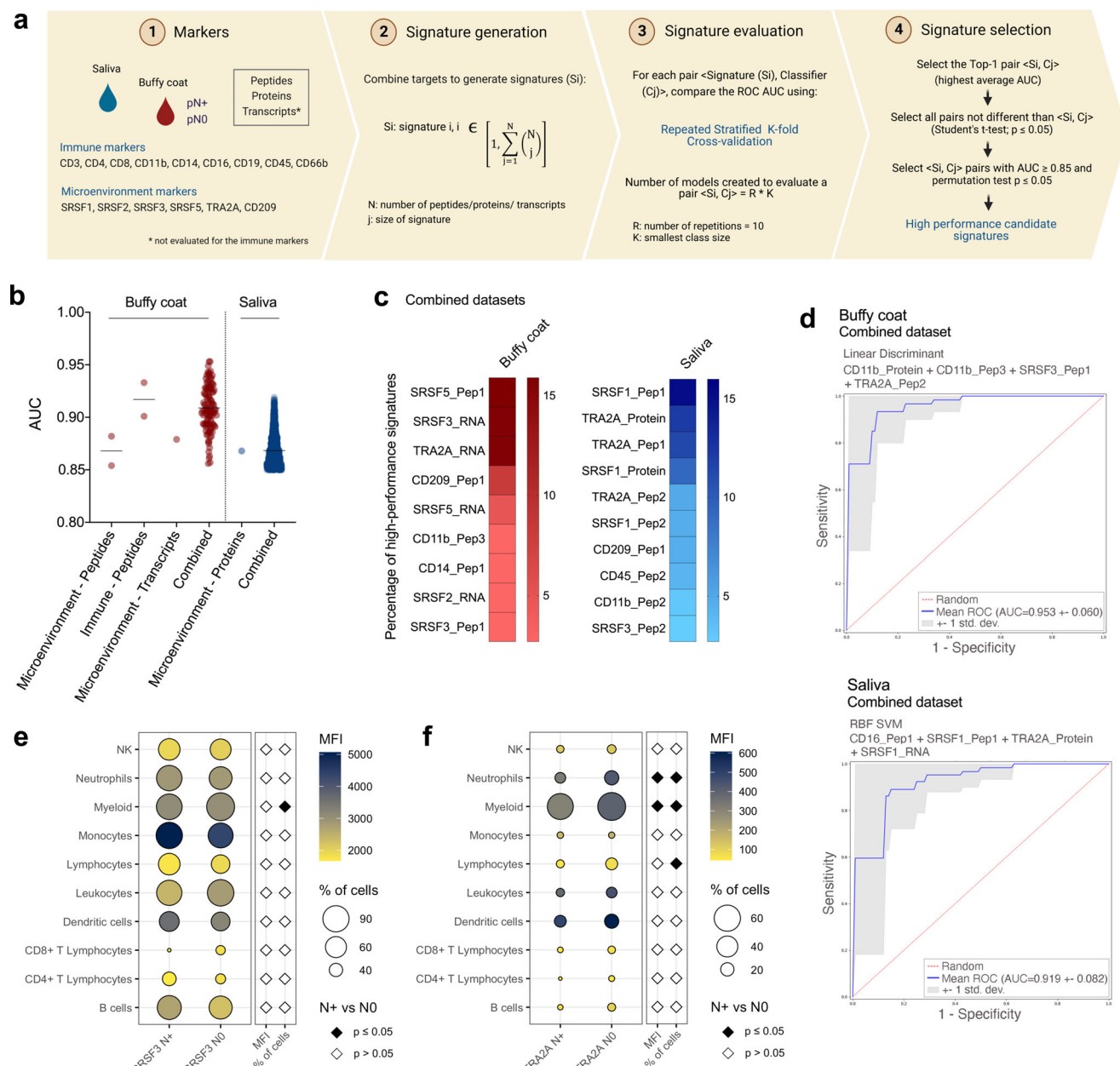

**Fig. 6 | Definition of the best prognostic signatures in liquid biopsies according to machine learning. a** Design used to determine the signatures of lymph node metastasis with the best performance according to ML. Created with BioRender.com. **b** High-performance signatures determined for buffy coat and saliva individual and combined datasets after filtering for ROC AUC ≥ 0.85 and permutation $p \le 0.05$. The solid line represents the means for each dataset. **c** Percentage of markers among the buffy coat (left panel) and saliva (right panel) signatures that exhibited high performance in regard to discriminating pN+ and pN0 patients. Combined datasets containing peptide, protein, and transcript data for the 15 markers were considered. **d** ROC curves indicating the top-1 pairs <Si, Cj> in buffy coat (upper panel) and saliva (lower panel) samples that discriminated HNSCC patients based on lymph node metastasis status (AUC > 0.919). Shaded region indicates 95% confidence interval for the AUC. Combined datasets containing peptide, protein, and transcript data for the 15 markers were considered. Mean fluorescence intensity (MFI) and percentage of immune cells expressing the proteins SRSF3 (**e**) and TRA2A (**f**) in 10 N+ and 10 N0 buffy coat samples. Significant differences between groups are represented as black diamonds (two-sided unpaired Student's $t$-test or two-sided Mann-Whitney test; $p \le 0.05$). AUC area under the curve. Source data are provided as a Source Data file.

(Supplementary Fig. 3d, e), and it can be useful for future proteomics studies. Besides defining the cell composition, our study revealed three categories of biological processes that are highly enriched for clusters of proteins, and these processes include neutrophil-mediated immunity, antigen processing and presentation (APP), and the regulation of humoral response mainly through the Fc receptor signaling pathway (Fig. 1d). It was just recently established that neutrophils have an important contribution in the initiation, development, and progression of cancer, and targeting this populations clinically is an emerging area of interest[44]. Alterations in the APP are known to be associated with immune evasion and can result in impaired antitumor responses and therapy resistance[45]. Additionally, the blockade of the interaction between the inhibitory Fc receptor FcγRIIB and immune complexes in mouse models has been proposed as an approach for cancer immunotherapy[46]. Thus, we identified immune cell players and components of specific pathways that can be further targeted to enhance the immunogenicity in HNSCC or that can be explored as markers of immunotherapy resistance.

Understanding the molecular principles underlying cancer invasion and metastasis is a highly complex endeavor and is of special

interest in HNSCC due to its association with poor prognosis[2]. Our work revealed that pN status is related to immunity in HNSCC by using a combination of our own proteome data, public RNASeq information, and cell counts obtained from clinical laboratories (Figs. 2b, d–f; 5; Supplementary Fig. 3f, h). We deciphered immune GO biological processes associated with metastasis markers that are commonly enriched in multiple sites, thus indicating that the development of lymph node metastasis depends upon an orchestrated interconnection of immune system across multiple HNSCC environments (Fig. 2b, d). This interconnection may be necessary to maintain an immunosuppressive systemic environment in HNSCC that is a prerequisite for preparing the pre-metastatic niche and thus allowing for the establishment and maintenance of cancerous cells in the pN+ lymph nodes[7]. Indeed, a suppressed pN+ environment may be supported by splicing proteins that are strongly associated with the metastatic phenotype in our HNSCC dataset, and the genes associated with these proteins include *SRSF1, SRSF2, SRSF3, SRSF5,* and *TRA2A*, and also the immune marker *CD209* (Fig. 4d). *CD209* exhibits high transcript levels in pN+ macrophages (Fig. 5a) that can potentially prevent CD8+ T cells from exerting their full cytotoxicity against tumor cells. Remarkably, a group of the splicing genes are highly expressed in the immunosuppressive Treg cell population and this expression is associated with the presence of lymph node metastasis in HNSCC based on scRNASeq data (Fig. 5a, b; Supplementary Fig. 3g). The Treg population exerts a potent immunosuppressive function that plays a crucial role in tumor immune escape[47], suggesting the involvement of these splicing markers in the suppressive environment associated with the metastatic phenotype. We further extended our analysis and showed a group of 61 splicing factors that is related to the metastatic phenotype in an immune cell-type- or multisite-specific way (Fig. 5c–f). Our results in a global manner suggest that downregulating splicing in immune cells can contribute to tumor progression and a poor patient prognosis; thus, targeting alternative splicing pathways might be a promising strategy for therapy. As an example, the alternative splicing in the immune system is known to generate a diverse repertoire of antigen receptors, such as those expressed by B and T cells[48], changing the endogenous antigenicity and then also have clinical potential for using such splicing-derived epitopes as checkpoint immunotherapy. Still other studies must be developed to better understand the splicing effect in RNA maturation and proteoforms, as well as functional changes.

Additionally, lymphatic metastasis is the primary route for the development of distant metastasis in a subset of tumors[49,50], and blocking the spread of tumor cells via lymphatic vessels may prolong the life of cancer patients and also mitigate the poor prognosis. By comparing malignant and non-malignant cells between primary sites and lymph nodes (Fig. 3a), we identified several proteins involved in the metastatic spread, including EMT-, motility- and translation-associated molecules (Fig. 3d-i). It was shown that the modulation of specific biomarkers can inhibit the metastatic colonization[51,52], and the proteins associated with metastasis herein shown are potential targets to be further evaluated in animal models and considered to clinically prevent the spread of HNSCC. Indeed, the hierarchical clustering analysis revealed that non-malignant cells possess a more homogeneous proteomic profile across patients than do malignant cells because the samples are grouped according to site instead of by patient (Fig. 3c). The same pattern was observed at the scRNA level in HNSCC[5], thus indicating that targeting the microenvironment cells may yield improved treatment responses due to its reduced dynamism across patients. Interestingly, the non-malignant proteomes from tumors and lymph nodes showed a differential profile for translation-associated proteins that reflects the heterogeneous cellular composition between sites. This constitution may be associated to the intrinsic function of lymph nodes in the immune response that harbors the transition of naive to effector T lymphocytes and is characterized by the high abundance of the translational machinery and enhanced protein synthesis[53,54]. As previously reported, the higher translational levels may also be related to the significant enrichment of B cells, T cells, macrophages, neutrophils, plasma cells, myofibroblasts, and specific CAF populations in breast cancer and HNSCC pN+ lymph nodes compared to that at primary sites[5,55], or these levels may even be associated with the recruitment of neutrophils, monocytes, and macrophages that are necessary for the preparation of pre-metastatic niches in pN0 sites of metastasis[56–58].

Although a long list of prognostic biomarker candidates can be observed in the literature, no molecular marker has been widely accepted for routine use in managing patients with HNSCC. Currently, only HPV and p16[INK4A] are considered to be prognostic markers for oropharyngeal cancer, and both are associated with an improved prognosis[59]. Hence, it appears that a multiparametric assessment will be necessary to dissect the complex tumor interactions and define prognosis in HNSCC, and this assessment will primarily include lymph node metastasis-dependent signatures that represent the primary poor prognosis feature. In this scenario, the multisite analysis led us to develop a multiparametric ML model that allowed for the indication of high-performance signatures of peptides, proteins, and/or transcripts in saliva or blood that can discriminate pN+ and pN0 HNSCC (Fig. 6; Supplementary Fig. 7). We successfully combined multiple datasets for ML analysis (peptides + proteins + transcripts for immune and microenvironment markers), and the high-performance signatures for buffy coat and saliva samples clearly outperformed signatures indicated by individual datasets in regard to separating patients according to pN status. Interestingly, evaluating the expression of the top-1 signature defined for buffy coat (SRSF3 and TRA2A proteins from the signature 'CD11b_Protein, CD11b_Pep3, SRSF3_Pep1, and TRA2A_Pep2'; linear discriminant; AUC = 0.953) (Fig. 6d–f) strengthened the paramountcy of the myeloid lineage, especially neutrophils, in the metastatic phenotype of HNSCC (Fig. 2b). Nevertheless, herein we show a disease snapshot and whether the modulation of neutrophils is a cause or effect of HNSCC dissemination needs further investigation. Overall, the ML analysis served to understand our datasets and the complexities and noises that differentiate the type of data sources evaluated, revealing promising metastasis-dependent signatures that can noninvasively guide the decision-making process for HNSCC patients. Indeed, substantial work is still required before a possible clinical application of these signatures, and this include the evaluation in larger independent cohorts together with assay optimizations.

In summary, our results explored the molecular mechanisms underlying HNSCC carcinogenesis through a multisite quantitative mass spectrometry-based proteomics analysis that identified potential cell players and biological processes related to immune system that can be further targeted for therapy or explored as prognosis markers. High-performance locoregional metastasis-dependent signatures have been depicted and are promising for future clinical implementation. We investigated the HNSCC multisite landscape providing the clinical and research communities with valuable information that may guide the management of patients.

## Methods
### Patients and sampling
A 93-patient cohort with HNSCC obtained from the oral cavity ($n = 81$ patients), larynx ($n = 11$ patients), and oropharynx ($n = 1$ patient) was included in this work. The study was compliant with the ethical standards and informed written consent was obtained from all individuals as approved by the Ethics Committees of Carlos Van Buren Hospital (Process 121), University of Valparaíso (Process CB051-14), University of São Paulo Academic Biobank of Research on Cancer, Centro de Investigação Translacional em Oncologia, Instituto do Câncer do Estado de São Paulo (ICESP) (Protocol CAEE 30658014.1.1001.0065), Faculty of Medicine of Jundiai (Protocol CAEE 45091715.1.0000.5412), and A.C. Camargo Hospital (Protocol 2532/18B). In total, we included in

this study 184 samples collected from the 93 HNSCC-patient cohort that were evaluated in a discovery and/or verification phase. The discovery cohort consisted of 59 HNSCC patients from whom 27 FFPE primary tumors, 27 FFPE lymph nodes, 24 blood, and 24 saliva samples were collected and used in discovery proteomics (DDA). The verification cohort comprised an 83-patient group, and 23 FFPE tissues, 47 blood, and 33 saliva samples were used in the PRM-MS, SRM-MS, RT-qPCR, and flow cytometry experiments. Of the 83 patients used in the verification, 34 were independent cohorts that were not evaluated in discovery proteomics. Details of the HNSCC cohorts and methodologies used in this work are summarized in Supplementary Data 1-1. For clarity, we also presented the sample sizes used in each methodology below or throughout the text.

Saliva was collected as previously described[4], and collection occurred preferably in the morning from individuals who had not eaten or ingested liquids (except water) and had undergone oral hygiene at least 1 h prior to collection. Patients were instructed to rinse their mouths with 5 mL of drinking water, and saliva was subsequently harvested without stimulation into a plastic receptacle. Four mL of peripheral blood were collected in BD Vacutainer® tubes with EDTA as an anticoagulant (BD Life Sciences, USA). The cases were histopathologically classified according to the recommendations of the World Health Organization[60], the American Joint Committee on Cancer (AJCC), and the International Union Against Cancer (IUAC)[61]. None of the patients received neoadjuvant radiotherapy or chemotherapy prior to sample collection. The clinical and pathological data are summarized in Supplementary Data 1-2.

### Isolation of cells from tissues

Malignant and non-malignant cells were harvested from FFPE primary tumor and lymph node samples using micro- or macrodissection. Histology-guided laser microdissection was used to recover malignant and/or adjacent non-malignant portions from FFPE primary tumors ($n = 27$ samples) and metastatic lymph nodes ($n = 13$ samples). Malignant cells from tumors were retrieved from the invasive front, an area with high prognostic potential where the most invasive and aggressive cells reside[62]. The invasive front contains the most advanced tumor cells that invade normal tissues such as muscle, connective tissue, salivary glands, and blood vessels[62]. The tumor invasive front was collected from the farthest area of the invasive surface of the tumor, and it was collected up to a depth of one mm in the histological section[4]. Hematoxylin and eosin-stained histological slides were cut into 5 μm sections to guide microdissection. Slices (10 μm) were mounted on Arcturus PEN Membrane Glass Slides (Life Technologies, USA), deparaffinized with xylene, hydrated in a graded ethanol sequence, and stained with hematoxylin for 1 min. An area of approximately 3,000,000 μm² was dissected from one to three slides per sample for (i) malignant cells from primary tumors, (ii) non-malignant cells adjacent to primary tumors (mucosal margins), (iii) malignant cells from lymph nodes, and (iv) non-malignant cells from metastatic lymph nodes adjacent to the metastasis using Leica LMD6 equipment (Leica Biosystems, Germany). Samples were collected in 600 μL tubes. Non-malignant cells from non-metastatic lymph nodes ($n = 14$ samples) were recovered from 10 μm slices mounted on standard slides, and they were used as the unique site for macrodissection due to the population homogeneity of the tissue. A 3,000,000 μm² area was selected according to a comparison to parallel hematoxylin and eosin stained 5 μm sections, scraped into 600 μL tubes, deparaffinized with xylene, and hydrated in a graded ethanol sequence. All dissected samples were stored at −80 °C until MS analysis.

### Isolation of cells from fluids

The buffy coat and saliva fractions were obtained using centrifugation. Saliva samples ($n = 38$ samples) from HNSCC patients were centrifuged

for 5 min at 1500 × $g$ and 4 °C to separate the cells. Peripheral blood samples ($n = 52$) were centrifuged for 8 min at 1000 × $g$ at room temperature, and the cellular portion (buffy coat) was washed with lysis buffer to eliminate red cells (10 mM Tris-HCl pH 7,6, 5 mM MgCl₂, 10 mM NaCl). For flow cytometry analysis, buffy coat samples were fixed with 4% paraformaldehyde prior to red cell lysis. The cellular portions from the fluids were frozen at −80 °C for further analysis. The buffy coat is primarily composed of leukocytes and granulocytes[63], while saliva samples possess high levels of epithelial cells, leukocytes, and microorganisms[64].

### Sample preparation for discovery proteomics

Malignant cells from tumors ($n = 27$ samples), non-malignant cells from tumors ($n = 27$ samples), malignant cells from lymph nodes ($n = 13$ samples), non-malignant cells from lymph nodes ($n = 27$ samples), buffy coat samples ($n = 24$ samples), and saliva cells ($n = 24$ samples), totalizing 94 tissues and 48 fluid samples, were submitted to discovery proteomics. Two tissue samples were excluded based on quality control analysis and the results for 92 out of the 94 tissue runs are presented in the manuscript. Proteins were isolated from buffy coat samples and saliva cells using TRIzol reagent (Invitrogen, USA) and resuspended in 200 μL of urea buffer (100 mM Tris-HCl pH 7.5, 8 M urea, and 2 M thiourea). Ultrasound treatment was performed for 10 min in an iced water bath for homogenization. Micro- or macrodissected tissue samples were transferred to an 8 M urea solution. All tissue and fluid samples were treated with 5 mM dithiothreitol for 25 min at 56 °C and 14 mM iodoacetamide for 30 min at room temperature for cysteine reduction and alkylation, respectively. Urea was diluted to a final concentration of 1.6 M with 50 mM ammonium bicarbonate, and 1 mM calcium chloride was added to the samples. For protein digestion, a total of 2.5 μg of trypsin (Promega, USA) was added in three steps that included 1 μg for 20 h, 1 μg for an additional 20 h, and 0.5 μg for the last 4 h at 37 °C. The reactions were quenched with 0.4% formic acid, and the peptides were desalted with C18 stage tips (3 M, USA)[65] and dried in a speed-vac instrument. Tissue samples were reconstituted in 0.1% formic acid that was applied proportionally to the dissected area (10 μL of formic acid for 1,000,000 μm²), and iRT peptides were added to the digested tissue sample at a final concentration of 11.1 fmol/μL for LC-MS/MS quality control (Pierce™ Peptide Retention Time Calibration Mixture, Thermo Scientific, USA). A volume of 4.5 μL was used (50 fmol of iRT peptides). Peptides from the buffy coat and saliva samples were quantified using the Pierce™ Quantitative Colorimetric Peptide Assay (Thermo Scientific, USA), and 2 μg of these proteins were subjected to LC-MS/MS analysis.

### Discovery proteomics and data analysis

Tissue ($n = 92$ samples after excluding two samples based on quality control) and fluid samples ($n = 48$ samples) were analyzed by LC-MS/MS using an ETD-enabled Orbitrap Velos mass spectrometer (Thermo Fisher Scientific, USA) connected to an EASY-nLC system (Proxeon Biosystem, USA) through a Proxeon nanoelectrospray ion source. The MS data was acquired using the Xcalibur software version 2.1 (Thermo Fisher Scientific, USA). To prevent bias during DDA measurements, samples were randomized in the R version 3.6.2 (Supplementary Data 1-3). Peptides were subsequently separated in a 2–90% acetonitrile gradient in 0.1% formic acid using a PicoFrit analytical column (20 cm × ID75, 5 μm particle size, New Objective) at a flow rate of 300 nL/min over a 212 min gradient (35% acetonitrile at 175 min) for tissue and buffy coat samples or a 170 min gradient (35% acetonitrile at 123 min) for saliva samples. The nanoelectrospray voltage was set to 2.2 kV, and the source temperature was 275 °C. All instrument methods were configured for data-dependent acquisition (DDA) in the positive ion mode. Full scan MS spectra ($m/z$ 300–1600) were acquired in the Orbitrap analyzer after accumulation to a target value of 1e6 ions.

Resolution in the Orbitrap was set to r = 60,000, and the 20 most intense peptide ions with charge states ≥ 2 were sequentially isolated with an isolation window of $m/z$ 3 to a target value of 5000 and then fragmented in the linear ion trap by low-energy CID (normalized collision energy of 35%). The signal threshold for triggering an MS/MS event was set at 1000 counts. Dynamic exclusion was enabled with an exclusion size list of 500, an exclusion duration of 60 s, and a repeat count of 1. An activation of $q$ = 0.25 and an activation time of 10 ms were used. Proteins were identified using MaxQuant version 1.5.8.0[66,67] against the Uniprot Human Protein Database (92,646 protein sequences, 36,874,315 residues, release May 2017) using the Andromeda search engine. Carbamidomethylation was set as fixed modification, and N-terminal acetylation and oxidation of methionine were used as variable modifications. Maximum 2 Trypsin/P missed cleavage, a tolerance of 4.5 ppm for precursor mass, and a tolerance of 0.5 Da for fragment ions were set for peptide identification. Protein groups (also referred to as proteins in the text) were automatically inferred by the Andromeda engine using the parsimony principle. A maximum of 1% FDR calculated using reverse sequences was set for both protein and peptide identification. Protein quantification was performed using the LFQ algorithm implemented in MaxQuant software to reflect a normalized protein quantity deduced from razor + unique peptide intensity values. A minimal ratio count of one and a 2-min window for matching between runs were both required for quantification. Protein identifications assigned as 'Reverse' were excluded from further analysis. Contaminants were not removed from the dataset, as keratins are of special interest in the study of squamous epithelial cells. LFQ intensities were log2 transformed in Perseus version 1.3.0.4[68] and used in subsequent analyses.

## Quality control in discovery proteomics

The quality of discovery proteomics assays for tissues and fluids was evaluated by measuring deviations in the retention time for three trypsin autolysis peaks at $m/z$ 421.7584 + 2, 523.2855 + 2, and 737.7062 + 3. Moreover, four iRT peptides that were spiked into tissue samples were selected for verification of the retention time, intensity normalized by mean per sample, and intensity normalized by mean per group (iRT_Pep1: SSAAPPPPPR; iRT_Pep2: HVLTSIGEK; iRT_Pep3: GISNEGQNASIK; iRT_Pep4: IGDYAGIK) (Pierce™ Peptide Retention Time Calibration Mixture, Thermo Scientific, USA; 50 fmol injected). The peaks were evaluated in Skyline version 19.1[69] using the MSstats tool version 3.13.6[70]. Samples with deviated or recurrent absent intensities/retention times were excluded from the analysis.

## HPV genotyping

HPV genotyping was performed in 27 primary tumors from HNSCC patients that were included in the discovery proteomics analysis using the INNO-LiPA HPV Genotyping kit in an Autoblot 3000 system (Fujirebio, Japan). HPV positivity was achieved for primary tumors and used in all tissue analyses once matched tumoral and lymph node samples from the same patients were included in this study. Viral infection was not assessed in HNSCC patients used for buffy coat and saliva analysis due to the unavailability of primary tumor tissues. Sections (10 μm) were cut from FFPE blocks, deparaffinized in xylene, and rehydrated in a graded ethanol sequence. Genomic DNA was extracted using a standard protocol with proteinase K, phenol/chloroform, and ethanol treatment. DNA samples were quantified by spectrophotometry (NanoDrop ND-1000, NanoDrop Technologies, USA). INNO-LiPA is a line probe assay based on the principle of reverse hybridization for the identification of 32 different HPVs, including 13 high-risk genotypes. Biotinylated consensus primers (SPF10) were used to amplify a 65-bp region within the L1 region of multiple HPV types, and the resulting biotinylated amplicons were denatured and hybridized with specific oligonucleotide probes. A primer set for the amplification of human HLA-DPB1 was used to monitor sample quality

and extraction. The tests were performed according to the manufacturer's instructions.

## Expression profiling of HPV

HPV DNA-positive tumors ($n$ = 8 samples) were evaluated for the expression of viral transcripts using RT-qPCR. Total RNA was extracted from 10 μm FFPE sections using TRIzol reagent (Invitrogen, USA) and quantified using a Nanodrop ND-1000 spectrophotometer (NanoDrop Technology, USA). RNA was reverse transcribed using the High-Capacity cDNA Reverse Transcription Kit (Thermo Scientific, USA). PCR amplification was performed in an ABI Prism 7500 Sequence Detection System (Applied Biosystems, USA) using SYBR Mix (Applied Biosystems, USA). Data were acquired using the 7500 software version 2.3 (Applied Biosystems, USA). Two primer pairs were used to amplify E6 gene expression, and two pairs were employed to analyze E7 transcripts from HPV16[71] (Supplementary Data 7-1). GAPDH was used as a control for the sample quality. PCR reactions were performed in duplicate. Samples with amplification of at least one primer pair for E6 or E7 were considered positive for HPV expression. Two out of the 8 samples could not be evaluated due to poor FFPE RNA quality.

## Clustering global proteome datasets

The overlay among proteins identified in the proteomes was visualized by upset plots generated using the Intervene tool version 0.6.1[72]. Hierarchical clusters were generated in Python version 3.6 using log2 LFQ intensities from the tissues and fluids datasets. Missing values were replaced by random numbers drawn from a normal distribution with a width of 0.3 and a down shift of 1.8. The normal distribution was generated according to negative ranking of mean and standard deviation (μ', σ') from columns using the formulas μ' = μ column – shift × σ column, shift = 1,8; σ' = width × σ column, width = 0,3. Hierarchical clustering measurements were performed using fastcluster version 1.1.27 and SciPy version 1.6 packages[73,74]. Methods ('complete', 'weighted', and 'ward') and metrics ('bray-curtis', 'canberra', 'chebyshev', 'cityblock', 'correlation', 'cosine', 'dice', 'euclidean', 'hamming', 'jaccard', 'jensenshannon', 'kulsinski', 'mahalanobis', 'yule', 'matching', 'minkowski', 'rogerstanimoto', 'russellrao', 'seuclidean', 'sokalmichener', 'sokalsneath', and 'sqeuclidean') were combined, and dendrograms exhibiting the most evident clustering of proteins (protein clusters PC$_n$) or patients (clusters C$_n$) were selected for further analysis. Once data generated from clustering were compared among the six datasets, a single criterion (method/metric) was considered for the multisites in PC or C analysis. Heat-maps were drawn using the clustermap tool from the seaborn package version 0.11.1[75].

## Annotation of clusters and association with clinical data

Meaningful GO biological processes that were significantly enriched in protein clusters (PCs; Fig. 1) for blood and tissue datasets were selected using the GSEApy package version 0.9.18 in Python version 3.7 against the 'GO_Biological_Process_2018' background and the two-sided Fisher's exact test followed by Benjamini-Hochberg for correction of multiple hypotheses (adjusted $p ≤ 0.05$). A two-sided Fisher's exact test was employed to associate patient clusters (Cs; Fig. 4) with clinical and pathological features in the IBM SPSS Statistics software version 28.0 (IBM Corp.), and the association with survival was evaluated using Kaplan-Meier survival curves and the log-rank test ($p ≤ 0.05$). For fluids, we considered clinical characteristics for comparison that included age, sex, smoking habits, alcohol consumption, vital status (dead or alive), pathologic N (pN)[61], pathologic T (pT)[61], pathologic stage[61], overall survival, lymphatic invasion, perineural invasion, vascular invasion, depth of invasion, desmoplasia, inflammatory infiltrate, and surgical margin status. HPV DNA, HPV RNA, age, sex, anatomical site of the tumor, vital status (dead/alive), smoking habits, alcohol consumption, overall survival, disease-free survival, surgical margin status, recurrence, pT[61], pN[61], pathologic stage[61], depth

of invasion, histological grade according to WHO[60], invasion pattern, inflammatory response, degree of keratinization, nuclear pleomorphism, and perineural invasion were used for tissue analysis. Significant associations were visualized using GraphPad Prism version 8.2.1 (GraphPad; https://www.graphpad.com).

### Definition and annotation of metastasis signatures

Log2 LFQ intensity values were used to determine differentially abundant proteins between pN+ and pN0 conditions in Perseus software version 1.3.0.4 ($p \leq 0.05$; two-sided unpaired Student's $t$-test or $q \leq 0.05$; two-sided unpaired Student's $t$-test with Benjamini-Hochberg correction)[68]. The datasets that were analyzed included primary tumor – malignant cells (11 pN+, 14 pN0 patients), primary tumor and lymph node - non-malignant cells (13 pN+, 14 pN0), buffy coat samples (11 pN+, 13 pN0), and saliva cells (13 pN+, 11 pN0). Differentially abundant proteins exhibiting higher expression in pN+ compared to that in pN0 or those exclusively detected in pN+ were termed 'upregulated', while proteins exhibiting lower expression in pN+ compared to that in pN0 or those exclusively detected in pN0 were termed 'downregulated'. The differences in protein abundances between the two groups were termed 'Ratio' and calculated as mean LFQ intensity pN+/mean LFQ pN0, followed by log2 transformation. The intersection among datasets was visualized in upset plots using UpSet version 1.4.0[76]. GO biological processes were enriched using ShinyGO version 0.61 ($FDR \leq 0.05$; hypergeometric test followed by FDR correction)[77], and the top-10 overrepresented GO terms or related biological processes were visualized[78]. Samples were grouped according to the protein profile by applying PCA using the FactoMineR package version 1.34 in R version 3.6.2[79]. The statistical significance and magnitude of changes between pN+ and pN0 conditions are represented in volcano plots (R version 3.6.2). STRING version 11.0 was employed to retrieve protein-protein interaction networks[80]. The relationship between protein abundance (SRSF1, SRSF2, SRSF3, SRSF5, and TRA2A) and clinicopathological data was determined for the following features: HPV DNA, HPV RNA, age, sex, anatomical site of tumor, vital status (dead/alive), smoking habits, alcohol consumption, overall survival, disease-free survival, surgical margin status, recurrence, pT[61], pN[61], pathologic stage[61], depth of invasion, histologic grade WHO[60], invasion pattern, inflammatory response, degree of keratinization, nuclear pleomorphism, and perineural invasion (IBM SPSS Statistics software version 28.0; IBM Corp.; $p \leq 0.05$; two-sided unpaired Student's $t$-test or ANOVA).

### Comparison between tumor and metastasis proteomes

To avoid deviations in protein identification and quantitation caused by searches in individual datasets, RAW data obtained from MS runs in malignant cells from the primary tumor and lymph nodes ($n = 11$ samples/each dataset) and for the proteome of non-malignant cells from both sites ($n = 27$ samples/each) were combined in a unique search for malignant cells and a single search for non-malignant cells using MaxQuant software version 1.5.8.0 as described in the 'Discovery proteomics and data analysis' subsection. Log2 intensities were compared between tumor and lymph nodes using a two-sided unpaired Student's $t$-test and the $p$ values were adjusted using the Benjamini-Hochberg method ($q \leq 0.05$). Differentially abundant proteins exhibiting higher expression in lymph nodes compared to that in primary sites or those exclusively detected in lymph nodes were termed 'upregulated', while proteins exhibiting lower expression in lymph nodes compared to that in primary sites or those exclusively detected in the tumor site were termed 'downregulated'. The differences in protein abundances between the two groups were calculated as mean LFQ intensity in primary tumor/mean LFQ intensity in lymph node and termed 'Ratio'. The frequency of deregulated proteins was compared between malignant and non-malignant populations in GraphPad Prism version 8.2.1 (GraphPad, https://www.graphpad.com; $p \leq 0.05$; Chisquare test). Overrepresented GO biological processes were

determined using ShinyGO version 0.61 (Enrichment $FDR \leq 0.05$; hypergeometric test followed by FDR correction)[77] and a group of proteins enriched for actin-based cell movement (ACTA1, ACTN2, CASQ1, GSTM2, MYH7, MYL1, MYO1G, NEB, TPM3) or associated with EMT (VIM, FN1, CDH1) was selected for verification using PRM-MS and/ or RT-qPCR. LFQ intensities from discovery proteomics (DDA) assigned for malignant cells were also used to calculate EMT scores based on a 76-gene expression signature reported by Byers et al.[33]. This method was developed using signatures of EMT identified from non-small cell lung tumors and validated in additional cancer types[81]. The proteome of malignant cells determined in the discovery phase was matched with the 76 genes and the scores were calculated as the average abundance level of 'Mes' proteins minus the average expression level of 'Epi' proteins. The scores were compared between primary tumor and lymph node sites using the two-sided paired Student's $t$-test (normal distribution determined by Shapiro Wilk test) in the GraphPad Prism version 8.2.1 (GraphPad, https://www.graphpad.com; $p \leq 0.05$). Samples with higher EMT scores are more 'Mes', whereas those with a lower score are more 'Epi'. Hierarchical clustering of malignant and non-malignant populations was performed as reported in the 'Clustering global proteome datasets'.

### Selection of targets for ML analysis

Using discovery proteomics, we determined sets of proteins in tissues and fluids from HNSCC patients that may be used as markers of lymph node metastasis in a clinical context. Thus, we defined a series of criteria to prioritize particular proteins for further verification using targeted methodologies, including (i) global proteome datasets with clustering patterns associated with nodal status, (ii) metastasis-associated datasets with the best segregation pattern using PCA, (iii) proteins detected exclusively in one group or with $p \leq 0.05$ when using $FDR$ correction to compare pN+ and pN0 groups based on the Benjamini-Hochberg test, (iv) significant association with other clinical and pathological features ($p \leq 0.05$; two-sided unpaired Student's $t$-test, ANOVA or two-sided Fisher's exact test), (v) generation of high-performance protein or transcript pairs <Si, Cj> to separate pN+ and pN0 using ML (please see section 'Definition of prognostic signatures using machine learning'), and (vi) association with the immune contexture highlighted by an elevated gene expression in lymphoid tissues compared to that of other tissue types according to The Human Protein Atlas database (list of 1419 elevated genes in the lymphoid tissue transcriptome available from http://www.proteinatlas.org). Six microenvironment proteins satisfying most (if not all) of the criteria were selected for verification in buffy coat and saliva samples using SRM-MS and RT-qPCR for SRSF1, SRSF2, SRSF3, SRSF5, TRA2A (fulfilling criteria [i] to [vi]), and CD209 (fulfilling criteria [i] to [iii] and [vi] to [vii]). Moreover, due to the relevance of the immune system highlighted in several analyses throughout this work, 19 immune markers were selected for assessment in fluids using SRM-MS to quantify the proteins associated with immune populations, including leukocytes (CD45), myeloid cells (CD11b), T lymphocytes (CD3), CD4+ T lymphocytes (CD4), CD8+ T lymphocytes (CD8), Tregs (FOXP3 and CD25), B lymphocytes (CD19), NK cells (CD56 and CD16), monocytes (CD14, and CD64), non-classical monocytes (CD16), macrophages (CD64 and CD163), neutrophils (CD16, CD66b, and CD15), dendritic cells (CD11c and HLADR) and activated cells (CD80 and CD86).

### Reverse transcription quantitative PCR

The expression of genes coding for proteins enriched for actin-based cell movement (ACTA1, ACTN2, CASQ1, GSTM2, MYH7, MYL1, MYO1G, NEB, TPM3) were evaluated in 11 paired FFPE malignant cells isolated from HNSCC primary tumor and lymph nodes. Transcript levels of the genes SRSF1, SRSF2, SRSF3, SRSF5, TRA2A, and CD209 were evaluated in a 19-patient cohort of lymph node tissues (9 pN+ and 10 pN0), a 24-patient cohort of buffy coats (10 pN+, 14 pN0), and a 22-patient cohort

of saliva cells (10 pN+, 12 pN0), and they were then associated with nodal status. Total RNA was extracted using TRIzol reagent (Invitrogen, USA) and quantified using a Nanodrop ND-1000 spectrophotometer (NanoDrop Technology, USA). RNA was reverse transcribed using the High-Capacity cDNA Reverse Transcription Kit (Thermo Scientific, USA). PCR amplification was performed in an ABI Prism 7500 Sequence Detection System (Applied Biosystems, USA) using SYBR Mix (Applied Biosystems, USA). Data were acquired using the 7500 software version 2.3 (Applied Biosystems, USA). Primer set sequences were designed using the Primer-BLAST tool (accessed in Oct 2018 and Jun 2022; http://www.ncbi.nlm.nih.gov/tools/primer-blast/), and IDT OligoanalyzerTM version 3.1 (http://www.idtdna.com/analyzer/applications/oligoanalyzer) was used to predict the occurrence of dimers and secondary structures (Supplementary Data 7-1). To improve the assay success for FFPE samples, which are known to have high RNA degradation, all the oligonucleotides were designed to amplify small amplicons (50–100 pb in length)[82]. Serial dilutions (1:5) of a pool of cDNA samples were used to evaluate the amplification efficiency (E) according to the equation $E = 10(-1/slope)-1$. Primers possessing E values ranging from 95% to 105% were used for further analysis. PCR amplifications were performed in duplicate, and the specificity of the products was verified based on melting curve analysis. Negative and positive controls were included for each reaction. Positive controls consisted of a pool of samples representative of each biological material that was used as a 'normalizing' sample and allowed for the evaluation of intra-assay variations. *GAPDH* and *HPRT* were used as the reference genes for buffy coat experiments and *GAPDH* alone was considered for saliva and FFPE analysis. Relative quantification was calculated using the model proposed by Pfaffl[83], and the appropriate two-sided unpaired parametric (Student's *t*-test) or non-parametric (Mann-Whitney test) tests were applied to compare transcript levels between lymph nodes and primary tumors or pN+ and pN0 samples after evaluating the normality assumption by Shapiro-Wilk test (IBM SPSS Statistics software version 28.0, IBM Corp.; GraphPad Prism version 8.2.1, GraphPad, https://www.graphpad.com; $p \leq 0.05$). From the nine actin-based cell movement genes evaluated, only two (*ACTA1* and *TPM3*) had Cq-values detected across FFPE samples and were suitable for RT-qPCR analysis. Non-detects were imputed by the maximum cycle number (Cq = 40) for FFPE samples with satisfactory reference Cq.

## Selection of proteotypic peptides and transitions for PRM-MS

A parallel reaction monitoring-mass spectrometry (PRM-MS) analysis was carried using an EASY-nLC 1200 coupled to Orbitrap Exploris 240 mass spectrometer (Thermo Scientific, USA) to assess the regulation seven actin-based cell movement proteins (ACTA1, ACTN2, GSTM2, MYL1, MYO1G, NEB, and TPM3) and three EMT markers (CDH1, FN1, and VIM), as indicated by discovery proteomics. To facilitate building the method in a PRM-enabled platform based on DDA information generated using the EASY-nLC-Orbitrap Velos, a retention time (RT) predictor was created using Skyline version 21.2[69] by converting the RT of peptides from the discovery phase to an iRT matrix. The RT predictor was composed by 14 CiRT (Common internal Retention Time standards) as reference, plus Pierce™ Peptide Retention Time Calibration Mixture (Thermo Scientific, USA) and 73 targeted peptides. Next, five pools of 2-3 random FFPE samples were analyzed by DDA in the EASY-nLC 1200-Exploris 240 platform to build a HCD spectral library and perform a retention time verification. In addition, a complementary library of predicted HCD spectra was built using Prosit (NCE = 27)[84] allowing the selection of peptide peaks not readily detected by the DDA run in the Orbitrap Exploris 240 (Supplementary Data 7-2 and 7-3).

## PRM-MS and data analysis

In total, 11 matched HNSCC FFPE samples from primary tumor and lymph node malignant cells previously evaluated in the discovery phase were analyzed via PRM-MS. A volume of 4 µL of peptides was injected (44 fmol of iRT peptides) into the LC system equipped with an Acclaim PepMap100 trap column (75 µm × 2 cm, 3 µm, 100 A) and PepMap RSLC analytical column (75 µm × 25 cm, 2 µm, 100 A) for reversed-phase chromatography using (A) 0.1% formic acid in water, and (B) 0.1% formic acid in 80% acetonitrile / 20% water as mobile phases. Peptides were resolved over a 120-min gradient (5–38% B) at 250 nl/min, 50 °C. Eluting peptides were transferred to gas-phase using an EASY-Spray source operating at 1.6 kV. Targeted ions were monitored within a 10-min window, with an isolation window set to 1.4 m/z, AGC target set to 50%, Maximum Injection Time of 54 ms for CiRT and Pierce peptides and 120 ms for the peptides. Precursor ions were activated by HCD with normalized collision energy (NCE) of 27, and product ions analyzed in the Orbitrap at 30,000 resolution at 200/mz. With these configurations an average 9.5 Hz scan rate across the gradient was achieved. Even at lower scan rates (5 Hz), when up to 20 concurrent precursors co-eluted, a minimum of 10 points per peak was measured per peptide (median peak width of 46 sec). The MS data was acquired using the Xcalibur software version 4.4 (Thermo Fisher Scientific, USA) and samples were randomized in the R version 3.6.2 (Supplementary Data 1-3).

For label-free quantification of PRM-MS data, total area of each peptide was calculated by integrating the XICs of top-ranking transitions. Selection of peaks was directed by a 10-min window of either the predicted RT times or the identification time from DDA runs on Orbitrap Exploris 240. Peaks were inspected using Skyline version 21.2[69] and peptide signal was considered specific upon matching with spectral libraries (DDA or Prosit; dotp > 0.7)[84], and low mass error (<10 ppm). Out of 1,430 datapoints (22 samples and 65 precursors), 90% exhibited dotp > 0.8 and <10 ppm mass error (Supplementary Data 4-1). The PRM-MS data was processed with the appropriate two-sided parametric or non-parametric paired test selected after testing normality of the data with the Shapiro-Wilk test (two-sided paired Student's *t*-test or two-sided Wilcoxon signed-rank test) in the GraphPad Prism version 8.2.1 (GraphPad, https://www.graphpad.com; $p \leq 0.05$).

## Selection of proteotypic peptides and transitions for SRM-MS

Proteotypic peptides were selected for each of the six microenvironment markers and 19 immune targets based on the number of residues, their hydrophobicity, the presence in our DDA datasets of buffy coat or saliva, and SRMAtlas (http://www.srmatlas.org/) evidence when the peptide was not identified in our data[85,86]. The selection of the best transitions was performed using spectral libraries built from our own DDA data, and transitions retrieved from the SRMAtlas were included when they were not detected with high reliability from DDA. The most intense transitions were selected for the final SRM-MS method. To monitor the microenvironment proteins associated with lymph node metastasis (SRSF1, SRSF3, SRSF5, TRA2A, and CD209), nine proteotypic peptides were selected and purchased as crude heavy-isotope-labeled peptide standards (JPT Peptide Technologies, USA) (Supplementary Data 7-2). SRSF2 did not possess proteotypic peptides fulfilling the criteria mentioned above and was excluded from the SRM-MS analysis. Stable isotope-labeled peptides were synthesized with $^{13}C_6,^{15}N_2$- lysine or $^{13}C_6,^{15}N_4$-arginine (+8 or +10 Da, respectively) that were localized preferentially at the C-terminal of the peptide. Three to four transitions were monitored in light and heavy channels for each peptide for a total of 68 transitions (Supplementary Data 7-4). The concentration of heavy peptides was optimized based on serial dilutions in buffy coat (1:2 and 1:4 dilutions) and saliva (1:2, 1:10, 1:30, and 1:60 dilutions) samples to determine the best concentration with a maximum 1:10 light/heavy ratio, and this resulted in a range of 9 to 200 fmol/µg of matrix. Considering the 19 immune markers (CD45, CD3, CD4, CD8, FOXP3, CD25, CD19, CD56, CD16, CD11b, CD14, CD64, CD163, CD66b, CD15, CD11c, HLADR, CD80,

and CD86), heavy peptides were not included in the analysis, and the retention times were predicted by building a calculator in Skyline software version 19.1[69] based on ACTB peptides retention times from DDA data while considering spectral libraries built from lymph node FFPE tissues, buffy coat, and saliva DDA information. When the calculator could not be applied due to the lack of peptide detection in our DDA libraries, transitions available from the SRMAtlas were monitored. Two to four peptides were included for each of the 19 immune proteins, and a total of 54 label-free peptides were monitored (Supplementary Data 7-2). Of the 19 targets, nine were detected in the DDA dataset (Supplementary Data 7-2). None of the peptides from 10 out of 19 markers matched the retention times predicted from the calculator or the transitions described in the SRMAtlas in one buffy coat and/or one saliva sample from HNSCC patients, and they were excluded from further analysis. The remaining nine immune markers that included CD3 (T lymphocytes), CD4 (CD4+ lymphocytes), CD8 (CD8+ lymphocytes), CD11b (myeloid cells), CD14 (monocytes), CD16 (neutrophils, NK, non-classic monocytes), CD19 (B lymphocytes), CD45 (leukocytes), and CD66b (neutrophils) were retained for subsequent analysis, and four to six transitions were monitored in light channels per peptide for a total of 80 transitions (Supplementary Data 7-4).

## SRM-MS and data analysis

The relative abundance of six prioritized microenvironment proteins from lymph nodes (SRSF1, SRSF2, SRSF3, SRSF5, TRA2A, and CD209) and nine immune markers (CD3, CD4, CD8, CD11b, CD14, CD16, CD19, CD45, and CD66b) was evaluated in 19 buffy coat (7 pN+, 12 pN0) and 25 saliva (16 pN+, 9 pN0) samples from HNSCC patients. Sample preparation was performed as described in the 'Sample preparation for discovery proteomics' subitem. Samples were analyzed on a Xevo TQ-XS triple quadrupole mass spectrometer (Waters, USA)[4,14]. Buffy coat or saliva digests (1.6 μg) were resolved over a 60-min gradient using an Acquity UPLC-Class M equipped with a trap column (Acquity UPLC BEH C18 130 A, 5 μm, 300 μm × 50 mm, Waters, USA) and a BEH Shield C18 IonKey column (10-cm × 150-μm ID packed with 1.7-μm C18 particles, Waters, USA) at 1.2 a flow rate and a temperature of 40 °C. The MeCN gradient started at 2% B (MeCN, 0.1% formic acid), and this was followed by a linear ramp to 40% B over 45 min, then by a step increase to 85% B for 47 min, and finally conditioning at 2% B for 60 min. Mass spectrometry analysis of eluting peptides was performed via SRM-MS mode with quadrupoles Q1 and Q3 operating as unit mass resolution (0.7 Th full width at half maximum). The optimal collision energy was determined for each peptide using Skyline 19.1[69]. Scheduled SRM-MS acquisition was adjusted to a 3-min elution window with dwell times automatically set in MassLynx version 4.2 (Waters, USA) to achieve at least ten points per peak over a 30-s elution profile. A blank sample (water) was injected between two consecutive sample runs with the same gradient to minimize carry-over. The MS data was acquired using the MassLynx software version 4.2 (Waters, USA) and samples were randomized in the R version 3.6.2 (Supplementary Data 1-3). Peaks were inspected using Skyline software version 19.1[69] and quantified by calculating the ratio between light and heavy intensities for each peptide (intensity = sum of transitions) for SRSF1, SRSF3, SRSF5, TRA2A and CD209. We observed the alignment of elution times, co-elution of all transitions, and relative intensity correlation with the spectral library (dotp, close to 1). The relative intensity correlation between light and heavy transitions (rdotp) was also evaluated, resulting in (i) the inclusion of peptides from samples exhibiting rdotp ≥ 0.9, (ii) exclusion of peptides possessing rdotp ≤ 0.8, and (iii) manual evaluation of peptides exhibiting rdotp varying from 0.8 to 0.9. For immune markers, light intensities were considered for quantification and peaks for light peptides were evaluated for the predicted elution time and co-elution of all transitions.

## Quality control in SRM-MS and PRM-MS

Several quality controls were used to evaluate the reliability of the SRM-MS and PRM-MS analysis. A system suitability protocol was implemented to assess the equipment performance by monitoring 18 peptides from a mixture of 200 ng of digested bovine serum albumin (BSA) and 5 fmol of iRT (Pierce™ Peptide Retention Time Calibration Mixture, Thermo Scientific, USA) prior to each batch of four samples. To monitor sample quality, an iRT mixture (32 fmol injected in SRM-MS and 44 fmol injected in PRM-MS) was spiked into each sample, and four peptides with their respective three or four transitions each were monitored as a control for retention time and intensity shifts in liquid chromatography (Supplementary Fig. 4a, b; Supplementary Fig. 4d–g; Supplementary Data 4-1 and 4-3; Supplementary Data 7-2). The percentage of carryover was assessed by comparing the absolute intensity of 13 heavy peptides (Supplementary Fig. 4c) in a blank (water) injection with the absolute intensity of the same peptides in a preceding matrix (buffy coat) run[38].

## Evaluation of external datasets

We herein used eight external datasets to verify the hypothesis raised by HNSCC proteomic analysis, including (i) scRNASeq data of tumors and lymph nodes from 18 HNSCC patients[5], (ii) scRNASeq data of a PBMC sample from a health donor (https://www.10xgenomics.com), (iii) scRNASeq and proteomic data of an atlas of 28 health tissues[21], (iv) immune cell counts (hemograms) of blood from 25 HNSCC patients, (v) RNASeq data of tumors from a 500 HNSCC patient-cohort (TCGA), (vi) RNASeq data of tumors from a 428 HNSCC patient-cohort (TCGA), (vii) proteomic data from 22 human PBMC subpopulations[22], and (viii) a list of reviewed proteins with their respective annotated gene ontology (GO) biological processes from Uniprot. A summary of the external datasets and their application in our study is presented in Supplementary Data 9, and appropriate accessible links or accession codes are provided under the 'Data availability' section.

## Association of cell counts with metastasis

Immune cell counts were obtained from the clinical laboratories of São Vicente de Paula and Sobam Hospitals, Brazil, for a 25 HNSCC-patient cohort (9 pN+; 16 pN0). From these, buffy coat or saliva samples from 22 patients were also included in this study for discovery proteomics (DDA), SRM-MS, and RT-qPCR analysis. The clinical laboratories quantified cells from blood samples (cell count/mm3) and included data for total leukocytes, lymphocytes, neutrophils, monocytes, eosinophils, and basophils. Cell counts were compared between pN+ and pN0 samples using a two-sided unpaired Student's t-test and a p under 0.05 was considered to determine significance.

## Searching for GO splicing factors related to metastasis

A list of reviewed proteins with their respective annotated gene ontology (GO) biological processes was retrieved from Uniprot (https://www.uniprot.org). Proteins annotated in GO processes containing the term 'splicing' were filtered out and the 421 remaining GO splicing proteins were selected to verify their gene expression in immune cells from pN+ and pN0 HNSCC primary tumors[5], as detailed in the section 'scRNASeq processing and differential expression'. These proteins were also searched in the bulk proteomes herein associated with metastasis that are presented in Supplementary Data 2 for non-malignant cells isolated from primary tumor ($n = 85$ proteins differentially abundant in pN+ vs. pN0), non-malignant cells from lymph nodes ($n = 201$ proteins), and fluids ($n = 80$ proteins for buffy coat and $n = 54$ proteins for saliva). A list of targeted genes that have experimentally validated interactions between RNA sites and the GO splicing factors associated with metastasis was recovered using the SpliceAid-F database[35]. A two-sided Fisher's exact test was employed to associate protein or gene expression (up- or downregulation in pN+

when compared to pN0) with the immune cell type or microenvironment in the IBM SPSS Statistics software version 28.0 (IBM Corp.)

## Cross validation of protein and gene expression
RNASeq samples of 500 HNSCC primary tumor samples were retrieved from TCGA (https://portal.gdc.cancer.gov) using the TCGA-HNSC identifier. Gene expression was compared to the protein levels determined in our study for malignant cells from the primary tumor. Spearman correlations were performed using the FPKM median from RNASeq data and the log2 LFQ mean of proteomes.

## scRNASeq processing and differential expression
A HNSCC scRNASeq dataset from 18 HNSCC patients was retrieved from the Gene Expression Omnibus using the query number GSE103322[5]. We followed the standard workflow from Seurat package version 4.0.0[87] with parameters min.cells=0 and min.features=200 for matrix import and nfeatures = 2000 for FindVariableFeatures(), and 30 dimensions were used after reduction with principal component analysis (PCA) followed by uniform manifold approximation and projection (UMAP) map construction. Clusters were identified using a resolution of 1.2 and the cell types were annotated according to the literature[5]. Additional annotation was made to subdivide fibroblasts into CAFs (expressing *FAP* and *THY1*), myofibroblasts (expressing *ACTA2*), and skeletal muscle cells (expressing *DES*), and to identify T cell subtypes. The original annotation of plasma/B cells was curated and updated based on B cell (*MS4A1* and *BANK1*) and plasma cell markers (*MZB1* and *IGLL5*). For each population within the HNSCC dataset, markers that define clusters via differential expression (cluster markers) from previous analysis, the microenvironment markers (*SRSF1, SRSF2, SRSF3, SRSF5, TRA2A,* and *CD209*) or a list of GO splicing factors were tested for differential expression among subpopulations using a two-sided Wilcoxon test within the FindMarkers() function and Benjamini-Hochberg as *p* adjustment method. The PBMC 3k dataset was retrieved at https://support.10xgenomics.com/single-cell-gene-expression/datasets/1.1.0/pbmc3k and processed as previously reported[87]. Cluster markers in PBMC populations were identified using the FindAllMarkers function with min.pct = 0.1 and logfc. threshold = 0.25. The differentially expressed proteins determined in our study for the buffy coat, tumor microenvironment, and lymph node microenvironment were selected and used to query the filtered cluster markers.

## Cell type deconvolution using CIBERSORTx
CIBERSORTx version 1.0 was used to infer cell types in our bulk proteome data using the scRNASeq matrices from section 'scRNASeq processing and differential expression' as reference. First, the signature matrix from HNSCC used in CIBERSORTx version 1.0[5] generated from scRNASeq data was tested to check if the estimation of a given cell type could be disfavored by using proteomic information. For that, correlations were performed between protein abundance and genes from this signature matrix using data from an expression atlas of 28 out of 29 available healthy tissues (we excluded testis)[21]. From the 2973 genes present in the CIBERSORTx version 1.0 signature matrix, 1840 were found in the atlas. Gene names with multiple isoforms were aggregated using the median FPKM to match protein annotations. Spearman correlations between protein abundance and gene expression were calculated across the 28 tissues. Each gene was assigned to a single cell type according to its highest value in the signature matrix. Additionally, the deconvolution cross-validation of PBMC proteomes in a scRNASeq matrix was performed using data generated by Rieckmann et al.[22]. Proteome LFQ intensities from steady-state cells were aggregated by median for different replicates and log2 transformed. A matrix containing the values from the genes associated with the proteins were then given as a mixture to CIBERSORTx

version 1.0 using the previously generated PBMC scRNASeq matrix (https://www.10xgenomics.com).

Cell type annotations provided in the processed matrix from the literature[5] were used to generate a scRNASeq signature matrix specific for HNSCC tissues and infer cell types in our tissue bulk proteomes (non-malignant cells from primary tumor and lymph nodes). The signature matrix was created using the following parameters: Min. expression = 0, replicates = 4, and sampling = 1. Proteomic data (log2 transformed LFQ intensities) from HNSCC non-malignant samples were used to infer cell type abundance from the scRNASeq signature matrix using CIBERSORTx version 1.0[20]. The cell signature derived from healthy PBMC subpopulations was used as a reference to deconvolute whole blood proteome information using log2 LFQ intensities. All runs were performed with no batch correction, disabled quantile normalization, absolute mode, and with 100 permutations.

## Cell type enrichment with xCell and association with metastasis
xCell version 1.0, which performs Gene-set enrichment analysis in previously defined immune and stromal cell types, was used to evaluate differences between pN+ and pN0 samples[30]. TCGA data available on the xCell website (https://xcell.ucsf.edu/) were downloaded and filtered using clinical information from cBioportal as Sample.Type = Primary and excluding samples without lymph node stage annotation. HNSCC pN+ and pN0 samples ($n = 428$ patients) were classified according to information in Neoplasm.Disease.Lymph.Node.Stage. American.Joint.Committee.on.Cancer.Code. Spearman correlations were performed between the signature scores and pN+ /pN0 outcome filtering for $p \leq 0.05$.

## RNASeq data analysis
Bioinformatics analysis of third-party RNAseq data was performed in R environment version v 4.0.0 using Hmisc version 4.4-0 for correlations, dplyr version 1.0.0, tibble version 3.0.1, data.table version 1.13.0, and readxl version 1.3.1 for data manipulation, SummarizedExperiment version 1.18.2, GEOquery version 2.56.0[88], and TCGAbiolinks version 2.16.0[89] for data retrieval, ggpubr version 0.3.0, ggplot2 version 3.3 for plotting, and Seurat version 4.0.0[87] for scRNASeq analysis and manipulation.

## Definition of prognostic signatures using machine learning
The predictive power of peptides, proteins, and/or transcripts from the microenvironment (SRSF1, SRSF2, SRSF3, SRSF5, TRA2A, CD209) and immune targets (CD3, CD4, CD8, CD11b, CD14, CD16, CD19, CD45, CD66b) to distinguish HNSCC patients based on locoregional metastasis status was determined using an ML approach (pN+ vs. pN0). The steps used to obtain and validate the signatures are presented in Fig. 6a. First, (1) quantitation data acquired for the microenvironment and immune targets in buffy coat and saliva using SRM-MS or RT-qPCR were selected (individual datasets: $n = 19$ and 24 patients for SRM-MS and RT-qPCR in buffy coat, respectively; $n = 25$ and 22 patients for SRM-MS and RT-qPCR in saliva, respectively; combined datasets: $n = 15$ patients for buffy coat, $n = 14$ patients for saliva; only patient samples that were evaluated by both SRM-MS and RT-qPCR were considered in the combined datasets) (Supplementary Data 4-5 and 4-6). We also performed ML analysis of protein (DDA) and transcript (RT-qPCR) data from lymph node microenvironment cells with the purpose of selecting targets (please see section 'Selection of targets for verification'). In total, we used 14 datasets representing the quantification of peptides, proteins, transcripts, and their combinations (Supplementary Data 5-1). The missing values were replaced with 0. For each dataset, we ran a sequence of steps to capture a set of candidate signatures that performed well in the classification task (pN+ vs. pN0) using ROC AUC as measurement for prioritization, with the pN+ as the positive class. Then, (2) variables were combined to create all possible signatures (Si) of size (N) 1 to 5, where N is the maximum number of peptides/

proteins/transcripts in each signature and was arbitrarily set to 5. Next, (3) each signature was used to create different classification models (Cj) that included ridge, linear SVM, lasso, linear discriminant, perceptron, decision tree, naive Bayes, GBM, random forest, and RBF SVM. Each pair signature-classifier <Si, Cj> was evaluated within a repeated stratified k-fold cross-validation. The total number of tested models per pair <Si, Cj> was defined as R * K, with R = 10 repetitions and K = the smallest class size. Finally, (4) a list of potential signatures <Si, Cj> to discriminate pN+ and pN0 patients was selected for each dataset considering the evaluation step (3). The pair <Si, Cj> with the highest average ROC AUC was named top-1, and the respective curves were represented. Hence, we identified pairs <Si, Cj> with potential equivalent performance, comprising the pairs for which the distribution of AUC values observed in the repeated cross-validation was not statistically different from the ones observed with the top-1 pair ($p \geq 0.05$; two-sided unpaired Student's $t$-test). Specifically, if we could not reject the hypothesis of a pair P possessing an equal average ROC AUC to the top-1 pair, we included pair P in the final list of selected pairs for each dataset. Furthermore, we executed a permutation test (PT) to calculate a $p$-value against the null hypothesis that the data from each pair and the classes pN+/pN0 are independent using the sklearn programming kit version 0.23.1[90]. Top-1 and all equivalent selected pairs with ROC AUC $\geq 0.85$ and PT $p \leq 0.05$ were prioritized as high-performance candidate signatures. In addition to AUC and PT $p$, we report sensitivity, specificity, and precision for every high-performance candidate pair <Si, Cj > (Supplementary Data 5-4). Heatmaps were generated using GraphPad Prism v8.2.1 (GraphPad software; https://www.graphpad.com).

## Flow cytometry analysis

The expression of SRSF3 and TRA2A proteins was evaluated in a 20-HNSCC patient cohort (10 N+, 10 N0) using flow cytometry. Cell suspensions were stained for surface markers using anti-CD45-BUV805 (BD Horizon, USA; clone HI30; 1:500 dilution), anti-CD3-BV421 (Biolegend, USA; clone UCHT1; 1:300 dilution), anti-CD4-BV605 (BD Horizon, USA; clone SK3; 1:300 dilution), anti-CD8-BV650 (BD Horizon, USA; clone RPA-T8; 1:500 dilution), anti-CD25-BUV563 (BD Horizon, USA; clone 2A3; 1:500 dilution), anti-CD56-APC (BD Horizon, USA; clone B159; 1:60 dilution), anti-CD19-BV750 (BD OptiBuild; clone SJ25CI; 1:500 dilution), anti-CD209-PE-Cy7 (Biolegend, USA; clone 9E9A8; 1:20 dilution), anti-CD14-APC-Cy7 (Biolegend, USA; clone 63D3; 1:200 dilution), anti-CD15-PE (Biolegend, USA; clone HI98; 1:20 dilution) and anti-CD11b-BV650 (BD OptiBuild; clone ICRF44; 1:500 dilution) monoclonal antibodies by incubation for 30 min with antibody solutions, followed by washes. TRA2A antibody (Sigma-Aldrich, USA; polyclonal; 1:100 dilution) was labeled by Zenon™ Alexa Fluor™ 750 Rabbit IgG Labeling Kit (Thermo Scientific, USA), as recommended by the manufacturer. The cells were permeabilized using the BD Pharmingen™ Transcription Factor Buffer Set (BD Biosciences, USA) for 40 min at 4 °C and further stained with anti-SRSF3-FITC (Abcam; clone EPR16976; 1:20 dilution) and anti-TRA2A-Alexa Fluor 750 for 40 min at 4 °C followed by washes. Two PBMC samples from HNSCC patients were included as controls for surface markers. The preparations were analyzed using a FAC-Symphony™ equipment (BD Biosciences, USA) and the FlowJo version 10.8 software (BD Biosciences, USA). At least 30,000 gated events were acquired assuring the reliability of positive populations. Comparisons between N+ and N0 groups were performed using MFI and proportion of cells in the GraphPad Prism v8.2.1 software (GraphPad; https://www.graphpad.com). Data were tested for normality using Shapiro-Wilk test to further guide the selection of the appropriate statistical test used to verify differences between N+ and N0 samples (two-sided unpaired Student's $t$-test or two-sided Mann-Whitney test; $p \leq 0.05$).

## Statistical analysis

Statistical analyses of the data herein generated were performed using R version 3.6.2, IBM SPSS Statistics version 28.0 (IBM Corp.) or GraphPad Prism version 8.2.1 (GraphPad; https://www.graphpad.com) and are indicated in the figure legends, results, and methods sections. All proteomics quantitative datasets from the discovery phase were log2 transformed to reduce skewness, and a parametric two-sided unpaired Student's $t$-test was used for group comparison (pN+ vs. pN0). Multiple comparisons were adjusted using the Benjamini-Hochberg correction when feasible. The relation of protein or transcript levels from targets evaluated by SRM-MS, PRM-MS, RT-qPCR, or flow cytometry to the desired features was verified using the appropriate two-sided parametric or non-parametric test selected after testing normality of the data with the Shapiro-Wilk test (unpaired or paired Student's $t$-test, Wilcoxon signed-rank test, or Mann-Whitney test). The statistics for functional annotation or cell type enrichment are presented in the respective sections. Statistical significance was established at $p \leq 0.05$.

## Reporting summary

Further information on research design is available in the Nature Portfolio Reporting Summary linked to this article.

## Data availability

The mass spectrometry proteomics data generated in this study are available at ProteomeXchange via the PRIDE partner repository[91] and Panorama repository. DDA proteomics data are available at ProteomeXchange with the dataset identifier PXD027780. SRM-MS data and machine learning results are available through the Panorama repository at the link https://panoramaweb.org/16fpvB.url and ProteomeXchange dataset identifier PXD027984. PRM-MS data are available at https://panoramaweb.org/0UwcEI.url in the Panorama repository and ProteomeXchange dataset identifier PXD036311.

The following public RNASeq and proteomic data were downloaded and analyzed in this study: scRNASeq data of tumors and lymph nodes from 18 HNSCC patients (Gene Expression Omnibus, dataset identifier GSE103322)[5], scRNASeq data of a PBMC sample from a health donor (https://support.10xgenomics.com/single-cell-gene-expression/datasets/1.1.0/pbmc3k), scRNASeq and proteomic data of an atlas of 28 health tissues (RNA-Seq data: Array Express, dataset identifier E-MTAB-2836; Mass-spectrometry based proteomic data: ProteomeXchange Consortium, dataset identifier PXD010154)[21], RNA-Seq data of tumors from a 500 HNSCC patient-cohort (TCGA; https://portal.gdc.cancer.gov), RNASeq data of tumors from a 428 HNSCC patient-cohort (TCGA; https://portal.gdc.cancer.gov), and proteomic data from 22 human PBMC subpopulations (ProteomeXchange Consortium, data set identifier PXD004352)[22]. A list of proteins with their respective annotated gene ontology (GO) biological processes was retrieved from Uniprot (https://www.uniprot.org). The remaining data are available within the Article, Supplementary Information. Source data are provided with this paper.

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

## Acknowledgements

This work was supported by FAPESP under Grant numbers 2009/54067-3 [A.F.P.L.], 2010/19278-0 [A.F.P.L], 2016/07846-0 [A.F.P.L.], 2018/18496-6 [A.F.P.L.], 2015/19191-6 [A.F.B.L.], and 2019/21815-9 [A.F.B.L.], CNPq under Grant numbers 305851/2017-9 [A.F.P.L.] and 310392/2021-7 [A.F.P.L.], and ANID-FONDECYT Grant number 1190775 [W.A.G.A.]. This work was also partially supported/or had resources from the Brazilian Federal Government provided to the Center for Research in Energy and Materials (CNPEM), a private non-profit organization under the supervision of the Brazilian Ministry for Science, Technology, and Innovation (MCTI). The proteomics analysis was performed at the Mass Spectrometry Laboratory of the Brazilian Biosciences National Laboratory (LNBio), that is part of CNPEM. The Mass Spectrometry Laboratory staff are acknowledged for their assistance during the experiments (Proposal number MAS-22044). We also acknowledge Prof. Dr. Tsai Siu Mui, CENA, USP, for the use of the Leica Laser Microdissection System LMD6 (FAPESP Grant number 2009/53998-3), Waters Corporation for providing access to the Acquity UPLC-Class M system coupled with a Xevo

TQ-XS triple quadrupole mass spectrometer, Prof. Guilherme Telles, IC, UNICAMP, for his support with the selection of proteotypic peptides for SRM-MS analysis, and the Laboratory for Integrative and System Biology (LaBIS) for the use of the LaBIS Cloud (FAPESP Grant numbers 2011/00417-3 and 2015/50612-8).

## Author contributions

Conceptualization: A.F.B.L., A.F.P.L.; Methodology: A.F.B.L., A.F.P.L.; Formal Analysis: A.F.B.L., L.X.N., G.A.C., M.A.M.P., H.H., F.M.S.P., N.A.L.G.; Investigation: A.F.B.L., L.X.N., G.A.C., D.C.G., J.S., S.Y., C.R., R.R.D., A.G.C.N., T.D.R., B.P.M., N.A.L.G., B.A.P., P.A.L., R.A.L.S., I.I.D.; Resources: A.A.N.R., A.L.M.C., N.K.C., F.V.M., K.J.G., L.L.V., M.U., A.C.P.R., T.B.B., M.B., W.A.G.A., A.F.P.L.; Writing – Original Draft: A.F.B.L.; Writing – Review & Editing: A.F.B.L., A.F.P.L., T.S.M., L.L.V., L.P.K.; Supervision: A.F.P.L.; Funding Acquisition: A.F.P.L.

## Competing interests

The authors declare no competing interests.

## Additional information

[1]Laboratório Nacional de Biociências - LNBio, Centro Nacional de Pesquisa em Energia e Materiais - CNPEM, Campinas, SP 13083-100, Brazil. [2]Laboratório de Bioinformática e Biologia Computacional, Divisão de Pesquisa Experimental e Translacional, Instituto Nacional do Câncer - INCA, Rio de Janeiro, RJ 20231-050, Brazil. [3]Instituto de Ciências Matemáticas e de Computação, Universidade de São Paulo - USP, São Carlos, SP 13566-590, Brazil. [4]Centro de Biologia Molecular e Engenharia Genética – CBMEG, Universidade Estadual de Campinas - UNICAMP, Campinas, SP 13083-887, Brazil. [5]Departamento de Diagnóstico Oral, Faculdade de Odontologia de Piracicaba, Universidade Estadual de Campinas - UNICAMP, Piracicaba, SP 13414-903, Brazil. [6]Departamento de Ciencias Básicas Biomédicas, Facultad de Ciencias de la Salud, Universidad de Talca - UTALCA, Talca, Maule, Chile. [7]Departamento de Radiologia e Oncologia, Faculdade de Medicina, Universidade de São Paulo – USP, São Paulo, SP 01246-903, Brazil. [8]Centro Internacional de Pesquisa - CIPE, A.C. Camargo Cancer Center, São Paulo, SP 01508-010, Brazil. [9]Departamento de Clínica Médica, Faculdade de Medicina de Jundiaí - FMJ, Jundiaí, SP 13202-550, Brazil. [10]Departamento de Cirurgia de Cabeça e Pescoço, Faculdade de Medicina de Jundiaí - FMJ, Jundiaí, SP 13202-550, Brazil. [11]Departamento de Patologia, Faculdade de Ciências Médicas, Universidade Estadual de Campinas – UNICAMP, Campinas, SP 13083-887, Brazil. [12]Hospital Israelita Albert Einstein, São Paulo, SP 05652- 900, Brazil. [13]Instituto do Câncer do Estado de São Paulo - ICESP, Faculdade de Medicina, Universidade de São Paulo - USP, São Paulo, SP 01246-000, Brazil. [14]Departamento de Cirurgia de Cabeça e Pescoço e Otorrinolaringologia, A.C. Camargo Cancer Center, São Paulo, SP 01509-900, Brazil. [15]Departamento de Cirurgia de Cabeça e Pescoço, Faculdade de Medicina, Universidade de São Paulo - USP, São Paulo, SP 01246-903, Brazil. [16]Facultad de Odontología, Patología Oral y Maxilofacial, Universidad de Los Andes, Santiago, Chile. [17]Centro de Investigación e Innovación Biomédica, Universidad de Los Andes, Santiago, Chile. ✉e-mail: adriana.paesleme@lnbio.cnpem.br

