## [Peer Review File · Nature Communications]

Connecting multiple microenvironment proteomes uncovers the biology in head and neck cancerREVIEWER COMMENTS

Reviewer #1, expertise in proteomic analysis and machine learning (Remarks to the Author):

In this study the authors have performed global proteomic analysis of HNSCC primary tumors with matched negative/positive lymph nodes, in addition to buffy coat and saliva samples. Furthermore, malignant tissue/lymph were separated into malignant and non-malignant sections for analysis. Conceptually, this should produce a valuable dataset that can help dissect HNSCC biology and metastasis, especially due to the collection of paired samples. But, in its current form, the study has major shortcomings that result in a failure to realize the potential of the data.

MAJOR CONCERNS:

* The tissue and lymph node samples were all collected in Chile, while all the buffy coat, saliva and flow cytometry samples were collected in Brazil. This stratification of sample collection can result in the introduction of unknown biases that cannot be computationally deconvoluted. Furthermore, the number of samples is relatively small for the extent of subdivision and type of analyses performed in the manuscript.

* A lot of analyses and conclusions are drawn from correlation to RNAseq data from TCGA or elsewhere. It is well know (and the authors also show) that the expected overall correlation between gene expression and protein abundance is about 0.4-0.5, with many genes/proteins in signaling pathways prone to have lower correlation. Given this, the study's reliance on gene-protein correlation has most likely skewed results to those gene/proteins/pathways that are more correlated.

* I believe the ML analysis is flawed. There are hundreds to thousands of models generated using a small number of samples (14-25) with cross validation for model evaluation and no test dataset. Under these conditions, it is highly likely that the models are overfit. When so many models are being evaluated, cross validation can only be used to choose the best model, and a separate test set is required to assess model performance. At the very least, permutation testing should be used to show that the model performance is not due to random chance.

An independent 20 HNSCC patient cohort is used to measure SRSF3 and TRA2A. The distribution of these proteins do not appear to dramatically separate the N0 and N+ populations. A better use of this independent cohort would be to generate an independent test set for validating the models.

* Use of statistical tests, multiple-testing correction and parameter choices are inconsistent throughout the paper. It almost appears like a lot of choices were tried till the desired result was obtained. Some illustrative (not exhaustive) examples include:

- Fig 1B: Each hierarchical clustering uses a different metric. A single metric should be used for all datasets.

- Nominal p-values are used in most analyses, and BH-FDR in some others. All analyses should consistently use BH-FDR p-values.

- Fig 6D uses 3 different tests, and the lowest p-value is chosen. A single test (preferably Wilcoxon) should be used.

* The manuscript repeatedly makes overstated causal claims based on correlation/GO enrichment results. Some (not exhaustive) examples include:

- "... metastasis recapitulates the content of primary tumors" (Pg 13, lines 282-3). Based on what?

- "... differential protein is strongly involved in actin-based cell movement" (Pg 13, lines 283-284). Based on this GO enrichment, the authors conclude that the EMT program is activated in the cell, even though no EMT related GO terms have been enriched. The rest of page 13 (lines 292-304) is therefore unsubstantiated speculation.

- Page 15, lines 336-7: Over-interpretation of correlation results combined with GO enrichments.

- Some other overstatements (in my opinion) in the Discussion include (page:lines) 24:558-9, 25:580-1, 25:585-6, 26:603-4.

* How are sample pairs from the same patient treated? It appears like the increased statistical power arising from the availability of multiple samples from the same patient is not fully utilized. When a paired analysis is done, it is not clear how the samples are analyzed to take sample-specific correlation into account.

OTHER CONCERNS/ISSUES:

* When referring to "Methods" in the body of the text, please specify which section in the Methods the reader should reference. As it stands now, it is unclear where in the methods to find the relevant

information. For example, there is no section in the methods (that I could find) where identification of cluster markers (page 11, line 232) is described.

* Figure 3D - Metastasis column does not show any strong correlations, contradicting the statement made in Page 11 lines 253-4.

* Figure 2C, 2D and 2E essentially restate the same information. The authors claim this as additional evidence (page 10, lines 222-24).

* PCA analysis (Pg 16) is just replicating results observed in Fig 5A since only differential proteins are used.

* Some figures take up too much space for the information they present (eg. Fig 2B, Fig 4B, etc.) These should be either eliminated or moved to the supplement.

* Title: Do the authors mean "Connecting multiple microenvironment proteomes ..."? "Wiring" is an unusual word to use in this context.

Reviewer #2, expertise in head and neck cancer and TME (Remarks to the Author):

The authors use proteomic approaches to profile multiple types of biospecimens from a cohort of 59 HNSCC patients and use the data to gain insight into HNSCC biology. This represents a large and fairly unique data set for the field. However, the manuscript currently has multiple issues that must be addressed. This includes a need for improved technical analysis: description of statistical analysis, justification to support the validity of the selected informatics approaches in this specific context, and orthogonal validation of results. At this point, the conclusions are not supported by the data, and conceptually, beyond identifying proteins that might be involved in some of these biological processes, the data does not currently represent a major advance over what is currently known about EMT, Immune content, etc., in the field.

Selected Major points:

The observation of differential expression of splicing genes in CD8 T effectors and B cells is interesting, and raises the question of how differential splicing may impact immune cell function in this context. If the authors are able to validate that the differential expression of the splicing factors is real, then the authors should also analyze the transcriptomes of immune cells in greater depth to understand if the differential expression of the splicing factors is significant enough to induce a splicing phenotype in the immune cells. Otherwise, the data do not currently support that the differential expression is phenotypically significant.

The approach to use these specific RNA signatures to help de-convolute proteome signatures with CIBERSORTx is neither well justified in the manuscript (a correlation with TCGA RNAseq data is insufficient) nor validated with supporting orthogonal experiments. The statistics of this approach would also need deeper description, given the matrix transposition limitations of using scRNAseq data for this purpose, and the details of how the authors address this statistical limitation should also be added to the methods. The manuscript would also benefit from leveraging proteomics signatures to support the analysis, including those manuscripts cited in the introduction, refs: 16-18.

The lack of validation of the outputs of informatics analysis throughout the manuscript is unacceptable. The authors frequently perform DE analysis followed by concept enrichment, but fail to validate protein expression directly (e.g. ACTIN and myosin family changes) or functional phenotype to support concept validation (e.g. EMT phenotypes). This should be corrected throughout the manuscript.

The manuscript would significantly benefit from testing the significance of the signatures in secondary validation cohorts. The current lack of high N or validation cohorts limits the significance of the liquid biopsy analysis, as it is essentially all pilot cohort analysis at this point.

Reviewer #3, expertise in mass-spec based proteomics and biomarkers (Remarks to the Author):

This is a very nice manuscript, which is well written. The data presented consists of both new and published data, and represents a new approach to evaluate immune cells in HNSCC. However, what type of data is used where is unclear. This needs to be better described in the text, and by adding a schematic overview of all the datasets used. Furthermore, there seems to be a discrepancy of the number of samples described in the text as being measured, and the number used in the analyses and/or in the figures.

I have a few questions that needs to be addressed.

1. To me it is very unclear how many samples are what of the 59 patient cohort. It is described 27 FFPE, 27 matching LN, 27 tissue matched normal, and 27 LN matched normal (total of 27 patients), and besides this, 15 buffy and paired saliva samples. $27 + 15 = 42$ patient-cohort in total.

2. When I look at Supl. Figure 1, it is 94 samples that were measured, while according to my calculation this should be 108/121 samples (inc PBMCs). $27 * 4 = 108$, + 13 PBMC = 212 samples. Where does the discrepancy comes from?

3. The labels of Figure 1A bottom graph are very unclear and needs a legend.

4. Many external datasets are used in the current study. When reading the manuscript text, it is hard to follow which data is extracted where and what came out of the analysis. Also for the external data, it wasn't clear how many samples as an input were given.

5. " ..., as this observation indicates that further studies focusing on CADs may benefit these patients." (page 8, lines 174-175). It is unclear to which evidence this conclusion is based on.

6. Figure 1D, only 8 samples are shown. What was the input dataset? This is unclear.

7. Page 9, line 177 "Buffy coat prediction...". Using which dataset is this analysis performed? Only 15 buffy coat samples were analyzed in the current dataset, but here is referred to 16 samples).

8. Fig 1 shows a nice overview about non-malignant cells. What about the malignant cells? Or LNs? Why weren't these analyzed?

9. Page 9 line 191, 11 pN+ and 14 pN0, were used for the analysis. What samples are these? Why not all the 27 samples that were measured?

10. The supplementary tables don't have a column the fold up/downregulated.

11. Supl. Table 1-1 and 2, what is this data, RAW? Discovery? This is unclear.

12. Figure 3 A, legend shows pN+ and pNO, red and blue. What do the colors mean? Up or down regulated? This is unclear

13. Figure 3B, the input was 5 genes, How can 5 different genes separate 6 cell types?

14. Figure 4, it is unclear what type of comparison has been made in this figure.

15. Figure 4, lymph node vs primary tumor, green/red is up/down. Does that mean uniquely expressed? Why isn't the fold up/down indicated?

16. Figure 4, legend says not detected. Do the authors mean not measured instead?

17. Figure 4B, the comparison is again unclear. Upregulated in what sample type, lymph node?

18. Figure 4c, it was indicated that malignant cells in the LNs weren't measured, but here they are indicated as being measured. This is unclear to me. What are the malignant cells in the LNs in this graph?

19. Figure 5C, what does the size of the dots indicate in this graph?

20. Figure 7, why is PCR used for the measurement of the proteins, and not ELISA? We know that gene expression usually doesn't correlate well to protein expression.

21. Figure 6 and 7, were these selected proteins differentially expressed between cancer and normal samples? Why were these selected, what criteria did they fall into?

POINT-BY-POINT RESPONSE TO THE REVIEWER'S COMMENTS

We greatly appreciate the reviewers' helpful comments for improving our manuscript. The suggestions triggered a substantial set of edits in the main text and led us to effectively update most of the figures. A version of the revised manuscript with changes highlighted in grey color has been uploaded. Our point-by-point response addressing each of the reviewer's concerns is presented below.

REVIEWER COMMENTS

Reviewer #1, expertise in proteomic analysis and machine learning (Remarks to the Author):

In this study the authors have performed global proteomic analysis of HNSCC primary tumors with matched negative/positive lymph nodes, in addition to buffy coat and saliva samples. Furthermore, malignant tissue/lymph were separated into malignant and non-malignant sections for analysis. Conceptually, this should produce a valuable dataset that can help dissect HNSCC biology and metastasis, especially due to the collection of paired samples. But, in its current form, the study has major shortcomings that result in a failure to realize the potential of the data.

Response: We thank the reviewer for the comments and appreciate the thoughtful review.

MAJOR CONCERNS:

* The tissue and lymph node samples were all collected in Chile, while all the buffy coat, saliva and flow cytometry samples were collected in Brazil. This stratification of sample collection can result in the introduction of unknown biases that cannot be computationally deconvoluted. Furthermore, the number of samples is relatively small for the extent of subdivision and type of analyses performed in the manuscript.

Response: We would like to thank the reviewer for this pertinent observation. We agree that the samples were collected in two different South American countries and unknown biases could be present. However, these biases are diminished because the **risk factors, clinico-histological features and epidemiological aspects** of head and neck squamous cell carcinomas (HNSCC) among South American countries are practically unchanged. Tobacco and alcohol consumption are the high-risk factors that occur in South America, as well as in most of the world, and heavy users of both substances have a >35-fold higher risk of developing HNSCC¹⁻³. It could be different if one of the countries was India, Taiwan or some provinces of China, where the use of areca nut or betel quid products are the main risk factors that are not really present in South America⁴. HNSCC from Chilean and Brazilian populations also share an overall diagnosis in advanced stages of the disease, higher incidence in males than females, and a common immunohistochemical behavior for some frequently evaluated markers like p53, Ki67, and HOX family members, among others⁵⁻¹⁰. In fact, we can see this homogeneity between Chilean and Brazilian HNSCC cases when analyzing the clinical and pathological data from samples used in this study that are presented in **Supplementary Table 1-2** (27

Chilean patients from one center vs. 66 Brazilian patients from three centers). Eight out of the 10 clinical and histopathological features evaluated were similar between patients from Chile and Brazil ($p > 0.05$; Fisher's exact test or unpaired two-tailed Student's t-test) (**Response Figure 1A**).

Response Figure 1. Comparison of clinical and pathological features between the Chilean and Brazilian HNSCC cohorts included in this study. The distribution of characteristics between all the patients from Chile ($n = 27$) and Brazil ($n = 66$) is shown in (A), whereas the specific comparison between Chilean samples and individual Brazilian centers (ICESP, $n = 27$ or FMJ, $n = 35$) are presented in (B). p-values are presented in the upper left side of the graph. LN: lymph node. * $p \leq 0.05$.

This include the distribution of age, gender, smoking and alcohol habits (which are the main risk factors for HNSCC), lymph node metastasis (which is the main clinical feature evaluated in this study), distant metastasis, surgical margins and perineural invasion ($p > 0.05$; Fisher's exact test or unpaired two-tailed Student's t-test) (**Response Figure 1A**). Two characteristics, tumor staging and WHO histological grade, had a differential pattern between HNSCC from Chile and Brazil, but this discrepancy may reflect different profiles of HNSCC patients treated in the centers rather than being a country-dependent issue (**Response Figure 1A**) ($p \leq 0.05$; Fisher's exact test). This statement becomes clearer when we compare the clinical features from the Chilean HNSCC cohort with the two different Brazilian cohorts from ICESP and FMJ. There is not statistically difference of tumor staging and WHO between Chilean and Brazilian samples from ICESP, indicating that these two centers share a more similar profile of HNSCC patients and, thus, the features are country independent ($p > 0.05$; Fisher's exact test) (**Response Figure 1B**). Data were not compared between Chilean and patients from A.C. Camargo Hospital because only 4 samples were used from this Brazilian center. We would like to highlight that this relative heterogeneity for some features is a challenge necessary to select more robust markers since HNSCC tumors have an intrinsic diversity evidenced by different clinical behaviors for patients with the same staging, different treatment responses, and a considerable variability in clinical outcomes ¹¹. As presented in the original manuscript, we could deal with this heterogeneity and selected markers that were verified in independent HNSCC cohorts by SRM-MS, RT-qPCR and flow cytometry.

Another point raised by the reviewer concerns the sample size limitation. Even though a relative high number of samples was included in the whole study (**92 tissue samples and 90 fluid samples from a 93 HNSCC cohort**), we are aware that only **15 to 27 samples** were used for comparisons in each methodology, meaning that we must be cautious in the biological interpretation of statistical results (**Supplementary Table 1-1**). In this scenario, we increased the robustness of findings through a series of confirmatory results to give us more certainty that the molecular profiles are related to the biology of HNSCC instead of occurring by chance in discovery proteomics, RT-qPCR, PRM-MS, SRM-MS, flow cytometry and machine learning analysis, as follows:

1. **Discovery proteomics (DDA)**: this phase is often carried out using a limited number of samples due to the cost and relatively low throughput ¹². To overcome this issue, some downstream analyses were performed and indicated that true entries were detected from DDA independently of the sample size, and included:
 - (i) The enrichment of biological processes: differentially abundant proteins from the multiple sites were analyzed in the context of biological processes and several of these processes were highly overrepresented for the distinct sites, indicating the value of the proteins detected in the discovery phase (**Fig. 1D; Fig. 2B; Fig. 2D**).
 - (ii) The detection of common proteins and biological processes among sites: once the interconnection of multiple environments is necessary to support a tumoral niche ¹³, one would expect that a set of common proteins and processes are modulated among sites. Indeed, we found common processes enriched in the multiple sites for the total

and metastasis-associated proteomes (**Fig. 1D; Fig. 2B**), as well as a common signature of 23 proteins that was significantly associated with lymph node metastasis at two or more sites (**Fig. 2C**).

(iii) The expression of metastasis markers by immune populations: the differentially abundant proteins associated with metastasis were enriched for immune-associated biological processes, and this assumption was reinforced when we found that a set of these proteins is expressed by immune cells according to public databases or flow cytometry analysis (**Fig. 2E; Fig. 5A-B; Fig. 6E-F**).

Additionally, to be more confident that strong metastasis-associated biomarkers were prioritized from the discovery phase to subsequent verification, we adopted some strategies that are listed in the **Methods section “Selection of targets for ML analysis”**. Once the selected targets could be verified using independent methodologies (PRM-MS, SRM-MS, RT-qPCR, machine learning) or independent cohorts (34 independent patients from whom samples were used in SRM-MS, RT-qPCR and/or machine learning), we can say that the discovery phase revealed molecules that are biologically associated with the metastatic phenotype and, therefore, the sample size did not limit the discovery of biomarkers.

2. **PRM-MS and RT-qPCR using FFPE tissues for the verification of actin-based movement and EMT proteins:** once the results obtained by these targeted methodologies agreed with the discovery phase, we can indicate that the sample size did not add any bias to our analysis (**Fig. 3**).
3. **SRM-MS, RT-qPCR and machine learning for the analysis of immune and microenvironment markers in saliva and plasma samples:** the success of the verification methods and machine learning strategy was highlighted by the robust signatures generated with the selected proteins ($AUC > 0.9$) (**Fig. 6D**).
4. **Flow cytometry:** the results obtained for the selected proteins SRSF3 and TRA2A in myeloid cells are in line with other findings reported in the manuscript (**Fig. 6E-F**).

Altogether, by evaluating multiple sites using methods based on protein abundance and gene expression, we extensively performed several layers of confirmatory results in different cohorts, indicating the robustness of the findings.

* A lot of analyses and conclusions are drawn from correlation to RNAseq data from TCGA or elsewhere. It is well known (and the authors also show) that the expected overall correlation between gene expression and protein abundance is about 0.4-0.5, with many genes/proteins in signaling pathways prone to have lower correlation. Given this, the study's reliance on gene-protein correlation has most likely skewed results to those gene/proteins/pathways that are more correlated.

Response: As mentioned in the manuscript and by the reviewer, the correlation between mRNA and protein is about 0.4-0.5 according to reports from the literature, and some genes have lower correlation with protein abundance. As stated by the reviewer, we depicted this correlation as well as in **Supplementary Fig. 3A** for HNSSC data from TCGA and our own proteomic data to show the moderated degree of such correlation. In the case of cell type

abundance, the use of CIBERSORTx relies on a signature matrix composed of many genes per cell type and we expect that the use of several genes mitigates the effect of the lowly correlated ones. To test this, we **analyzed the correlation between protein abundance and gene expression** of the genes presented in the HNSCC signature matrix of CIBERSORTx used in this study¹⁴. This analysis was performed with data from an expression atlas of healthy tissues that generated both proteome and transcriptome data¹⁴. From the 2,973 genes present in the CIBERSORTx signature matrix, 1,840 were found in the atlas with matching protein abundance. We calculated the Spearman correlation of protein abundance with gene expression across 28 out of the 29 tissues (we excluded Testis) and crossed this information with the genes associated with the cell types (gene with the highest signature for a given cell type). Next, we observed the distribution of the correlation coefficients by cell type and noted that the vast majority of the subpopulations have median correlation scores around 0.4, with the exception of macrophages and cancer-associated fibroblasts (CAF), with scores around 0.5. These results show that the signature is overall similar across cell types and should not disfavor the estimation of a given cell type (**Supplementary Fig. 3D**). To ensure the reliability of our results, we **performed a second cross-validation** using proteomic data of 22 PBMC subpopulations from Rieckmann et al.¹⁵ to deconvolute in CIBERSORTx with the reference matrix built from scRNASeq of PBMC. We observed strikingly good associations between the expected and observed cell types (**Supplementary Figure 3E**). Of note, we were unable to find any manuscript describing these verification approaches and we included as supplementary figures since it can be useful for future works. In summary, we showed that the use of an RNA signature to estimate cell types based on protein abundance may not skew the prediction. Besides adding the verification results in **Supplementary Figures 3**, all the related sentences were adjusted in the **Results section “Protein profiles indicate specific immune phenotypes across datasets”** and **Methods section “Cell type deconvolution using CIBERSORTx”**. For fully disclosure and clarity, we also included the following sentence in the **Discussion** regarding CIBERSORTx analysis:

“Even though the use of an RNA signature to estimate cell types based on protein abundance in CIBERSORTx might not be as accurate as using RNA data, external validations showed that cell type estimation using transcript matrix for proteomics data may not skew the prediction (Supplementary Fig. 3D-E), and it can be useful for future proteomics studies.”.

* I believe the ML analysis is flawed. There are hundreds to thousands of models generated using a small number of samples (14-25) with cross validation for model evaluation and no test dataset. Under these conditions, it is highly likely that the models are overfit. When so many models are being evaluated, cross validation can only be used to choose the best model, and a separate test set is required to assess model performance. At the very least, permutation testing should be used to show that the model performance is not due to random chance.

Response: We really appreciate the reviewer’s suggestion and the opportunity of improving this point. We agree that the small number of samples limits our analysis and can contribute to overfit the models. Indeed, the results based on statistical analysis using small sample sizes can

contain high rate of Gayle positives and false negatives, and thus, substantial work is still required before a possible clinical application of these signatures, including the evaluation in larger independent cohorts together with assay optimizations. In fact, we have adopted some criteria in the original manuscript to handle the machine learning (ML) results from small sample sizes. For example, we selected not one model generated by ML, but provided an overview of possibilities and how the data behave under different conditions, being more flexible and reporting a set with many candidates. Given the cross-validation variance within the evaluation of one pair (model-type, signature), we picked the best pair signature-classifier $\leq S_i, C_j \rangle$ and equivalent pairs that statistically did not discriminate from the best one (two-paired Student's t-test, $p \geq 0.05$). After filtering out the high-performance signatures (please see **Methods section "Definition of prognostic signatures using machine learning"**), we observed in this data that a few peptides/proteins/transcripts appeared with much higher frequency, with 10 to 15% of total signatures, in candidate pairs (e.g., SRSF5_Pep1, SRSF3_RNA, and TRA2A_RNA for the buffy coat combined dataset; SRSF1_Pep1, TRA2A_Protein, and TRA2A_Pep1 for the saliva combined dataset) than others, while some appeared in just a few signatures or never were selected in combination with any classifier (e.g., CD4_Pep1, CD66b_Pep1, and CD19_Pep1 for buffy cot and CD14_Protein for saliva) (**Fig. 6C; Supplementary Table 5-4**). This data might indicate that the individual peptides/proteins/transcripts provide different contribution to the performance of the signature, but the high-performance signature is achieved when a few pair candidates are combined.

As suggested by the reviewer, **we improved the power of our analysis by running an additional permutation test**: we created 111 random models per top-1 pair signature-classifier $\leq S_i, C_j \rangle$ and equivalent pairs by shuffling classes ($pN+/pN0$) and recalculating ROC AUC. The p-value tests against the null hypothesis of features being independent from targets ($pN+/pN0$). Overall, the permutation test p-values correlate with the reported ROC AUCs, as expected, and are low, indicating the results are not by chance in the context of the studied small datasets (**Response Figure 2**). 89.1% and 85.1% of the high-performance pairs $\leq S_i, C_j \rangle$ presented in the original manuscript for buffy coat and saliva, respectively, had permutation p-value lower than 0.05 (**Response Table 1**). In the revised manuscript, we also filtered out the high-performance pairs considering permutation p-values ($p \leq 0.05$) and only signatures with $AUC \geq 0.85$ were prioritized. **Most permutation p-values from pairs with $AUC \geq 0.85$ are under 0.05** and in the **confidence interval 0.022 ± 0.0008** ($\alpha=0.0001$); the maximum p-value from pairs with $AUC \geq 0.85$ is equal to 0.098. The resulted p-values were added to the **Supplementary Table 5-4** and the reformulated Results and Methods were included in the appropriate sections. Besides the permutation results, the ML results were also strengthened by flow cytometry analysis, once they demonstrated that SRSF3 and TRA2A proteins from the top-1 signature found in blood are in fact associated with the metastatic phenotype by an independent methodology and cohort (**Fig. 6E-F**). Thus, it was an interesting way to explore the data and confirm that we are working with biologically relevant proteins connected to a good multivariate performance.

Response Figure 2. Permutation p-values for pairs $\leq S_i, C_j \rangle$ retrieved by machine learning in the original manuscript. Note that most permutation p-values from pairs with AUC ≥ 0.85 are under 0.05. These signatures were filtered out and kept in the analysis. For datasets description, please see **Supplementary Table 5-1**.

Response Table 1. High-performance signatures generated by machine learning.

Dataset ID*	Sample	Number of pairs $\leq S_i, C_j \rangle$		
		Original manuscript	Filtering permutation $p \leq 0.05$ (% from original)	Filtering permutation $p \leq 0.05$ and AUC ≥ 0.85 (% from original)
Dataset 1**	Lymph node	82	82 (100%)	82 (100%)
Dataset 2**		18	18 (100%)	18 (100%)
Dataset 3	Buffy coat	3	2 (66%)	2 (66%)
Dataset 4		11	0 (0%)	0 (0%)
Dataset 5		2	2 (100%)	2 (100%)
Dataset 6		7	5 (71%)	0 (0%)
Dataset 7		1	1 (100%)	1 (100%)
Dataset 8		113	112 (99%)	112 (99%)
Dataset 9	Saliva	5	0 (0%)	0 (0%)
Dataset 10		1	1 (100%)	1 (100%)
Dataset 11		5	0 (0%)	0 (0%)
Dataset 12		6	0 (0%)	0 (0%)
Dataset 13		1	0 (0%)	0 (0%)
Dataset 14		4106	3509 (85%)	1727 (42%)
Total number for buffy coat		137	122 (89%)	117 (85%)
Total number for saliva		4124	3510 (85%)	1728 (42%)

* For datasets description, please see **Supplementary Table 5-1**.

** Machine learning was applied just for target selection.

Finally, we understand the reviewer's concerns considering the absence of an independent test set in machine learning. Unfortunately, we could not access additional buffy coat and saliva samples from a new HNSCC cohort to quantify and use in the ML analysis. Sample collection by our collaborators was highly impacted by the pandemic that restricted the access to the hospitals, and we were not able to gather a sufficient number of samples that could benefit the ML analysis. However, for the reasons mentioned above, we consider that the models herein generated by the ML strategy were strong enough to guide us in the decision-making process for next steps with larger sample sizes. We made clearer in the revised manuscript the limitations that the sample size introduced in our overall conclusions and specifically in the machine learning analysis by reformulating the related sentences in the **Discussion** and **Methods** section "**Definition of prognostic signatures using machine learning**".

* An independent 20 HNSCC patient cohort is used to measure SRSF3 and TRA2A. The distribution of these proteins do not appear to dramatically separate the N0 and N+ populations. A better use of this independent cohort would be to generate an independent test set for validating the models.

Response: We thank the reviewer for this suggestion. In this part of the manuscript, we meant to biologically verify if the proteins present in the top-1 combined signature detected in buffy coat (SRSF3 and TRA2A proteins) are relevant in the metastatic cascade through the expression in specific immune populations. That is the reason why we used flow cytometry, which makes it possible to quantify these proteins in targeted cell populations and further give insights to HNSCC biology. We then used a univariate analysis to show that we indeed had reduced levels of the proteins in specific immune populations, and these results are in line with other information shown throughout the manuscript (**Fig. 6E-F**). We appreciate the reviewer's suggestion of using flow cytometry information as an independent cohort to test SRSF3 and TRA2A signatures from the ML. However, we believe this analysis has some limitations, mainly because the combination of SRSF3 and TRA2A levels determined by SRM-MS could not properly separate the classes in the ML models. This was evidenced by the reduced AUCs retrieved for the size-2 signatures combining SRSF3 and TRA2A peptides or proteins that were reported in the original and revised versions of the manuscript (AUC ranging from 0.26 to 0.67; **complete table deposited in Panorama repository**). That happened because SRSF3 and TRA2A proteins were considered in the best ML models only when in combination with specific peptides and proteins (**Supplementary Table 5-4**), but we do not have flow cytometry information for the other markers included in the top-1 or other buffy coat high-performance signatures. That turns the use of flow cytometry data inviable to validate the ML models. However, even though we did not have an independent test for validating the models, we consider that the models herein generated by ML are strong enough to guide us in the decision-making process for the next steps of additional investigation, such as flow cytometry experiments, and future studies with larger sample sizes as mentioned in the previous question.

* Use of statistical tests, multiple-testing correction and parameter choices are inconsistent throughout the paper. It almost appears like a lot of choices were tried till the desired result was obtained. Some illustrative (not exhaustive) examples include:

Response: We thank the reviewer for this significant observation and apologize for the lack of clarity. In addition to adjusting the points raised below, we also modified the **Methods section “Statistical analysis”** to make clearer the approach used to determine significance for each methodology. In summary, data from discovery proteomics were log₂ transformed and two-sided unpaired Student’s t-test was applied for group comparison (pN+ vs. pN0; $p \leq 0.05$). The comparison between groups in SRM-MS, PRM-MS, RT-qPCR, or flow cytometry experiments (N+ vs. N0 or lymph node vs. primary tumor; $p \leq 0.05$) was verified using the appropriate two-tailed parametric or non-parametric test selected after testing the normality of the data with the Shapiro-Wilk test (unpaired or paired Student’s t-test, Wilcoxon signed-rank test, or Mann-Whitney test; $p \leq 0.05$). The statistics for functional annotation or cell type enrichment are presented in the respective sections. We have also carefully reviewed the manuscript to make sure that the statistical parameters are clearly indicated in each Figure legend, Results, and Methods sections.

- Fig 1B: Each hierarchical clustering uses a different metric. A single metric should be used for all datasets.

Response: We would like to thank the reviewer for the opportunity of improving this point. In fact, different methods and metrics were combined and dendrograms exhibiting the most evident clustering of proteins (protein clusters PCs) were selected for further analysis, as stated in the **Methods section “Clustering global proteome datasets”**. However, the reviewer is correct that it is more appropriate to consider a single criterion for all the six datasets (malignant cells from primary tumor, non-malignant cells from primary tumor, malignant cells from lymph nodes, non-malignant cells from lymph nodes, buffy coat, and saliva cells), once we made comparisons of the GO biological processes enriched among them. Thus, we reanalyzed the PCs obtained for the different datasets and considered that dendrograms generated with the Ward’s method based on Bray-Curtis distance resulted in the best segregation patterns (PCs) for all the sites. We next reevaluated the GO biological processes enriched for the revised PCs and the immune system terms were still overrepresented in all the clusters analyzed. The related sentences and visualization of dendrograms or enriched GO biological processes were reformulated throughout the revised text (**Fig. 1C-D; Supplementary Fig. 2b; Results section “Global proteomes are collectively implicated in immune response”; Methods section “Clustering global proteome datasets”**).

We would like to highlight that patients’ clusters (Cs) have also been generated considering the most evident grouping pattern and the methods and metrics were not standardized in the first version of the manuscript (**Fig. 4A**). To follow the same assumption employed for PCs in the revised version, we reevaluated the clustering patterns and looked for common hierarchical clustering criteria that could effectively group samples for every dataset. Ward’s method based on Chebyshev distance was the best combination and could separate patients in two evident

clusters for almost all datasets, except one. Cluster 1 generated for the proteomes of malignant cells isolated from primary tumors consisted only of 3 samples (**Response Figure 3**) and we did not consider that was an acceptable segregation pattern due to the highly unbalanced number of samples between the two clusters. Thus, we decided to keep the dendrogram obtained with Complete method and Canberra metric that was presented in the original manuscript for malignant cells isolated from primary site and made this statement clear in the manuscript, whereas proteomes from all the other sites (malignant and non-malignant cells from lymph nodes, non-malignant cells from primary tumors, saliva, and buffy coat) were effectively clustered by Ward Chebyshev. Even though we changed the clustering criteria to segregate samples for some datasets, the association with clinical data showed similar results when compared to the ones presented in the original manuscript. The revised sentences and images were reformulated (**Fig. 5A; Results section “Microenvironment proteomes group samples according to metastasis and highlight candidate markers of locoregional spread”; Methods section “Clustering global proteome datasets”**).

Response Figure 3. Dendrogram obtained after grouping the proteomes of malignant cells from primary tumor samples using Ward’s method and Chebyshev distance. Note that unbalanced clusters were formed, and we kept the dendrogram built upon Complete method and Canberra metric in the revised version of manuscript.

- Nominal p-values are used in most analyses, and BH-FDR in some others. All analyses should consistently use BH-FDR p-values.

Response: We would like to thank the reviewer for the suggestion, and we agree that correcting for multiple comparisons is highly recommended in large datasets. Proteomes from the following five sites were compared between pN+ and pN0 patients and raw p-valued are presented in the manuscript instead of adjusted p-values: (i) malignant cells from primary tumor, (ii) non-malignant cells from primary tumor, (iii) non-malignant cells from lymph nodes, (iv) buffy coat cells, and (iv) saliva cells (**Fig. 2**). In fact, we tested these datasets for multiple comparisons using several procedures (Bonferroni, Benjamini-Hochberg, Holm, Hochberg, Hommel, and BY) in R environment, but only non-malignant cells isolated from lymph nodes returned proteins with significative adjusted p-values (**Response Figure 4**),

which were further explored to define the “microenvironment markers” submitted to the verification steps (Fig. 4D).

Response Figure 4. Multiple testing corrections for determining lymph node metastasis markers in proteomics data from the multiple datasets. Bonferroni, Benjamini-Hochberg, Holm, Hochberg, Hommel, and BY procedures were tested to adjust a series of p-values generated from the pN+ vs. pN0 comparisons in tissues and fluids proteomes. The proteins from each dataset significantly associated with lymph node metastasis (two-sided unpaired Student’s t-test; $p \leq 0.05$; pN+ vs. pN0) are counted in the x-axis and non-adjusted (raw) p-values are represented as grey lines. The dashed lines in y-axis show the threshold to determine an adjusted p-value as significant (adjusted $p \leq 0.05$). Note that only differential proteins of non-malignant cells from the lymph nodes had significant p-values after multiple corrections.

It was reported that the application of multiple testing correction in proteomics data still faces specific challenges, and these tests can fail to detect any true positives even when many exist¹⁶. Thus, in this manuscript we decided to consider the metastasis-associated signatures generated with raw p-values for some downstream analyses used to verify the results, as follows:

- (i) **Enrichment of biological processes:** Differentially abundant proteins from the multiple sites were analyzed in the context of biological processes and several of these processes were highly overrepresented for the distinct sites, indicating the value of the proteins associated with metastasis (Fig. 2B).

- (ii) **Detection of common proteins and biological processes among sites:** once the interconnection of multiple environments is necessary to support a tumoral niche¹³, one would expect that a set of common proteins and processes are modulated among sites due to the metastatic process. Indeed, we found a common signature of 23 proteins that was significantly associated with lymph node metastasis at two or more sites, as well as common biological processes overrepresented among sites, reinforcing that true entries were selected from statistical analysis (**Fig. 2B-D**).
- (iii) **Detection of metastasis markers expressed by immune populations:** the differentially abundant proteins associated with metastasis were enriched for immune-associated biological processes, and this assumption was reinforced when we found that a set of these proteins is potentially expressed by immune cells. Therefore, this “in silico” analysis corroborated that true values were generated from not corrected statistical analysis (**Fig. 2E; Fig. 5A-B**).

- Fig 6D uses 3 different tests, and the lowest p-value is chosen. A single test (preferably Wilcoxon) should be used.

Response: We thank the reviewer for the comment. In fact, we implemented three distinct statistical tests in the original manuscript to verify transcript differences between N+ and N0 cells: Wilcoxon, Bimodal and MAST. Following this analysis, we considered a differential expression between groups any protein altered in a given population with $p \leq 0.05$ independently of the statistical test employed. However, we agree that this was not the best approach and, following the reviewer’s suggestion, we refined the statistics and considered only the significant differences between N+ and N0 cell populations highlighted by Wilcoxon test (**Fig. 5B-C**). In the revised analysis, the differential expression presented in the original text was kept for SRSF2 and SRSF5 in dendritic cells and macrophages, respectively, while the association of SRSF2 and SRSF5 from B cells and lymphocytes with metastasis was just “on the border” and not significant using Wilcoxon. The related sentences and images were reformulated in the revised version (**Fig. 5B-C; Results section “Microenvironment and other splicing markers may be expressed by immune populations”, Methods section “scRNASeq processing and differential expression”**).

* The manuscript repeatedly makes overstated causal claims based on correlation/GO enrichment results. Some (not exhaustive) examples include:

We thank the reviewer for the comment and apologize for making overstated claims based on proteomics results. The manuscript was carefully revised to search for this kind of observation and the sentences were amended.

- "... metastasis recapitulates the content of primary tumors" (Pg 13, lines 282-3). Based on what?

Response: We thank the reviewer for pointing out this vague observation. When analyzing the frequencies of upregulated/downregulated proteins associated with the site of isolation (lymph nodes vs. primary tumor) in the malignant and non-malignant cells of HNSCC, we noticed a

lower number of deregulated proteins in the malignant portion when compared to the non-malignant, even though there was a similar total number of proteins identified and quantified on these cells (6% deregulated proteins in the malignant cells vs. 37% in the non-malignant). Thus, we stated that malignant cells from lymph nodes (metastasis) recapitulates the content of malignant primary tumors more strongly than non-malignant cells of lymph nodes represent their counterpart in primary tumors. However, we apologize for not presenting the statistics and not being clear in the short sentence included in the manuscript. We included **Fig. 3B** to illustrate the frequencies of deregulated proteins in malignant and non-malignant portions, as well as the statistical analysis, and the related sentences were reformulated in the main text (**Results section “Nodal metastasis cells resemble the molecular signature from tumors”**; **Methods section “Comparison between tumor and metastasis proteomes”**).

- "... differential protein is strongly involved in actin-based cell movement" (Pg 13, lines 283-284). Based on this GO enrichment, the authors conclude that the EMT program is activated in the cell, even though no EMT related GO terms have been enriched. The rest of page 13 (lines 292-304) is therefore unsubstantiated speculation.

Response: We agree with the reviewer that our evidence was not robust enough to conclude that there is a modulation of EMT in HNSCC and we really appreciate your comment to be able to improve our statement. In the revised version of the manuscript, a series of experiments and analysis were conducted to verify this point. In summary, the nine proteins involved in the enriched **GO biological processes of actin-based movement** (ACTA1, ACTN2, CASQ1, GSTM2, MYH7, MYL1, MYO1G, NEB, and TPM3) and three **EMT canonical markers** (E-cadherin - CDH1, fibronectin - FN1, and vimentin - VIM) were evaluated by PRM-MS and/or RT-qPCR in the 11 paired FFPE malignant cells from primary tumor and lymph nodes used in discovery proteomics. We also calculated an **EMT score** for lymph node and primary tumor samples considering a 76-gene signature reported in the literature¹⁷. In summary, with this additional analysis we validated the levels of a set of molecules involved in actin-based movement and the EMT analysis showed a more “epithelial” phenotype in lymph node samples and a more “mesenchymal” profile for primary tumors, indicating the modulation of EMT/MET processes in HNSCC spread. Detailed information and images were included in the revised version of the manuscript (**Fig. 4E-I**; **Results section “Nodal metastasis cells resemble the molecular signature from tumors”**; **Methods sections “Comparison between tumor and metastasis proteomes”, “Selection of proteotypic peptides and transitions for PRM-MS”, “PRM-MS and data analysis”**). As suggested by the reviewer, we also removed all the subsequent speculation for the malignant enrichment included in this section of the Results, and the statements about the non-malignant GO biological enrichment of translational processes were moved to the **Discussion**.

- Page 15, lines 336-7: Over-interpretation of correlation results combined with GO enrichments.

Response: We appreciate the reviewer’s recommendation. In the sentence “... thus providing an **unprecedented** view of potential processes and molecules that are implicated in tumor

invasion and spread.”, the word “unprecedented” was replaced by “detailed” (**Results section “Nodal metastasis cells resemble the molecular signature from tumors”**). The following sentences was also amended: “identifying an immune landscape” was replaced by “identifying a proteome composition”, and the word “remarkable” was removed from “**Remarkably**, malignant cells from primary tumor, buffy coat and saliva shared similar ...” (**Results section “HNSCC multi-sites exhibit immune-associated nodal metastasis markers”**)

- Some other overstatements (in my opinion) in the Discussion include (page:lines) 24:558-9, 25:580-1, 25:585-6, 26:603-4.

Response: Thanks again for the pertinent suggestions. We amended the suggested and other sentences across the **Discussion**, as follows:

- In the sentence “...however, a **breakthrough** in our study is the identification of the overrepresentation of immune processes in the multisite samples evaluated.”, the word “breakthrough” was replaced by “advance”.
- In the sentence “**Remarkably**, the immune response is coordinated across tissues, and ...”, the word “remarkably” was removed.
- In the sentence “**Remarkably**, our work revealed that pN status is **clearly** related to immunity in HNSCC by using ...”, the words “remarkably” and “clearly” were removed.
- The sentence “By deciphering the immune GO biological processes associated with metastasis markers that are commonly enriched in multiple sites, we revealed that the development of lymph node metastasis depends upon an orchestrated interconnection of immune system across multiple HNSCC environments” was adjusted to “We deciphered the immune GO biological processes associated with metastasis markers that are commonly enriched in multiple sites, thus indicating that ...”.
- The sentence “...we identified several proteins involved in the metastatic spread that can be targeted for therapy, including motility- and translation-associated molecules” was amended to “...we identified several proteins involved in the metastatic spread, including ECM-, motility-, and translation-associated molecules. It is interesting that the modulation of specific molecules has inhibited metastatic colonization^{18,19}, and the proteins associated with metastasis herein shown are potential targets to be further evaluated in animal models and considered to clinically prevent the spread of HNSCC.”.
- The sentence “Still other studies must be developed to better understand the splicing effect in RNA maturation and proteoforms, as well as functional changes.” Was included in the discussion of splicing factors.

* How are sample pairs from the same patient treated? It appears like the increased statistical power arising from the availability of multiple samples from the same patient is not fully utilized. When a paired analysis is done, it is not clear how the samples are analyzed to take sample-specific correlation into account.

Response: This is a very interesting issue, and we thank the reviewer for the comment. In the original version of the manuscript, we have analyzed 11 paired FFPE samples isolated from primary site and lymph nodes using **discovery proteomics (Fig. 3)**. A major issue of mass spectrometry-based proteomics (discovery phase) is that the power of statistical inference is greatly impacted by the presence of missing values in the protein abundance data²⁰. This issue is even more noticeable when working with FFPE samples, which is our case, due to the highly formalin crosslinked and/or degraded molecules that decrease the proteome coverage. For this reason, we did not benefit from the paired analysis due to the lack of intensities for one sample of the matched pairs, turning the paired statistics impracticable for some proteins. To prove this point, we compared the differential proteomes between primary sites and lymph nodes for the malignant and non-malignant portions using paired and unpaired two-sided Student's t-test followed by multiple testing correction (**Response Figure 5**).

Response Figure 5. Paired and unpaired Student's t-test to compare the proteomes of lymph node and primary tumor sites for malignant and non-malignant cells. The overall distribution of adjusted p-values (lymph node vs. primary tumor; q-values – Benjamini-Hochberg (BH) test) obtained for malignant and non-malignant cells is presented in (A). The intersection of differentially abundant proteins identified from paired and unpaired t-tests is represented in (B) for malignant (upper) and non-malignant (bottom) analysis. *Besides differentially abundant proteins ($q \leq 0.05$), proteins identified exclusively in one condition (lymph node or primary tumor) were also considered in the Venn diagrams.

Our data indicate that significant information was missed when performing paired analysis, especially for non-malignant cells, as evidenced (i) by the higher q-values generated for the paired t-test when compared to the unpaired statistics (**Response Figure 5A**), and (ii) because of the lower number of differential molecules when conducting the paired analysis (869 for unpaired vs. 776 for paired) (**Response Figure 5B**). Thus, the unpaired statistic was more effective in detecting differences when using large-scale data and agrees with our aim of defining the proteomic content associated with primary tumor and lymph node cells, not necessarily in a matched manner. Thus, we kept the unpaired t-test analysis in the discovery

phase presented in the manuscript and the related sentences were reformulated for clarity of the statistical analysis (**Results section “Nodal metastasis cells resemble the molecular signature from tumors”, Methods section “Comparison between tumor and metastasis proteomes”**).

We also would like to mention that we applied the same assumptions used for discovery proteomics for the **PRM-MS and RT-qPCR** data of the 11 matched FFPE samples included in the revised version of the manuscript (**Fig. 3**). Once **RT-qPCR** consists of a high number of missing data due to the low quality of FFPE RNA, we considered **unpaired tests** for statistical analysis (unpaired Student’s t-test for normally distributed and Mann-Whitney test for not normally distributed samples). For **PRM-MS**, we nearly did not have missing values for the matched pairs and the two-sided **paired tests** were used (paired Student’s t-test for normally distributed and Wilcoxon signed-rank test for not normally distributed samples).

OTHER CONCERNS/ISSUES:

* When referring to "Methods" in the body of the text, please specify which section in the Methods the reader should reference. As it stands now, it is unclear where in the methods to find the relevant information. For example, there is no section in the methods (that I could find) where identification of cluster markers (page 11, line 232) is described.

Response: Thank you for the suggestion. All the Methods sections were specified in the body of the text.

* Figure 3D - Metastasis column does not show any strong correlations, contradicting the statement made in Page 11 lines 253-4.

Response: We appreciate this pertinent assessment about the correlation analysis. In this study, the Spearman correlation coefficients (ρ) between pN outcome (pN+ and pN0) and immune signatures ranged from -0.23 to 0.12 for HNSCC samples. We agree with the reviewer that we must be cautious in interpreting these results once the ρ values may indicate a weak correlation between the parameters, even though we had significant associated p-values ($p \leq 0.05$). However, the “cutoff points” to measure the strength of the relationship are just arbitrary and it is recommended that the correlation coefficients are evaluated in the context of the analysis²¹. Indeed, additional associations between lymph node metastasis and immune system were extensively proved throughout the main text using our own proteomics data or exploring data from public databases, including (i) the enrichment of immune processes for the metastasis signatures from multiple sites (**Figs. 2B; Fig. 2D**), (ii) the presence of immune cluster markers in the metastasis signatures (**Figs. 2E**), (iii) the association of lymphocyte counts from the clinical laboratory with nodal status (**Fig. 2F**), and (iv) the association of selected “microenvironment markers” expressed by immune cells with lymph node metastasis (**Fig. 5B**). Therefore, the significant p-values associated with the correlation coefficients and a repeatedly relation between lymph node metastasis and immunity throughout the text reinforced the strength of the correlation results, even though the ρ values were not considerable high. Although we were able to strengthen this data with other findings, we agree

that they are not primordial to the main conclusions of the manuscript, and the graph was moved to supplementary material (**Supplementary Fig. 3F**).

* Figure 2C, 2D and 2E essentially restate the same information. The authors claim this as additional evidence (page 10, lines 222-24).

Response: We thank the reviewer for pointing this out and apologize for the lack of clarity in the interpretation of **Fig. 2**. First, we would like to highlight that, as suggested by this reviewer, **Fig. 2B** from the original manuscript was removed because the same information was included in the **Supplementary Table 2**. The remaining figures (**Figs. 2B, 2C and 2D** in the new version) in fact indicate different information, even though they represent results generated from the same datasets. In **Fig. 2B**, we presented the **biological processes significantly enriched** ($FDR \leq 0.05$) for the metastasis-associated signatures identified across the five distinct sites (pN+ vs. pN0; two-sided unpaired Student's t-test; $p \leq 0.05$) that included 110 proteins for the malignant cells from primary tumor, 85 proteins for the non-malignant cells from primary tumor, 201 proteins for the non-malignant cells from lymph node, 80 proteins for buffy coat, and 54 proteins in saliva. Considering that the interconnection of multiple environments is necessary to support the tumoral niche¹³, we next used these sets of differentially abundant proteins to search for molecules commonly associated with metastasis across the five environments. The **23 proteins significantly related to nodal metastasis in two or more sites** are presented in **Fig. 2C**, and the **biological processes enriched** for this group of 23 proteins is shown in **Fig. 2D**. We have updated the manuscript to reflect this point more clearly (**Results section "HNSCC multisites exhibit immune-associated nodal metastasis markers"**).

* PCA analysis (Pg 16) is just replicating results observed in Fig 5A since only differential proteins are used.

Response: We thank the reviewer for the opportunity of clarifying this point and we apologize for the difficulty in understanding PCA and dendrograms analysis from **Fig. 5A** (**Fig. 4** in the new version of the manuscript). In fact, hierarchical clustering dendrograms (**Fig. 4A**) and PCA plots (**Fig. 4C**) have been generated with distinct datasets: (i) **global proteomes** were used as input for the grouping unsupervised hierarchical clustering analysis (**Fig. 4A**) (2,451; 1,984; 2,137; 2,188; and 1,154 proteins, respectively, identified across 25 samples of malignant cells from the primary tumor, 27 samples of non-malignant cells adjacent to the primary tumor and lymph node, and 24 buffy coat and saliva samples) and (ii) the **differential proteomes** between pN+ and pN0 samples were considered for the PCA analysis conducted for the multiple sites (**Fig. 4C**) (pN+ vs. pN0; $p \leq 0.05$; two-tailed unpaired Student's t-test or proteins detected exclusively in one group; 201, 110, 85, 80, and 54 differentially abundant proteins from non-malignant cells from lymph nodes, malignant cells from primary tumor, non-malignant cells from primary tumor, buffy coat, and saliva, respectively). We have amended the sentence referring to hierarchical clustering in the **Results** to make it clear that global proteomes instead of differential profiles were considered in this analysis (**Section "Microenvironment**

proteomes group samples according to metastasis and highlight candidate markers of locoregional spread”).

* Some figures take up too much space for the information they present (eg. Fig 2B, Fig 4B, etc.) These should be either eliminated or moved to the supplement.

Response: We agree with the reviewer that some images looked oversized and would like to thank you for the comment. As suggested, we performed an extensive review of the figures and most of them were improved in the new version of the manuscript, as follows (figure numbers are provided as in the revised version):

- **Fig. 1:** The heat map from **Fig. 1C** (**Fig. 1D** in the new version) was compacted.
- **Fig. 2:** The graphs evidencing the frequency of up- and downregulated (**Fig. 2B** in the original manuscript) were removed and the information is presented in **Supplementary Table 2-1 to 2-5**. We also reduced the images from (A), (B), and (C), and combined **Fig. 2** and **Fig. 3** of the original version as this **Fig. 2** in the revised manuscript.
- **Fig 3:** This was **Fig. 4** in the original article and the figure was completely amended in the revised version of the manuscript. Overall, the enrichment graphs of **Figs. 4A-B** (old version) were summarized in **Fig. 3D** and the additional data were included as **Supplementary Table 3**.
- **Fig. 5:** This was **Fig. 6** in the original article and the figure was completely amended in the revised version of the manuscript as well. **Figs. 6A-B** (old version) were moved to the supplement and **Fig. 6C** was reduced.
- **Fig. 6:** This was **Fig. 7** in the original article. **Figs. 7D-E** (old version) were moved to **Supplementary Fig. 7**, and **Figs. 7B-C** (old version) were replaced by **Figs. 6B-C** in the revised version.

* Title: Do the authors mean "Connecting multiple microenvironment proteomes ..."? "Wiring" is an unusual word to use in this context.

Response: We greatly appreciate the advice in helping us to improve the title of the manuscript. We agree with the reviewer that “Connecting” is a more traditional word that reflects the idea of the manuscript. The word “Wiring” has been replaced by “Connecting”.

References Reviewer #1

- 1 Chamoli, A. et al. Overview of oral cavity squamous cell carcinoma: Risk factors, mechanisms, and diagnostics. *Oral Oncol* 121, 105451, doi:10.1016/j.oraloncology.2021.105451 (2021).
- 2 Blot, W. J. et al. Smoking and drinking in relation to oral and pharyngeal cancer. *Cancer Res* 48, 3282-3287 (1988).
- 3 Johnson, D. E. et al. Head and neck squamous cell carcinoma. *Nat Rev Dis Primers* 6, 92, doi:10.1038/s41572-020-00224-3 (2020).
- 4 Gupta, S., Gupta, R., Sinha, D. N. & Mehrotra, R. Relationship between type of smokeless tobacco & risk of cancer: A systematic review. *Indian J Med Res* 148, 56-76, doi:10.4103/ijmr.IJMR_2023_17 (2018).

- 5 Perdomo, S., Martin Roa, G., Brennan, P., Forman, D. & Sierra, M. S. Head and neck cancer burden and preventive measures in Central and South America. *Cancer Epidemiol* 44 Suppl 1, S43-S52, doi:10.1016/j.canep.2016.03.012 (2016).
- 6 Bitu, C. C. et al. HOXA1 is overexpressed in oral squamous cell carcinomas and its expression is correlated with poor prognosis. *BMC Cancer* 12, 146, doi:10.1186/1471-2407-12-146 (2012).
- 7 De Souza Setubal Destro, M. F. et al. Overexpression of HOXB7 homeobox gene in oral cancer induces cellular proliferation and is associated with poor prognosis. *Int J Oncol* 36, 141-149 (2010).
- 8 BÓRQUEZ M, P., CAPDEVILLE F, F., MADRID M, A., VELOSO O, M. & CÁRCAMO P, M. Sobrevida global y por estadios de 137 pacientes con cáncer intraoral: Experiencia del Instituto Nacional del Cáncer. *Revista chilena de cirugía* 63, 351-355 (2011).
- 9 Rodrigues, P. C. et al. Clinicopathological prognostic factors of oral tongue squamous cell carcinoma: a retrospective study of 202 cases. *Int J Oral Maxillofac Surg* 43, 795-801, doi:10.1016/j.ijom.2014.01.014 (2014).
- 10 Rivera, C. et al. Clinicopathological and immunohistochemical evaluation of oral and oropharyngeal squamous cell carcinoma in Chilean population. *Int J Clin Exp Pathol* 7, 5968-5977 (2014).
- 11 Carnielli, C. M. et al. Combining discovery and targeted proteomics reveals a prognostic signature in oral cancer. *Nat Commun* 9, 3598, doi:10.1038/s41467-018-05696-2 (2018).
- 12 Nakayasu, E. S. et al. Tutorial: best practices and considerations for mass-spectrometry-based protein biomarker discovery and validation. *Nat Protoc* 16, 3737-3760, doi:10.1038/s41596-021-00566-6 (2021).
- 13 Hiam-Galvez, K. J., Allen, B. M. & Spitzer, M. H. Systemic immunity in cancer. *Nat Rev Cancer*, doi:10.1038/s41568-021-00347-z (2021).
- 14 Puram, S. V. et al. Single-Cell Transcriptomic Analysis of Primary and Metastatic Tumor Ecosystems in Head and Neck Cancer. *Cell* 171, 1611-1624.e1624, doi:10.1016/j.cell.2017.10.044 (2017).
- 15 Rieckmann, J. C. et al. Social network architecture of human immune cells unveiled by quantitative proteomics. *Nat Immunol* 18, 583-593, doi:10.1038/ni.3693 (2017).
- 16 Pascovici, D., Handler, D. C., Wu, J. X. & Haynes, P. A. Multiple testing corrections in quantitative proteomics: A useful but blunt tool. *Proteomics* 16, 2448-2453, doi:10.1002/pmic.201600044 (2016).
- 17 Byers, L. A. et al. An epithelial-mesenchymal transition gene signature predicts resistance to EGFR and PI3K inhibitors and identifies Axl as a therapeutic target for overcoming EGFR inhibitor resistance. *Clin Cancer Res* 19, 279-290, doi:10.1158/1078-0432.CCR-12-1558 (2013).
- 18 Owyong, M. et al. MMP9 modulates the metastatic cascade and immune landscape for breast cancer anti-metastatic therapy. *Life Sci Alliance* 2, doi:10.26508/lsa.201800226 (2019).
- 19 Gui, J. et al. Activation of p38 α stress-activated protein kinase drives the formation of the pre-metastatic niche in the lungs. *Nat Cancer* 1, 603-619, doi:10.1038/s43018-020-0064-0 (2020).
- 20 Lazar, C., Gatto, L., Ferro, M., Bruley, C. & Burger, T. Accounting for the Multiple Natures of Missing Values in Label-Free Quantitative Proteomics Data Sets to Compare Imputation Strategies. *J Proteome Res* 15, 1116-1125, doi:10.1021/acs.jproteome.5b00981 (2016).
- 21 Akoglu, H. User's guide to correlation coefficients. *Turk J Emerg Med* 18, 91-93, doi:10.1016/j.tjem.2018.08.001 (2018).

Reviewer #2, expertise in head and neck cancer and TME (Remarks to the Author):

The authors use proteomic approaches to profile multiple types of biospecimens from a cohort of 59 HNSCC patients and use the data to gain insight into HNSCC biology. This represents a large and fairly unique data set for the field. However, the manuscript currently has multiple issues that must be addressed. This includes a need for improved technical analysis: description of statistical analysis, justification to support the validity of the selected informatics approaches in this specific context, and orthogonal validation of results. At this point, the conclusions are not supported by the data, and conceptually, beyond identifying proteins that might be involved in some of these biological processes, the data does not currently represent a major advance over what is currently known about EMT, Immune content, etc., in the field.

Response: We would like to thank the reviewer for raising several issues critical to the improvement of our manuscript. We apologize for not being clear in the description of statistical analysis and the **Methods section “Statistical analysis”** has been modified to make clearer the approaches used to determine significance in each experiment. In summary, data from discovery proteomics were log₂ transformed and two-sided unpaired Student’s t-test was applied for group comparison (pN+ vs. pN0; $p \leq 0.05$). The comparison between groups in SRM-MS, PRM-MS, RT-qPCR, or flow cytometry experiments (N+ vs. N0 or lymph node vs. primary tumor; $p \leq 0.05$) was achieved using the appropriate two-tailed parametric or non-parametric test selected after testing the normality of the data with the Shapiro-Wilk test (unpaired or paired Student’s t-test, Wilcoxon signed-rank test, or Mann-Whitney test; $p \leq 0.05$). The statistics for functional annotation or cell type enrichment are presented in the respective sections. Statistical significance was established at $p \leq 0.05$. We have also carefully reviewed the manuscript to make sure that the statistical parameters are clearly indicated in the figure legends, results, and methods sections.

We agree with the reviewer that some points presented in the manuscript reinforce the current knowledge of the field for HNSCC. However, our data also presented valuable novelties to the scientific community that would further benefit HNSCC patients in a clinical context. We believe the two main advances of our research were:

- (i) The evaluation of multiple sites that provided an overview of the wired tumoral niches by analyzing the proteomic composition and enriched biological processes commonly modulated across tumors, lymph nodes, blood, and saliva microenvironments. Even though HNSCC samples have been tested in previous studies using large scale methodologies¹⁻³, these manuscripts evaluated only the transcript levels of primary tumors and/or lymph nodes^{1,3} or did not include saliva and/or blood samples¹⁻³. Additionally, to the best of our knowledge, none of the large-scale studies presented in the literature for HNSCC evaluated lymph node samples from pN0 patients and that is a valuable information that can be further explored by the scientific community. That means that we herein conducted the first study deciphering the connected proteomic landscape of HNSCC microenvironments both in global and metastasis-

dependent manners, thus providing valuable biological information that may be evaluated as therapeutic, predictive, or prognostic biomarkers.

- (ii) The multisite analysis led us to develop a multiparametric machine learning model that allowed for the indication of high-performance signatures in saliva or blood that can stratify pN+ and pN0 HNSCC patients. These signatures are promising since AUCs > 0.9 resulted from these models and would greatly benefit HNSCC patients considering that no prognostic markers have so far been accepted for routine use. We agree with the reviewer that there are some limitations in machine learning analysis mainly due to the low sample size that made the separation of a validation cohort unfeasible. However, we applied some additional filters and performed confirmatory flow cytometry analysis that assured the validity of the signatures generated by machine learning (please see the answer below).

Selected Major points:

The observation of differential expression of splicing genes in CD8 T effectors and B cells is interesting, and raises the question of how differential splicing may impact immune cell function in this context. If the authors are able to validate that the differential expression of the splicing factors is real, then the authors should also analyze the transcriptomes of immune cells in greater depth to understand if the differential expression of the splicing factors is significant enough to induce a splicing phenotype in the immune cells. Otherwise, the data do not currently support that the differential expression is phenotypically significant.

Response: We appreciate the comments of the reviewer and the amazing suggestion. Indeed, the induction of a splicing phenotype in immune cells due to the metastatic condition would be of great significance to prove their functional association in HNSCC. We are aware that the identification of splicing isoforms and detection of differential splicing isoforms between pN0 and pN+ conditions might be the ideal way to prove such a concept. Unfortunately, the nature and design of RNASeq libraries do not favor this analysis. Because about 70% human genome territories undergo transcription^{4,5}, most of the novel junctions are possibly by-products of adjacent transcription, which may be quickly degraded and not translated. Moreover, some novel junctions might be false positives introduced by short reads from RNASeq and most of these novel junctions have lower read coverage than the known ones. Thus, detecting the induction of a splicing phenotype in immune cells is far from ideal with RNASeq data. We also envisioned to search for the splicing phenotype in our bulk proteomics data, even though we could not have the information in an immune cell type-specific manner. However, in our proteomics approach, named bottom-up, the proteins are digested into peptides and thus information about sequence variations within a given proteoform is incomplete, turning the evaluation of the splicing phenotype not accurate.

A different approach would be to expand our analysis and evaluate the differential expression patterns of a larger set of splicing genes to finally provide a deeper understanding of the modulation of alternative splicing in the immune cells. We followed this path and retrieved a list of 421 gene ontology (GO) splicing genes (<https://www.uniprot.org>) to verify their

expression in immune cells using scRNASeq data from HNSCC primary tumors ¹, as performed for the splicing markers SRSF1, SRSF2, SRSF3, SRSF5, and TRA2A in the original version of this manuscript. From the 421 splicing factors, 41 were significantly down- or upregulated in immune cells from pN+ when compared to pN0, most of them significantly reduced in pN+ immune cells ($p = 0.021$; Fisher's exact test) (**Fig. 5C-D**). In parallel, we searched the 421 splicing proteins in our bulk proteome datasets from the multisites to better understand the pattern of splicing across HNSCC non-malignant cells and fluids. Even though we just considered sites that contain immune cells, we unfortunately could not separate the splicing modulation at single-cell level because these are bulk data. A group of 21 GO splicing proteins was associated with the metastatic phenotype across the multisites and, again, downregulated splicing factors prevailed in the pN+ phenotypes ($p = 0.0008$; Fisher's exact test) (**Fig. 5E-F**). Altogether, we had 61 non-redundant splicing factors differentially expressed between pN+ and pN0 when considering single cell RNA and proteome data that can target 41 unique genes according to SpliceAidF database. Two of these targets (HBB and TRA2B) were associated with metastasis in the proteome of non-malignant cells isolated from lymph nodes with downregulation in the pN+ phenotype.

Considering all of these, even though we could not determine a splicing phenotype associated with metastasis due to the limitations of the methodologies, we extended our findings and showed a greater group of splicing factors that is modulated in isolated immune cells or bulk subpopulations in relation to the metastatic phenotype, as well as their targeted genes, still reinforcing that splicing in microenvironment cells may significantly impact the metastatic phenotype. That was a significant result and we included it in the appropriate sections of the revised manuscript (**Fig. 5C-F, Supplementary Table 6, results section "Microenvironment and other splicing markers may be expressed by immune populations", discussion, and methods section "Searching for GO splicing factors related to metastasis"**)

The approach to use these specific RNA signatures to help de-convolute proteome signatures with CIBERSORTx is neither well justified in the manuscript (a correlation with TCGA RNAseq data is insufficient) nor validated with supporting orthogonal experiments. The statistics of this approach would also need deeper description, given the matrix transposition limitations of using scRNAseq data for this purpose, and the details of how the authors address this statistical limitation should also be added to the methods. The manuscript would also benefit from leveraging proteomics signatures to support the analysis, including those manuscripts cited in the introduction, refs: 16-18.

Response: We thank the reviewer for raising the above points and we understand the concern of using RNA signatures to deconvolute protein abundance. To support that the prediction with CIBERSORTx RNAseq matrix would not add any bias to our analysis, we performed two additional verifications that are detailed below.

- (i) First, we verified gene vs. protein correlations considering only genes that have been selected as important in the definition of cell types according to the scRNASeq data input. Thus, we verified the correlation between protein abundance and gene

expression of the genes included in the HNSCC tissue signature matrix of CIBERSORTx used in this manuscript ¹. For that, we used the results from an expression atlas of healthy tissues that generated both proteome and transcriptome data ⁶. From the 2,973 genes present in the CIBERSORTx signature matrix, 1,840 were found in the atlas with matching gene and protein expression. We calculated the Spearman correlation of protein abundance with gene expression across 28 out of the 29 tissues (we excluded testis) and crossed this information with the genes associated with the cell types (gene with the highest signature for a given cell type). Next, we observed the distribution of the correlation coefficients by cell type and noted that the vast majority of cell types have median correlation scores around 0.4, with the exception of macrophages and cancer-associated fibroblasts (CAF), with scores around 0.5, showing that the signature is overall similar across cell types and should not disfavor the estimation of a given cell type (**Supplementary Figure 3D**).

- (ii) To ensure the reliability of our results, we performed a second cross-validation using proteomic data of 22 PBMC subpopulations from Rieckmann et al. ⁷ to deconvolute with the reference matrix built from scRNASeq of PBMC. We observed strikingly good associations between the expected and observed cell types (**Supplementary Figure 3E**). Of note, we were unable to find any manuscript describing these verification approaches and we included as supplementary figures since it can be useful for future works.

Regarding the strategy that was suggested by the reviewer of using other proteomics signatures for deconvolution, we did not pursue this path because CIBERSORTx provide sample composition by leveraging cell type expression signatures from single-cell experiments or sorted cell subsets ⁸, and that is not the case of the referenced studies ^{2,9} or any other that we found in the literature and that could be used as matrix to benefit the prediction for HNSCC samples. Nevertheless, by the verifications herein presented, we showed that cell type estimation using CIBERSORTx for proteomics data may not skew the prediction. Besides adding the verification results as **Supplementary Figures 3D and 3E**, all the related sentences were adjusted in the **results section “Protein profiles indicate specific immune phenotypes across datasets”**. The **methods section “Cell type deconvolution using CIBERSORTx”** has also been amended for a better description of the CIBERSORTx analysis, and for fully disclosure and clarity, we included the following sentence in the **Discussion**:

“Even though the use of an RNA signature to estimate cell types based on protein abundance in CIBERSORTx might not be as accurate as using RNA data, external validations showed that cell type estimation using transcript matrix for proteomics data may not skew the prediction (Supplementary Fig. 3D-E), and it can be useful for future proteomics studies.”

The lack of validation of the outputs of informatics analysis throughout the manuscript is unacceptable. The authors frequently perform DE analysis followed by concept enrichment, but fail to validate protein expression directly (e.g. ACTIN and myosin family changes) or

functional phenotype to support concept validation (e.g. EMT phenotypes). This should be corrected throughout the manuscript.

Response: We appreciate the reviewer's comment and agree that our evidence was not robust enough without a validation step. Thus, we conducted a series of experiments and analysis to verify the abundance of the **nine proteins involved in the enriched GO biological processes of actin-based movement** (ACTA1, ACTN2, CASQ1, GSTM2, MYH7, MYL1, MYO1G, NEB, and TPM3) and **three EMT canonical markers** (E-cadherin - CDH1, fibronectin - FN1, and vimentin - VIM). These molecules were evaluated using PRM-MS and/or RT-qPCR in the 11 paired FFPE malignant cells from primary tumor and lymph nodes used in discovery proteomics. We also calculated an EMT score for lymph node and primary tumor samples considering a 76-gene signature reported in the literature ¹⁰. In summary, we validated the levels of a set of molecules involved in actin-based movement and the EMT analysis showed a more "epithelial" phenotype in lymph node samples and a more "mesenchymal" profile for primary tumors, which indicate a modulation of EMT/MET processes in HNSCC spread. Detailed information and images were included in the revised version of the manuscript (**Fig. 4E-I; Results section "Nodal metastasis cells resemble the molecular signature from tumors"; Methods sections "Comparison between tumor and metastasis proteomes", "Selection of proteotypic peptides and transitions for PRM-MS", "PRM-MS and data analysis"**).

The manuscript would significantly benefit from testing the significance of the signatures in secondary validation cohorts. The current lack of high N or validation cohorts limits the significance of the liquid biopsy analysis, as it is essentially all pilot cohort analysis at this point.

Response: We really appreciate the reviewer's suggestion and the opportunity of improving this point. We agree that the small number of samples limits our analysis and can contribute to overfit the models. Indeed, the results based on statistical analysis using small sample sizes can contain high rate of false positives and false negatives, and thus, substantial work is still required before a possible clinical application of these signatures, including the evaluation in larger independent cohorts together with assay optimizations. In fact, we have adopted some criteria in the original manuscript to handle the machine learning (ML) results from small sample sizes. For example, we selected not one model generated by ML, but provided an overview of possibilities and how the data behave under different conditions, being more flexible and reporting a set with many candidates. Given the cross-validation variance within the evaluation of one pair (model-type, signature), we picked the best pair signature-classifier $\leq S_i, C_j$ and equivalent pairs that statistically did not discriminate from the best one (two-paired Student's t-test, $p \geq 0.05$). After filtering out the high-performance signatures (please see **Methods section "Definition of prognostic signatures using machine learning"**), we observed in this data that a few peptides/proteins/transcripts appeared with much higher frequency, with 10 to 15% of total signatures, in candidate pairs (e.g., SRSF5_Pep1, SRSF3_RNA, and TRA2A_RNA for the buffy coat combined dataset; SRSF1_Pep1, TRA2A_Protein, and TRA2A_Pep1 for the saliva combined dataset) than others, while some

appeared in just a few signatures or never were selected in combination with any classifier (e.g., CD4_Pep1, CD66b_Pep1, and CD19_Pep1 for buffy cot and CD14_Protein for saliva) (**Fig. 6C; Supplementary Table 5-4**). This data might indicate that the individual peptides/proteins/transcripts provide different contribution to the performance of the signature, but the high-performance signature is achieved when a few pair candidates are combined.

As suggested by the reviewer, we improved the power of our analysis by running an additional permutation test: we created 111 random models per top-1 pair signature-classifier $\leq S_i, C_j$ and equivalent pairs by shuffling classes (pN+/pN0) and recalculating ROC AUC. The p-value tests against the null hypothesis of features being independent from targets (pN+/pN0). Overall, the permutation test p-values correlate with the reported ROC AUCs, as expected, and are low, indicating the results are not by chance in the context of the studied small datasets (**Response Figure 1**). 89.1% and 85.1% of the high-performance pairs $\leq S_i, C_j$ presented in the original manuscript for buffy coat and saliva, respectively, had permutation p-value lower than 0.05 (**Response Table 1**). In the revised manuscript, we also filtered out the high-performance pairs considering permutation p-values ($p \leq 0.05$) and only signatures with $AUC \geq 0.85$ were prioritized. **Most permutation p-values from pairs with $AUC \geq 0.85$ are under 0.05** and in the **confidence interval 0.022 ± 0.0008** ($\alpha=0.0001$); the maximum p-value from pairs with $AUC \geq 0.85$ is equal to 0.098. The resulted p-values were added to the **Supplementary Table 5-4** and the reformulated Results and Methods were included in the appropriate sections. Besides the permutation results, the ML results were also strengthened by flow cytometry analysis, once they demonstrated that SRSF3 and TRA2A proteins from the top-1 signature found in blood are in fact associated with the metastatic phenotype by an independent methodology and cohort (**Fig. 6E-F**). Thus, it was an interesting way to explore the data and confirm that we are working with biologically relevant proteins connected to a good multivariate performance.

Finally, we understand the reviewer's concerns considering the absence of an independent test set in machine learning. Unfortunately, we could not access additional buffy coat and saliva samples from a new HNSCC cohort to quantify and use in the machine learning analysis. Sample collection by our collaborators was highly impacted by the pandemic that restricted the access to the hospitals, and we were not able to gather a sufficient number of samples that could benefit the ML analysis. However, for the reasons mentioned above, we consider that the models herein generated by the ML strategy were strong enough to guide us in the decision-making process for next steps with larger sample sizes. We made clearer in the revised manuscript the limitations that the sample size introduced in our overall conclusions and specifically in the machine learning analysis by reformulating the related sentences in the **Discussion** and **Methods** section "**Definition of prognostic signatures using machine learning**".

Response Figure 1. Permutation p-values for pairs $\leq S_i, C_j \rangle$ retrieved by machine learning in the original manuscript. Note that most permutation p-values from pairs with AUC ≥ 0.85 are under 0.05. These signatures were filtered out and kept in the analysis. For datasets description, please see **Supplementary Table 5-1**.

Response Table 1. High-performance signatures generated by machine learning.

Dataset ID*	Sample	Number of pairs $\leq S_i, C_j \rangle$		
		Original manuscript	Filtering permutation $p \leq 0.05$ (% from original)	Filtering permutation $p \leq 0.05$ and AUC ≥ 0.85 (% from original)
Dataset 1**	Lymph node	82	82 (100%)	82 (100%)
Dataset 2**		18	18 (100%)	18 (100%)
Dataset 3	Buffy coat	3	2 (66%)	2 (66%)
Dataset 4		11	0 (0%)	0 (0%)
Dataset 5		2	2 (100%)	2 (100%)
Dataset 6		7	5 (71%)	0 (0%)
Dataset 7		1	1 (100%)	1 (100%)
Dataset 8		113	112 (99%)	112 (99%)
Dataset 9	Saliva	5	0 (0%)	0 (0%)
Dataset 10		1	1 (100%)	1 (100%)
Dataset 11		5	0 (0%)	0 (0%)
Dataset 12		6	0 (0%)	0 (0%)
Dataset 13		1	0 (0%)	0 (0%)
Dataset 14		4106	3509 (85%)	1727 (42%)
Total number for buffy coat		137	122 (89%)	117 (85%)
Total number for saliva		4124	3510 (85%)	1728 (42%)

* For datasets description, please see **Supplementary Table 5-1**.

** Machine learning was applied just for target selection.

References Reviewer #2

- 1 Puram, S. V. *et al.* Single-Cell Transcriptomic Analysis of Primary and Metastatic Tumor Ecosystems in Head and Neck Cancer. *Cell* **171**, 1611-1624.e1624, doi:10.1016/j.cell.2017.10.044 (2017).
- 2 Huang, C. *et al.* Proteogenomic insights into the biology and treatment of HPV-negative head and neck squamous cell carcinoma. *Cancer Cell* **39**, 361-379.e316, doi:10.1016/j.ccell.2020.12.007 (2021).
- 3 Cillo, A. R. *et al.* Immune Landscape of Viral- and Carcinogen-Driven Head and Neck Cancer. *Immunity* **52**, 183-199.e189, doi:10.1016/j.immuni.2019.11.014 (2020).
- 4 Wang, X., Liu, Q. & Zhang, B. Leveraging the complementary nature of RNA-Seq and shotgun proteomics data. *Proteomics* **14**, 2676-2687, doi:10.1002/pmic.201400184 (2014).
- 5 Djebali, S. *et al.* Landscape of transcription in human cells. *Nature* **489**, 101-108, doi:10.1038/nature11233 (2012).
- 6 Wang, D. *et al.* A deep proteome and transcriptome abundance atlas of 29 healthy human tissues. *Mol Syst Biol* **15**, e8503, doi:10.15252/msb.20188503 (2019).
- 7 Rieckmann, J. C. *et al.* Social network architecture of human immune cells unveiled by quantitative proteomics. *Nat Immunol* **18**, 583-593, doi:10.1038/ni.3693 (2017).
- 8 Newman, A. M. *et al.* Determining cell type abundance and expression from bulk tissues with digital cytometry. *Nat Biotechnol* **37**, 773-782, doi:10.1038/s41587-019-0114-2 (2019).
- 9 Sinha, A. *et al.* The Proteogenomic Landscape of Curable Prostate Cancer. *Cancer Cell* **35**, 414-427.e416, doi:10.1016/j.ccell.2019.02.005 (2019).
- 10 Byers, L. A. *et al.* An epithelial-mesenchymal transition gene signature predicts resistance to EGFR and PI3K inhibitors and identifies Axl as a therapeutic target for overcoming EGFR inhibitor resistance. *Clin Cancer Res* **19**, 279-290, doi:10.1158/1078-0432.CCR-12-1558 (2013).

Reviewer #3, expertise in mass-spec based proteomics and biomarkers (Remarks to the Author):

This is a very nice manuscript, which is well written. The data presented consists of both new and published data, and represents a new approach to evaluate immune cells in HNSCC. However, what type of data is used where is unclear. This needs to be better described in the text, and by adding a schematic overview of all the datasets used. Furthermore, there seems to be an discrepancy of the number of samples described in the text as being measured, and the number used in the analyses and/or in the figures.

Response: We would like to thank the reviewer for the positive comments and suggestions. We apologize for our negligence in reporting the number of samples throughout the manuscript. The text has been extensively revised to fix this issue and **Supplementary Table 1-1** has been significantly improved to provide a detailed description of the HNSCC cohorts included in this study and analyzed by different methodologies (discovery proteomics (DDA), SRM-MS, PRM-MS, RT-qPCR, flow cytometry, and machine learning). As suggested by the reviewer, we have designed a schematic overview of all the samples herein included showing

the overlap among sites or methodologies. This image is presented as **Response Figure 1**, and we can further include as a supplementary figure in the main text whether the reviewer consider this information is useful for clarifying the number of samples. It is confusing because we used samples from different sites that came from the same patients, or the same samples overlap across two or more methodologies, and we tried to clarify this point by adding detailed information in **Supplementary Table 1-2** and **Response Figure 1**. The **Methods section “Patients and sampling”** has also been amended for a full clarity about the samples used and we presented the sample sizes used in each methodology throughout the **Results, Methods, and Figure legends**. In total, we used in this study 182 samples collected from a 93 HNSCC-patient cohort (after excluding 2 samples in the quality control of discovery proteomics).

Response Figure 1. Schematic representation of samples and methodologies used in this study. (A) Total number of FFPE tissue- and fluid-samples isolated from 93 HNSCC patients included in the study. After excluding two samples due to poor quality, 182 samples were herein analyzed by DDA, SRM-MS, PRM-MS, RT-qPCR, flow cytometry and machine learning. The connecting lines indicate the number of samples originated from the same patient isolated from the two connected sites (e.g., 25 primary tumor – malignant and primary tumor – non-malignant were isolated from the same patients). The colors represent the methodologies included in the study. (B) Methodologies employed to analyze the HNSCC samples. The number of samples from distinct sites (colors) evaluated by each method is shown inside the rectangles and the connecting lines indicate samples shared across techniques. HNSCC: head and neck squamous cell carcinoma.

I have a few questions that needs to be addressed.

1. To me it is very unclear how many samples are what of the 59 patient cohort. It is described 27 FFPE, 27 matching LN, 27 tissue matched normal, and 27 LN matched normal (total of 27

patients), and besides this, 15 buffy and paired saliva samples. $27 + 15 = 42$ patient-cohort in total.

Response: We thank the reviewer for the comment, and we totally agree that this information requires better description. Considering the **discovery phase** that was herein questioned by the reviewer, we used 27 FFPE samples from malignant cells isolated from primary tumor (we excluded two samples without enough quality, so we had 25 samples for final analysis), 13 malignant cells microdissected from the lymph nodes, 27 non-malignant cells isolated from the primary tumors, 27 non-malignant cells from the lymph nodes, 24 buffy coat samples, and 24 saliva samples, totalizing **140 samples monitored that had appropriate quality** (92 from tissues and 48 from fluids), as shown in **Supplementary Figure 2** and in the related **Methods sections**. Because in this phase we used samples collected in different sites from the same patient, the **140 samples in fact came from a 59-HNSCC cohort** (27 patients for tissues and 32 patients for fluids) (**Supplementary Table 1-2**). The reviewer also mentioned the following sentence from the original **Results section** “**Global proteomes are collectively implicated in immune response**”: “This group of samples included 27 matched formalin-fixed paraffin-embedded (FFPE) tumors and lymph node tissues and also 15 paired buffy coat and saliva samples”. Here we referred only to the paired samples used in the discovery phase, i. e., samples isolated from the same patients, and this included 27 primary tumors and lymph nodes dissected from the same patients and 15 saliva and buffy coat samples from the same cohort (please see **Supplementary Table 1-2**). But we agree that this sentence is difficult to understand, and we have removed it from the final version of the manuscript.

2. When I look at Supl. Figure 1, it is 94 samples that were measured, while according to my calculation this should be 108/121 samples (inc PBMCs). $27 * 4 = 108$, + 13 PBMC = 212 samples. Where does the discrepancy comes from?

Response: We would like to thank the reviewer for this comment and once again we agree that sample sizes must be better described in the text and figures. **Supplementary Figure 1** represents the quality control for every DDA run (iRT and trypsin peptides, described in the **Methods section** “**Quality control in discovery proteomics**”). In (A) and in the first graph of (B), it is shown the **94 runs for FFPE tissues** that include 27 FFPE samples from malignant primary tumors (note the two samples with poor quality that were further excluded), 13 malignant cells microdissected from the lymph nodes, 27 non-malignant cells isolated from the primary tumors, and 27 non-malignant cells from the lymph nodes, whereas the two last graphs of (B) represent the **DDA runs for the 24 buffy coat and 24 saliva samples** (please see **Supplementary Table 1-2**). We therefore ran a total of 142 samples in the discovery phase; 140 of them had a great quality and were selected for the subsequent analysis presented in the manuscript.

3. The labels of Figure 1A bottom graph are very unclear and needs a legend.

Response: We would like to thank the reviewer for pointing this out. **Fig 1A (Fig. 1B** in the new version of the manuscript) was amended and an explanation was added in the figure legend.

4. Many external datasets are used in the current study. When reading the manuscript text, it is hard to follow which data is extracted where and what came out of the analysis. Also for the external data, it wasn't clear how many samples as an input were given.

Response: We thank the reviewer for pointing this out. To clarify the 8 external datasets and the analysis performed, we have added the section “**Evaluation of external datasets**” to the **Methods** and also **Supplementary Table 9** in the revised version of the manuscript.

5. “ ..., as this observation indicates that further studies focusing on CADs may benefit these patients.” (page 8, lines 174-175). It is unclear to which evidence this conclusion is based on.

Response: We thank the reviewer for the comment and that is an interesting point. In this study, the cancer-associated fibroblasts (CAFs) were enriched in the fraction of non-malignant cells isolated from primary tumors of HNSCC patients according to CIBERSORTx analysis. This makes the search for CAF specific biomarkers a promising strategy to study the role of microenvironment and its crosstalk with the tumor cell. Furthermore, this strategy could allow to propose therapeutic strategies to interfere directly in this crosstalk and potentially guide the management of HNSCC patients. Because this subpopulation is enriched across the patients analyzed, both CAF biomarkers and therapeutic target candidates could be very promising¹. For clarity, a sentence considering the therapeutic use of CAFs was added to the manuscript (**Results section “Protein profiles indicate specific immune phenotypes across datasets”**)

6. Figure 1D, only 8 samples are shown. What was the input dataset? This is unclear.

Response: Thanks for this observation and we apologize for the lack of clarity in the text. In the analysis presented in **Fig. 1D** (**Fig. 1E** in the new version of the manuscript), we have used all the 25 bulk proteomes (our data) evaluated for non-malignant cells from primary tumor samples to infer cell type abundance in CIBERSORTx. However, only eight samples returned with significant p-values for the global deconvolution ($p \leq 0.1$), and they were presented in **Fig. 1E**. To make it clear, we have updated the legend from **Fig. 1E**.

7. Page 9, line 177 “Buffy coat prediction...”. Using which dataset is this analysis performed? Only 15 buffy coat samples were analyzed in the current dataset, but here is referred to 16 samples).

Response: We appreciate the reviewer's comment and once again apologize for the negligence when presenting this analysis. As reported in the previous question, the prediction of cell type abundance in CIBERSORTx was also performed considering the 24 bulk proteomes from buffy coat samples (our data). From these, only 16 samples returned significant p-values for the global deconvolution ($p \leq 0.1$), and they were presented in **Fig. 1F** (in the new version of the manuscript). We have updated the legend from **Fig. 1F** to make it clearer. In fact, we analyzed a total of 24 buffy coat samples in the discovery phase. The number “15” mentioned by the referee in this question is the number of buffy coat samples that were paired with saliva samples, which were isolated from the same patients.

8. Fig 1 shows a nice overview about non-malignant cells. What about the malignant cells? Or LNs? Why weren't these analyzed?

Response: Thanks for pointing this out and we are now aware that the cell type inference analysis needs a better description. As we have used laser microdissection to isolate the malignant and non-malignant cells from HNSCC primary tumors and lymph nodes (metastasis), the malignant portion was primarily composed of neoplastic cells, and it was inconsistent to search for additional cell types on this material using the same strategies performed for non-malignant cells. For this reason, we only predicted the immune and non-immune cellular composition considering the proteomes of non-malignant portion of primary tumors (n = 25 samples), non-malignant lymph nodes (n = 27 samples) and buffy coat (n = 24 samples). Therefore, the cell types could be inferred for non-malignant cells isolated from primary tumors (**Fig. 1E**) and buffy coat (**Fig. 1F**). For saliva, we could have analyzed these samples to infer cell types because they are composed of neoplastic and non-neoplastic populations, but we did not pursue that path due to the lack of an appropriate single cell background of saliva cells that could be used as a reference matrix in CIBERSORTx. We have included in the reviewed version of the manuscript an explanation for not considering the malignant cells in the investigation of cell types and also reinforced the lack of an appropriate reference matrix to deconvolute salivary proteomes (**Results section "Protein profiles indicate specific immune phenotypes across datasets"**).

9. Page 9 line 191, 11 pN+ and 14 pN0, were used for the analysis. What samples are these? Why not all the 27 samples that were measured?

Response: We would like to thank the reviewer for this observation. We were really careful with the quality control of samples evaluated in the manuscript. Thus, two primary tumor malignant samples measured in DDA experiments were excluded due to inconsistent detection of control peptides from iRT and trypsin (patients 2875 and 4417), and this resulted in 25 remaining samples that were kept in subsequent analysis (please see **Supplementary Fig. 2 and Results section "Global proteomes are collectively implicated in immune response"**). That is the reason why the comparison of malignant cells from primary tumors between pN+ and pN0 patients was carried out with a total of 25 samples.

10. The supplementary tables don't have a column the fold up/downregulated.

Response: We appreciate the reviewer's comment and the possibility of explaining this point. The differences in protein abundances between two groups have been reported in **Supplementary Tables 2 and 3**, but they were herein termed "Ratio" instead of "Fold change" to avoid confusion with other ways of fold change calculation. By "Ratio", we meant the ratio of the mean LFQ intensities (not log transformed) for group 1 (for example, pN+) over group 2 (for example, pN0) followed by log transformation. This information was included in the **Methods sections "Definition and annotation of metastasis signatures" and "Comparison between tumor and metastasis proteomes"**.

11. Suppl. Table 1-1 and 2, what is this data, RAW? Discovery? This is unclear.

Response: Thanks for this observation and we apologize for the lack of clarity in this Table. In **Supplementary Table 1-1**, we have listed all the 93 patients included in the study with additional data informing the place where samples were collected, tumor site, and lymph node status (N status). The sites from where the samples were collected for every patient, as well as the methodologies used to evaluate these samples, are also presented. In **Supplementary Table 1-2**, we informed the main clinical and pathological features for the HNSCC patients included in this study. For clarity, the patients were separated by the center where the samples were collected.

12. Figure 3 A, legend shows pN+ and pN0, red and blue. What does the colors mean? Up or down regulated? This is unclear

Response: We thank the reviewer for allowing us to make this statement clearer. The red and blue colors in **Fig. 3A (Fig. 3E in the revised manuscript)** correspond to the LFQ abundances of proteins associated with metastasis in our discovery proteomics (DDA) datasets. Higher LFQ levels in pN+ are represented in red and higher LFQ levels in pN0 are in blue. We have included this information in the **Fig. 3E** legend of the new version of the manuscript.

13. Figure 3B, the input was 5 genes, How can 5 different genes separate 6 cell types?

Response: Thanks for pointing this issue out and we apologize for not being clear in the description. In this part of the manuscript, we first defined cluster markers of immune and non-immune cell types using scRNASeq data as markers that designate populations (please see the **Method section “scRNASeq processing and differential expression”**). Then, we compared the list of differentially abundant proteins associated with metastasis in our proteomes (pN+ vs. pN0; **Supplementary Table 2**) with the list of the cluster markers to verify if these proteins are markers of specific immune populations. From the 85 proteins with differential expression between pN+ and pN0 non-malignant cells from primary tumors, five are clusters markers of immune cells, specifically dendritic cells, mast cells, and CD8+ exhausted T cells, indicating that immune populations may be modulated in the metastatic phenotype. Thus, the input for **Fig. 3B (Fig. 2E lower left panel in the new version of the manuscript)** was the list of 85 proteins associated with metastasis in non-malignant primary tumors, and the five proteins were actually the genes considered as cluster markers using scRNASeq data. We amended some related sentences in the man text to clarify this point (**Results section “HNSCC multisites exhibit immune-associated nodal metastasis markers”**; **Methods section “scRNASeq processing and differential expression”**).

14. Figure 4, it is unclear what type of comparison has been made in this figure.

Response: We are really thankful for the opportunity of explaining this point and improving the manuscript based on the reviewer’s comment. In this part of the manuscript (**Fig. 3 in the new version**), we aimed at exploring the protein profile of tumors and lymph nodes to elucidate the effect of tumor cell spread from the primary sites to the nodes. To achieve this, we

compared the proteome composition of two subpopulations (malignant and non-malignant cells) between primary tumors and lymph nodes. To be clearer, we included an experimental design as **Fig. 3A** in the new version of the manuscript and the related sentences were amended in the **Results** section “**Nodal metastasis cells resemble the molecular signature from tumors**”. In addition, to meet the suggestions of reviewers 1 and 2, we have included some confirmatory results of the discovery phase using PRM-MS and RT-qPCR data in **Fig. 3**, as explained in the revised text.

15. Figure 4, lymph node vs primary tumor, green/red is up/down. Does that mean uniquely expressed? Why isn't the fold up/down indicated?

Response: We appreciate the opportunity of clarifying this point. The comparison between lymph node and primary tumor proteomes returned a list of differentially abundant proteins revealed by the statistical analysis ($q \leq 0.05$; Benjamini-Hochberg test) or detected exclusively in one group, i.e., primary tumor or lymph node. Thus, proteins exhibiting higher expression in lymph nodes compared to that in primary sites or those exclusively detected in lymph nodes were named “upregulated”, while proteins exhibiting lower expression in lymph nodes compared to that in primary sites or those exclusively detected in the tumor site were termed “downregulated” (**Methods** section “**Comparison between tumor and metastasis proteomes**”). The “up- and downregulation” information previously presented in **Fig. 4A-B** was kept only in **Supplementary Tables 2-6 and 2-7**, as suggested by reviewer 2. The fold changes available for the differentially abundant proteins (but not for the ‘exclusive’ proteins once they were detected exclusively in one group) are indicated on these tables and termed “Ratio” instead of “Fold change” to avoid confusion with other ways of fold change calculations.

16. Figure 4, legend says not detected. Do the authors mean not measured instead?

Response: We thank the reviewer for this comment. The reviewer is correct and by “Not detected” we meant proteins that could not be measured by LC-MS/MS in a given sample during data acquisition. However, as suggested by one of the reviewers, **Fig. 4A- B** from the old version of the manuscript was simplified as **Fig. 3D** in the revised article, and the information “Not detected” was removed. Anyway, these “Not measured” proteins are represented as a missing value in the search file and referred as “NaN” in **Supplementary Table 2-6 and 2-7** and **Supplementary Table 3-2 and 3-3**.

17. Figure 4B, the comparison is again unclear. Upregulated in what sample type, lymph node?

Response: Thanks for pointing this out and we apologize for the lack of clarity in the figure. We compared cells from lymph nodes vs. primary tumors to determine signatures associated with tumor spread. Thus, by “upregulated” we meant “upregulated in lymph node” and the same is true for “downregulated”. As a suggestion of reviewer 2, we have reformulated **Fig. 4** and the information of deregulated proteins was moved to **Supplementary Table 3-1 to 3-3**. In this **Supplementary Table**, we incorporated your important suggestion and mentioned

“upregulated in lymph node” or “downregulated in lymph node” in the abundance columns of **Sheets 2- 3**.

18. Figure 4c, it was indicated that malignant cells in the LNs weren't measured, but here they are indicated as being measured. This is unclear to me. What are the malignant cells in the LNs in this graph?

Response: We thank the reviewer for the comment and apologize for the unclear description. In this part of the manuscript, we have compared the proteomes of tumor and lymph node sites to elucidate signatures associated with tumor spread (lymph nodes vs. primary tumors; $p \leq 0.05$; two-sided unpaired Student's t-test or proteins detected exclusively in one group). For this, we used data from a 27-patient HNSCC cohort, being 11 patients positive (pN+) and 16 patients negative (pN0) for lymph node metastasis. The analysis was conducted separately for the malignant portion (11 matched samples from primary tumor and 11 samples from lymph nodes) and for the non-malignant cells (27 matched samples from primary tumor and 27 samples from lymph nodes). Answering the reviewer's question, the malignant cells from lymph nodes are metastasis cells that were recovered from the 11 metastatic lymph nodes (pN+) and compared to the matched primary tumor malignant counterpart that was available only for these 11 patients. Conversely, all the patients had the non-malignant cells isolated from the lymph nodes ($n = 27$; both pN+ and pN0 patients) and thus they were compared with the non-malignant counterpart from the primary tumor of all the HNSCC patients. To be clearer, we included an experimental design as **Fig. 3A** in the new version of the manuscript.

19. Figure 5C, what does the size of the dots indicate in this graph?

Response: We thank the reviewer for the opportunity of explaining this point. In PCA analysis, concentration ellipses were drawn around the categories pN+ and pN0 and the means of groups' observations, i.e., values for an pN+ or pN0 “average individual”, were displayed by default as larger red or blue circles. This question from the reviewer made us aware that the mean dots may cause confusion in the readership besides do not adding any important knowledge in the PCA since they do not represent patients. So, we have updated the manuscript with a new version of the PCA graphs plotted without the mean points and now the size of all patients is the same (**Fig. 4C** in the revised article).

20. Figure 7, why is PCR used for the measurement of the proteins, and not ELISA? We know that gene expression usually doesn't correlate well to protein expression.

Response: We thank the reviewer for raising this interesting question. We have opted to use PRM-MS, SRM-MS and RT-qPCR to confirm discovery proteome and transcript data, respectively, since most selected proteins are intracellular or located at the membrane, and thus could not be properly measured by ELISA. Also, PRM-MS and SRM-MS allows the analysis to be done in a multiplex manner, facilitating the concomitant measurement of all selected proteins in the same sample. Another advantage of performing these targeted methodologies for the validation of the selected targets is that they are more cost-effective. Also, we have some limitations when performing ELISA compared to the other two methodologies, once (i)

you require a validated antibody for each target, and (ii) it is recommendable to work with a sample in which the protein of interest is soluble and in its native form can be recognized by the antibody. For all these reasons, we opted to perform PRM-MS, SRM-MS and RT-qPCR as the targeted methodologies in this study.

We understand the reviewer's concern regarding the use of transcripts to verify proteomics data because there's a moderate degree of correlation between proteome and transcriptome levels, as shown in the literature and in our manuscript ($\rho = 0.53$; $p \leq 0.05$) (**Supplementary Fig. 3A**). However, we have several other reasons to not see this question as a concern at this point of the manuscript because (i) the microenvironment markers SRSF1, SRSF2, SRSF3, SRSF5, TRA2A, and CD209 selected in the discovery proteomics have also been tested for transcription levels in the FFPE tissues in the discovery phase and there was a significant association with nodal status, as well as observed for proteins (p or $q \leq 0.05$; **Supplementary Fig. 6C-D**), and (ii) transcript levels of the 6 microenvironment markers could discriminate pN+ and pN0 HNSCC patients with elevated AUCs (mean AUC = 0.85), thus indicating the high performance of the six proteins in distinguishing patients according to pN (**Fig. 4F**).

21. Figure 6 and 7, were these selected proteins differentially expressed between cancer and normal samples? Why were these selected, what criteria did they fall into?

Response: We thank the reviewer for the opportunity of clarifying this point. The description of the criteria used to select the prognostic targets for the verification step, i.e. SRM-MS and RT-qPCR followed by machine learning analysis, are described in the **Methods section "Selection of targets for ML analysis"** of the manuscript. To make it clearer, we have also added the sentence "**please see Methods section Selection of targets for ML analysis**" in the appropriate results sentences that referred to target selection (**Results sections "HNSCC multisites exhibit immune-associated nodal metastasis markers", "Microenvironment proteomes group samples according to metastasis and highlight candidate markers of locoregional spread", and "Metastasis-associated tissue targets can be detected in liquid biopsies"**). These molecules were evaluated in fluids with the final aim of testing their ability to discriminate pN+ and pN0 patients in a machine learning model.

Reference Reviewer #3

- 1 Chen, X. & Song, E. Turning foes to friends: targeting cancer-associated fibroblasts. *Nat Rev Drug Discov* **18**, 99-115, doi:10.1038/s41573-018-0004-1 (2019).

REVIEWERS' COMMENTS

Reviewer #1 (Remarks to the Author):

In this revised version of the manuscript, the authors have expended significant effort to address the concerns raised by the reviewers, resulting in a much improved manuscript. While most of my concerns have been satisfactorily addressed, there are still a couple where my original issue has not been fully addressed, or the action/revision is unclear.

* Permutation test and p-values.

From the current description in the methods, it is unclear how the permutation test is performed:

"... we executed a permutation test (PT) to calculate a p-value against the null hypothesis that the data from each pair and the classes pN+/pN0 are independent".

In general, a permutation test scrambles the data (here, the pN+/pN0 labels) and repeats the entire analysis. The scores obtained using the permuted data are compared to the actual score (from the real data) to calculate a p-value. For example, for the 111 permutations the authors use:

Let A_t be the true Top-1 ROC AUC for pair $\langle S_i, C_j \rangle$

Let the Top-1 ROC AUCs from the permuted datasets be P_1, P_2, \dots, P_{111} (set of P's)

Then, permutation p-value = $(\text{SUM } I(P_i > A_t)) / 111$

where the SUM extends over $i = 1, 2, \dots, 111$;

$I(x)$ is the indicator function which has a value of 1 if x is true and 0 otherwise.

Is this the procedure that the authors followed? If not, please include details on how exactly the permutation p-values were calculated.

* Paired analysis.

The authors have dismissed the possibility of a paired analysis based on a protein by protein application of the t-test using paired samples. A more general mixed effects model treating patient ID as a random effect would overcome the restriction of having missing values.

Reviewer #2 (Remarks to the Author):

The authors have addressed my primary concerns for the manuscript and significantly improved the description and statistical analysis presented throughout the proposal, while also toning down the strength of language used when describing the results, which was appropriate given the data set presented.

Reviewer #4 (Remarks to the Author):

This is a nice manuscript by Busso-Lopes and colleagues that utilizes a richly annotated cohort of head and neck cancer tissues for downstream proteomics analysis. The authors perform LCM to enrich tumor and microenvironment compartments of both primary tumor, TME and lymph node metastasis. In addition, saliva and buffy coat samples were analyzed with the goal to develop liquid biopsies for lymph node metastasis. In addition to their own data, a number of external datasets are used for integration and data mining.

While the cohort is relatively small, the ample size is well in-line with many other clinical tissue proteomics datasets, especially those that utilize LCM for specific compartment enrichments. Considering the small amounts of samples available from LCM and the applied LC-MS technology, the results are of excellent quality.

The authors performed a number of standard comparative analysis (i.e., clustering, differential analysis, enrichment analysis, etc.), but also perform some novel and potentially interesting comparisons to scRNAseq data. From what I can read in the extensive rebuttal letter and the associated primary results, the authors have nicely responded to most reviewer questions. While there are many additional analysis and validations that could be performed, the current manuscript compares favorably to many similar tissue proteomics project that have been published in the last 5 years. I hence recommend acceptance of the manuscript in Nature Communications.

There are a couple of minor comments the authors should consider.

1) Page 4, line 82: "The composition of body fluids, which is able to wire diverse microenvironments, can also be affected by cancer". Not completely clear what this sentence means.

2) Lines 231-235: I would phrase this observation more cautiously.

- 3) Line 250: Remove “signature”. This is not a signature, but rather 23 commonly differentially expressed proteins.
- 4) Line 254-255: ...”and this strengthens the relevance of the immune response in HNSCC multisites”. I would again recommend to phrase this a bit more cautious.
- 5) Figure 2F: Couldn’t the authors simply plot actual counts of these immune populations? It would be significantly easier to interpret for most readers of this paper.
- 6) Supplemental Figure 3F: color scale needs a label? Maybe better plotted as a dot map for simplicity to readers?
- 7) Lines 327-329: Rephrase to something along the lines of: “The current proteomics data suggest that less proteomics changes are observed in the primary versus metastatic tumor compartment compared to non-malignant stromal cells”. Or something along these lines.
- 8) Line 589: replace “high-performance” with “best”.
- 9) While I don’t think there are major technical issues with the signature generation associated with Figure 6 (or as the authors call it “machine learning”). I would add a note of caution to the ROC curves in Figure 6d. Basically the authors use differentially expressed proteins/RNAs and use targeted approaches to verify their expression levels in the same samples and then evaluate what’s the best combination. AUC >0.9 would be amazing, but in an independent cohort of several hundred patients it would most likely be significantly lower. Just to be clear. I am not saying to remove them, and I am also not suggesting that these marker combinations could not be useful in the clinic, but at this very moment they are more or less “candidates” that will need multiple rounds of rigorous validations. And as we all know, this is extremely difficult, time consuming and expensive.

POINT-BY-POINT RESPONSE TO THE REVIEWER'S COMMENTS

We appreciate the reviewers' kindly help to improve the manuscript. A new version of the revised text has been uploaded with changes highlighted in grey color.

REVIEWERS' COMMENTS

Reviewer #1 (Remarks to the Author):

In this revised version of the manuscript, the authors have expended significant effort to address the concerns raised by the reviewers, resulting in a much improved manuscript. While most of my concerns have been satisfactorily addressed, there are still a couple where my original issue has not been fully addressed, or the action/revision is unclear.

Response: We deeply appreciate the reviewer's feedback and the additional suggestions.

* Permutation test and p-values.

From the current description in the methods, it is unclear how the permutation test is performed:

"... we executed a permutation test (PT) to calculate a p-value against the null hypothesis that the data from each pair and the classes pN+/pN0 are independent".

In general, a permutation test scrambles the data (here, the pN+/pN0 labels) and repeats the entire analysis. The scores obtained using the permuted data are compared to the actual score (from the real data) to calculate a p-value. For example, for the 111 permutations the authors use:

Let A_t be the true Top-1 ROC AUC for pair $\langle S_i, C_j \rangle$

Let the Top-1 ROC AUCs from the permuted datasets be P_1, P_2, \dots, P_{111} (set of P's)

Then, permutation p-value = $(\text{SUM } I(P_i > A_t)) / 111$

where the SUM extends over $i = 1, 2, \dots, 111$;

$I(x)$ is the indicator function which has a value of 1 if x is true and 0 otherwise.

Is this the procedure that the authors followed? If not, please include details on how exactly the permutation p-values were calculated.

Response: We apologize for the lack of explanation regarding the permutation test in the machine learning analysis. In fact, we used the method described by the reviewer and it was implemented by the sklearn programming kit version 0.23.1 (1). The method reference was included in the **Methods** section "Definition of prognostic signatures using machine learning".

* Paired analysis.

The authors have dismissed the possibility of a paired analysis based on a protein by protein application of the t-test using paired samples. A more general mixed effects model treating patient ID as a random effect would overcome the restriction of having missing values.

Response: We really thank the reviewer for this important suggestion to help us dealing with missing data. Indeed, missing values are frequent in mass spectrometry-based proteomics and their handling is not trivial. A popular strategy to deal with the incomplete data in proteomics is to apply imputation, but there is not a consensus about the best imputation strategies, and poor-performing imputation methods may bias subsequent analysis steps, making missing value analysis an ongoing cause for

debate within the bioinformatics community (2-6). Due to these uncertain impacts of imputation in proteomics data, some authors recommend avoiding imputation for the analysis tools that do not require it (5), and this is how we have been handling missing data in our group for the statistical comparison of proteome abundances. Nonetheless, we really appreciate the reviewer's suggestion of employing a mixed effects model as a way to overcome the missing value issue. Even though the application of mixed models in proteomics research is still incipient (7) some methods have been tested and are promising to analyze proteomes containing missing values (8). Thus, we will strongly consider the reviewer's recommendation in further studies. Finally, we would like to comment that even though we adopted a more conservative position for the statistical analysis performed in this part of the work, what is evidenced by not imputing missing values or even not benefitting from the increased statistical power of a paired analysis, our results were not penalized, and we still presented some valuable data that helped to understand the biology behind HNSCC primary tumors and lymph node metastases sites.

References Reviewer #1

1. Ojala MG, Gemma C. Permutation Tests for Studying Classifier Performance. *Journal of Machine Learning Research*. 2010;11:1833 - 63.
2. Välikangas T, Suomi T, Elo LL. A comprehensive evaluation of popular proteomics software workflows for label-free proteome quantification and imputation. *Brief Bioinform*. 2018;19(6):1344-55.
3. Shen M, Chang YT, Wu CT, Parker SJ, Saylor G, Wang Y, et al. Comparative assessment and novel strategy on methods for imputing proteomics data. *Sci Rep*. 2022;12(1):1067.
4. Egert J, Brombacher E, Warscheid B, Kreutz C. DIMA: Data-Driven Selection of an Imputation Algorithm. *J Proteome Res*. 2021;20(7):3489-96.
5. Webb-Robertson BJ, Wiberg HK, Matzke MM, Brown JN, Wang J, McDermott JE, et al. Review, evaluation, and discussion of the challenges of missing value imputation for mass spectrometry-based label-free global proteomics. *J Proteome Res*. 2015;14(5):1993-2001.
6. Liu M, Dongre A. Proper imputation of missing values in proteomics datasets for differential expression analysis. *Brief Bioinform*. 2021;22(3).
7. Liu CW, Bramer L, Webb-Robertson BJ, Waugh K, Rewers MJ, Zhang Q. Temporal expression profiling of plasma proteins reveals oxidative stress in early stages of Type 1 Diabetes progression. *J Proteomics*. 2018;172:100-10.
8. Välikangas T, Suomi T, Chandler CE, Scott AJ, Tran BQ, Ernst RK, et al. Enhanced longitudinal differential expression detection in proteomics with robust reproducibility optimization regression. *bioRxiv*. 2021:2021.04.19.440388.

Reviewer #2 (Remarks to the Author):

The authors have addressed my primary concerns for the manuscript and significantly improved the description and statistical analysis presented throughout the proposal, while also toning down the strength of language used when describing the results, which was appropriate given the data set presented.

Response: We would like to thank the reviewer for the feedback and the excellent suggestions that improved the manuscript.

Reviewer #4 (Remarks to the Author):

This is a nice manuscript by Busso-Lopes and colleagues that utilizes a richly annotated cohort of head and neck cancer tissues for downstream proteomics analysis. The authors perform LCM to enrich tumor and microenvironment compartments of both primary tumor, TME and lymph node metastasis. In addition, saliva and buffy coat samples were analyzed with the goal to develop liquid biopsies for lymph node metastasis. In addition to their own data, a number of external datasets are used for integration and data mining.

While the cohort is relatively small, the ample size is well in-line with many other clinical tissue proteomics datasets, especially those that utilize LCM for specific compartment enrichments. Considering the small amounts of samples available from LCM and the applied LC-MS technology, the results are of excellent quality.

The authors performed a number of standard comparative analysis (i.e., clustering, differential analysis, enrichment analysis, etc.), but also perform some novel and potentially interesting comparisons to scRNAseq data. From what I can read in the extensive rebuttal letter and the associated primary results, the authors have nicely responded to most reviewer questions. While there are many additional analysis and validations that could be performed, the current manuscript compares favorably to many similar tissue proteomics project that have been published in the last 5 years. I hence recommend acceptance of the manuscript in Nature Communications.

Response: We thank the reviewer for the valuable suggestions and appreciate the time you took to review our work.

There are a couple of minor comments the authors should consider.

1) Page 4, line 82: “The composition of body fluids, which is able to wire diverse microenvironments, can also be affected by cancer”. Not completely clear what this sentence means.

Response: We thank the reviewer for the opportunity of improving this point and we agree that the information is confusing. For clarity, the sentence was reformulated in the **Introduction section** of the revised manuscript.

2) Lines 231-235: I would phrase this observation more cautiously.

Response: That is a great point, and we appreciate the reviewer’s comment. Based on the higher counts of proteins upregulated in the pN+ condition for non-malignant cells isolated from primary sites and lymph nodes, we proposed that, in general, the protein function is upregulated in these two microenvironments. However, we agree with the reviewer’s comment that we cannot make a rule that protein upregulation will necessarily increase its biological functions because there is a high complexity of the proteome associated with the protein functionalities (1). Thus, we removed the sentence “indicating that overall protein function is upregulated in these microenvironments” and updated the remaining paragraph to improve clarity (**Results section “HNSCC multisites exhibit immune-associated nodal metastasis markers”**).

3) Line 250: Remove “signature”. This is not a signature, but rather 23 commonly differentially expressed proteins.

Response: We agree with the reviewer and really appreciate the suggestion. The word “signature” was removed, and the sentence was reformulated in the **Results section “HNSCC multisites exhibit immune-associated nodal metastasis markers”**.

4) Line 254-255: ...”and this strengthens the relevance of the immune response in HNSCC multisites”. I would again recommend to phrase this a bit more cautious.

Response: Thank you for pointing this out and we do understand that the sentence may represent an overstated claim. As suggested, we have rephrased the sentence and the words “strengthens the relevance” were replaced by “suggests a role” (**Results section “HNSCC multisites exhibit immune-associated nodal metastasis markers”**).

5) Figure 2F: Couldn't the authors simply plot actual counts of these immune populations? It would be significantly easier to interpret for most readers of this paper.

Response: We thank the reviewer for this suggestion. The radar plot from **Fig. 2f** was replaced by a bar plot to make the results clearer.

6) Supplemental Figure 3F: color scale needs a label? Maybe better plotted as a dot map for simplicity to readers?

Response: We appreciate the great suggestion and apologize for the missing label. The color scale refers to the Spearman correlation coefficients generated for the comparison of nodal status and immune populations. This information was added to the figure and the heatmap was updated as a dot map, as suggested by the reviewer.

7) Lines 327-329: Rephrase to something along the lines of: “The current proteomics data suggest that less proteomics changes are observed in the primary versus metastatic tumor compartment compared to non-malignant stromal cells”. Or something along these lines.

Response: Thank you for the relevant suggestion to improve this part of the manuscript. The sentence was amended accordingly in the **Results section “Nodal metastasis cells resemble the molecular signature from tumors”**.

8) Line 589: replace “high-performance” with “best”.

Response: We appreciate the reviewer suggestion. As our version did not have the expression “high-performance” on the line 589, we have replaced the closest sentence that was on the line 586 (**Results section “Machine learning predicts liquid biopsy-metastasis signatures with high performance”**).

9) While I don't think there are major technical issues with the signature generation associated with Figure 6 (or as the authors call it “machine learning”). I would add a note of caution to the ROC curves in Figure 6d. Basically the authors use differentially expressed proteins/RNAs and use targeted approaches to verify their expression levels in the same samples and then evaluate what's the best combination. AUC >0.9 would be amazing, but in an independent cohort of several hundred patients it would most likely be significantly lower. Just to be clear. I am not saying to remove them, and I am also not suggesting that these marker combinations could not be useful in the clinic, but at this

very moment they are more or less “candidates” that will need multiple rounds of rigorous validations. And as we all know, this is extremely difficult, time consuming and expensive.

Response: We thank the reviewer for raising this excellent observation. Indeed, the concerns about ML were also pointed out by reviewers #1 and #2, and we are aware that substantial work is still required before a possible clinical application of these signatures, including the evaluation in larger independent cohorts together with assay optimizations. To make the limitations of this part of the work even clear, we moved the following sentence from the **Methods** to the **Results** section where **Fig. 6** is explained (section “**Machine learning predicts liquid biopsy-metastasis signatures with high performance**”).

“Due to the sample size limitation for this machine learning analysis, we did not intend to select one model, but to report a set with many candidates’ high-performance signatures that could be used in the decision-making process regarding future studies with larger samples sizes.”

We also would like to mention that, for clarity, the following sentence had been added in the previous revised version of the manuscript (**Discussion section**).

“Overall, the ML analysis served to understand our datasets and the complexities and noises that differentiate the type of data sources evaluated, revealing promising metastasis-dependent signatures that can non-invasively guide the decision-making process for HNSCC patients. Indeed, substantial work is still required before a possible clinical application of these signatures, and this include the evaluation in larger independent cohorts together with assay optimizations.”

References Reviewer #4

1. Harper JW, Bennett EJ. Proteome complexity and the forces that drive proteome imbalance. Nature. 2016;537(7620):328-38.